# BINARY LOSSES FOR DENSITY RATIO ESTIMATION

**Werner Zellinger**
LIT AI Lab, Institute for Machine Learning
Johannes Kepler University Linz, Austria
`zellinger@ai-lab.jku.at`

## ABSTRACT

Estimating the ratio of two probability densities from a finite number of observations is a central machine learning problem. A common approach is to construct estimators using binary classifiers that distinguish observations from the two densities. However, the accuracy of these estimators depends on the choice of the binary loss function, raising the question of which loss function to choose based on desired error properties. For example, traditional loss functions, such as logistic or boosting loss, prioritize accurate estimation of small density ratio values over large ones, even though the latter are more critical in many applications.

In this work, we start with prescribed error measures in a class of Bregman divergences and characterize all loss functions that result in density ratio estimators with small error. Our characterization extends results on composite binary losses from Reid & Williamson (2010) and their connection to density ratio estimation as identified by Menon & Ong (2016). As a result, we obtain a simple recipe for constructing loss functions with certain properties, such as those that prioritize an accurate estimation of large density ratio values. Our novel loss functions outperform related approaches for resolving parameter choice issues of 11 deep domain adaptation algorithms in average performance across 484 real-world tasks including sensor signals, texts, and images.

## 1 INTRODUCTION

Estimating the Radon-Nikodým derivative $\beta := \frac{\mathrm{d}P}{\mathrm{d}Q} : \mathcal{X} \to \mathbb{R}$ of some probability measure $P$ with respect to another probability measure $Q$, from iid observations $(x_i)_{i=1}^n$ and $(x_i')_{i=1}^m$ of the two measures, respectively, is a common machine learning task; with applications in two-sample testing (Neyman & Pearson, 1933; Kanamori et al., 2011; Hagrass et al., 2022), anomaly detection (Smola et al., 2009; Hido et al., 2011), divergence estimation (Nguyen et al., 2010; Rhodes et al., 2020), covariate shift adaptation (Shimodaira, 2000; Dinu et al., 2023; Gruber et al., 2024), energy-based modeling (Gutmann & Hyvärinen, 2012; Tu, 2007), generative modeling (Mohamed & Lakshminarayanan, 2016; Grover et al., 2019), conditional density estimation (Schuster et al., 2020), and classification from positive and unlabeled data (Kato et al., 2019). When both measures $P$ and $Q$ have densities w.r.t. the Lebesgue reference measure, then $\beta$ equals the *ratio of their densities*.

A large class of methods for density ratio estimation follows Algorithm 1 to construct an estimator $\widehat{\beta}$ of $\beta$ from binary classifiers which separate the observations $(x_i)_{i=1}^n$ of $P$ from the observations $(x_i')_{i=1}^m$ of $Q$ (Qin, 1998; Bickel et al., 2007; Cheng & Chu, 2004; Sugiyama et al., 2012a; Menon & Ong, 2016; Kato & Teshima, 2021; Zellinger et al., 2023). The estimator $\widehat{\beta}$ constructed by Algorithm 1 converges (for $m, n \to \infty$) to the minimizer of a Bregman divergence[1]

$$B_\phi(\beta, \widehat{\beta}) := \mathbb{E}_{x \sim Q}\left[\phi(\beta(x)) - \phi(\widehat{\beta}(x)) - \phi'(\widehat{\beta}(x))(\beta(x) - \widehat{\beta}(x))\right] \tag{1}$$

with generator $\phi : \mathbb{R} \to \mathbb{R}$ (Sugiyama et al., 2012b; Menon & Ong, 2016; Zellinger et al., 2023).

---

[1] Assuming $\beta \in L^1(Q)$ and defining $F_\phi : L^1(Q) \to \mathbb{R}$ by $F_\phi(h) := \int_{\mathcal{X}} \phi(h(x))\, \mathrm{d}Q(x)$ identifies Eq. 1 as a Bregman divergence (Bregman, 1967) of the form $B_\phi(\beta, \widehat{\beta}) = F_\phi(\beta) - F_\phi(\widehat{\beta}) - \nabla F_\phi(\widehat{\beta})[\beta - \widehat{\beta}]$.

---

**Algorithm 1:** Density ratio estimation by binary classification

---

**Setup** : Loss function $\ell : \{-1, 1\} \times \mathbb{R} \to \mathbb{R}$, invertible probability link function $\Psi^{-1} : \mathbb{R} \to [0, 1]$ and function class $\mathcal{F} \subset \{f : \mathbb{R} \to \mathbb{R}\}$.

**Input** : Observations $(x_i)_{i=1}^n \sim P$ and $(x_i')_{i=1}^m \sim Q$.

**Output:** Estimator $\widehat{\beta} : \mathcal{X} \to \mathbb{R}$ of Radon-Nikodým derivative $\beta = \frac{\mathrm{d}P}{\mathrm{d}Q}$.

**Step 1** : Compute binary classifier

$$\widehat{f} := \arg\min_{f \in \mathcal{F}} \frac{1}{m+n} \sum_{(x,y) \in (x_i, 1)_{i=1}^n \cup (x_i', -1)_{i=1}^m} \ell(y, f(x)) \tag{2}$$

and estimate by $\Psi^{-1}\big(\widehat{f}(x)\big)$ the probability[2] $\rho(y = 1 | x)$ that $x$ is drawn from $P$.

**Step 2** : Construct $\widehat{\beta}$ using Bayes theorem $\widehat{\beta}(x) := \frac{\Psi^{-1}(\widehat{f}(x))}{1 - \Psi^{-1}(\widehat{f}(x))} \approx \frac{\rho(y=1|x)}{\rho(y=-1|x)} = \beta(x)$.

---

However, the error measure $B_\phi$ in Eq. 1 is only implicitly defined by the choice of the binary loss function $\ell$ used to instantiate Algorithm 1. That is, choosing the wrong loss function might lead to unexpected behavior of the density ratio estimator. For example, typical loss functions for Algorithm 1 are the logistic loss used in (Bickel et al., 2009, Section 7), the boosting loss applied in (Menon & Ong, 2016, Section 7), the square loss (Menon & Ong, 2016, Table 1) and logarithm-based loss functions (Nguyen et al., 2007; 2010; Sugiyama et al., 2008a). All of the loss functions above lead to error measures in Eq. 1 that put higher weight on smaller values of the density ratio, despite the fact that large values are crucial in many applications, see (Menon & Ong, 2016).

In this work, instead of choosing a classical loss function for Algorithm 1, we propose to start from an error measure $B_\phi$ in Eq. 1. Our first contribution is, given such an error measure, to combine and extend results from Reid & Williamson (2010; 2011) and Menon & Ong (2016) for characterizing all loss functions that lead to a minimization of this measure $B_\phi$ by Algorithm 1. More precisely, given some (strictly convex) Bregman generator $\phi : [0, \infty) \to \mathbb{R}$ and some (monotonically increasing) density ratio link $g : \mathbb{R} \to \mathbb{R}$, we provide the *unique* form of a (strictly proper composite) loss function $\ell : \{-1, 1\} \times \mathbb{R} \to \mathbb{R}$ such that Algorithm 1 minimizes $B_\phi(\beta, \widehat{\beta})$ with $\widehat{\beta} := g \circ \widehat{f}$. The link function is given by $\Psi^{-1}(x) := \frac{g(x)}{1+g(x)}$. Following Reid & Williamson (2010), we can show that the choice $g(\widehat{y}) := (\phi')^{-1}(\widehat{y})$ leads to convex loss functions.

Our characterizations provide us with a simple recipe for constructing convex loss functions that allow Algorithm 1 to find estimators with certain properties. Our second contribution is to provide novel loss functions that (in contrast to classical losses) prioritize an accurate estimation of large density ratio values over smaller ones, see Figure 1.

One important application of density ratio estimation is importance weighting. Our third contribution is to show that our novel loss functions can lead to improvements for the parameter selection procedures of Sugiyama et al. (2007); Dinu et al. (2023) when their importance weights are computed with our novel loss function instead of the ones of Example 1. More precisely, importance weighted validation (Sugiyama et al., 2007) (respectively importance weighted aggregation (Dinu et al., 2023)) selects better parameters on two (respectively three) out of three datasets when it is based on our loss function instead of the ones in Example 1; where "better" refers to a higher accuracy on average over 11 deep learning algorithms and all domain adaptation tasks of three datasets for text data (Blitzer et al., 2006), human body sensor signals (Stisen et al., 2015; Ragab et al., 2023) and images (Peng et al., 2019; Zellinger et al., 2021).

---

[2]Denote by $y = 1$ the event "drawn from $P$" and by $y = -1$ the event "drawn from $Q$". Then, the probability measure $\rho$ on $\mathcal{X} \times \{-1, 1\}$ is defined by $\rho(x|y = 1) := P(x)$, $\rho(x|y = -1) := Q(x)$ and the marginal (w.r.t. $y$) Bernoulli measure $\rho_{\mathcal{Y}}$ realizing $y = 1$ with probability $\frac{1}{2}$ and $y = -1$ otherwise. It follows that $\beta(x) = \frac{\rho(x|y=1)}{\rho(x|y=-1)} = \frac{\rho(x,y=1)\rho_{\mathcal{Y}}(y=-1)}{\rho(x,y=-1)\rho_{\mathcal{Y}}(y=1)} = \frac{\rho(x,y=1)}{\rho(x,y=-1)} = \frac{\rho(x,y=1)\rho_{\mathcal{X}}(x)}{\rho(x,y=-1)\rho_{\mathcal{X}}(x)} = \frac{\rho(y=1|x)}{\rho(y=-1|x)}$.

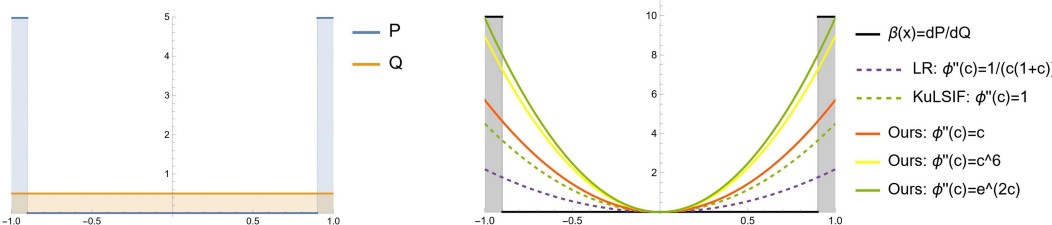

Figure 1: Left: Two piecewise constant probability measures $P$ and $Q$ on $[-1, 1]$. Right: Estimators $\widehat{\beta}(x) := kx^2 + d$ of density ratio $\beta = \frac{\mathrm{d}P}{\mathrm{d}Q}$ with $k, d > 0$ computed by Algorithm 1 (for $m + n \to \infty$). Compared to LR (Bickel et al., 2009) and KuLSIF (Kanamori et al., 2009), our estimators originate from error measures $B_\phi$ in Eq. 1 with increasing weight functions $\phi''(c) = c, c^6, e^{2c}$ and consequently obtain better estimates for large values $\beta(x), x \in [-1, -0.9] \cup [0.9, 1]$, see Section 5.

Summarizing above, we provide a complete set of techniques for designing binary loss functions for Algorithm 1 that (a) prioritize the estimation of large density ratio values and (b) allow high performance in practice. Our three key contributions are:

- Characterization (Section 4): Given an error measure $B_\phi(\beta, \widehat{\beta})$, we extend results from Reid & Williamson (2010; 2011); Menon & Ong (2016) and characterize all loss functions that lead to a minimization of $B_\phi(\beta, \widehat{\beta})$ by Algorithm 1.

- Loss Functions (Section 5): We design novel loss functions with increasing weight functions $\phi''$ (cf. Figure 1).

- Experiments (Section 6): We obtain state-of-the-art average performance in benchmark experiments for re-solving parameter choice issues in deep domain adaptation; involving 9174 neural networks, 484 real-world tasks from three datasets for texts, images and human body sensor signals.

## 2 RELATED WORK

It has been first observed by Sugiyama et al. (2012b) that many density ratio estimation methods minimize a Bregman divergence of the form of Eq. 1, see Example 1.

**Example 1.**

- *Following (Sugiyama et al., 2012b, Section 3.2.1), the kernel unconstrained least-squares importance fitting procedure (KuLSIF) of Kanamori et al. (2009) uses $\ell(1, y) := -y$, $\ell(-1, y) := \frac{1}{2}y^2$ and the link function $\Psi^{-1}(y) := \frac{y}{1+y}$ in Algorithm 1. Eq. 1 is realized with $\phi(x) = (x - 1)^2/2$ and $\widehat{\beta}(x) = \widehat{f}(x)$ (cf. Kanamori et al. (2012, Eq. (4))) such that $B_\phi(\beta, \widehat{\beta}) = \frac{1}{2}\|\beta - \widehat{\beta}\|_{L^2(Q)}^2$.*

- *The logistic regression (Nelder & Wedderburn, 1972) approach (LR) applied in (Bickel et al., 2009, Section 7) realizes Algorithm 1 with $\ell(1, y) := \log(1 + e^{-y})$, $\ell(-1, y) := \log(1 + e^y)$ and link $\Psi^{-1}(y) := (1 + e^{-y})^{-1}$. The implicit error measure is Eq. 1 with $\phi(x) = x \log(x) - (1 + x) \log(1 + x)$ and $\widehat{\beta}(x) = e^{\widehat{f}(x)}$; see (Sugiyama et al., 2012b, Section 3.2.3).*

- *The Kullback-Leibler estimation procedure (KLest) of Nguyen et al. (2010) uses $\ell(1, y) := -\log(x)$, $\ell(-1, y) := y$ and $\Psi^{-1}(y) := \frac{y}{1+y}$, leading to Eq. 1 with $\phi = x \log(x) - x$ and $\widehat{\beta} = \widehat{f}$; see (Sugiyama et al., 2012b, Section 3.2.4) and cf. (Sugiyama et al., 2008b).*

- *The exponential loss (Boost) defined by $\ell(1, y) := e^{-y}$ and $\ell(-1, y) := e^y$, as applied in AdaBoost (Freund & Schapire, 1995), is used in (Menon & Ong, 2016, Section 7) with link $\Psi^{-1}(y) := (1 + e^{-2y})^{-1}$. Eq. 1 is used with $\phi(x) = x^{-3/2}$ and $\widehat{\beta}(x) = e^{2\widehat{f}(x)}$.*

A general formula of the Bregman divergence for given (strictly proper composite) loss functions has been derived by Menon & Ong (2016) based on characterizations of Reid & Williamson (2010). We extend their result by showing that the constructions in (Menon & Ong, 2016, Appendix B) are

in fact *necessary* if their loss representation (Menon & Ong, 2016, Proposition 3) holds. Our proofs are based on the theoretical works of Shuford Jr et al. (1966); Savage (1971); Gneiting & Raftery (2007); Buja et al. (2005); Reid & Williamson (2010); Bao (2023) on proper loss functions.

There is a line of research concerned with the sampling behavior of algorithms following Eq. 1. Specifically, for the special case of KuLSIF, error rates (for $m + n \to \infty$) have been discovered by Kanamori et al. (2012); Gizewski et al. (2022); Nguyen et al. (2024). These results have been extended by Zellinger et al. (2023); Gruber et al. (2024) to the general case of Eq. 1 for loss functions that satisfy a self-concordance property. Although, our (continuous) analysis relies on similar theoretical arguments, except for the polynomial case of $k = 0$, our novel loss functions are not self-concordant. That is, these results do not apply without modification.

Situations of quite dis-similar distributions have been recently discussed by Rhodes et al. (2020) (density-chasm problem), Kato & Teshima (2021); Kiryo et al. (2017) (PU-learning) and Srivastava et al. (2023). Our work can be seen as analysis of a sub-problem of these works, where $P$ is large and $Q$ is small. Choi et al. (2021) propose an approach by training normalizing flows to obtain closer and simpler densities.

Another recent line of research applies density ratio estimation for correcting deep generative diffusion models (Kim et al., 2023). In (Kim et al., 2024), the Bregman divergence approach in Eq. 1 is extended and further interesting Bregman divergences are identified.

## 3 NOTATION AND PROBLEM

Let $P, Q \in \mathcal{M}_+^1(\mathcal{X})$ be two probability measures from the set $\mathcal{M}_+^1(\mathcal{X})$ of probability measures on a compact space $\mathcal{X} \subset \mathbb{R}^d$, $d \in \mathbb{N}$ such that $P$ is absolutely continuous with respect to $Q$, $P \ll Q$. Sugiyama et al. (2012b) observed that many methods for estimating the density ratio $\beta := \frac{\mathrm{d}P}{\mathrm{d}Q}$ are minimizations of estimated Bregman divergences Eq. 1. Menon & Ong (2016) further proved that there is in fact a general mathematical structure underlying this observation, which we review in the following.

We start with the data generation model of Reid & Williamson (2010); Menon & Ong (2016): a probability measure $\rho$ on $\mathcal{X} \times \mathcal{Y}$ for $\mathcal{Y} := \{-1, 1\}$ with conditional probability measures[3] defined by $\rho(x|y = 1) := P(x)$ and $\rho(x|y = -1) := Q(x)$ and a marginal (w.r.t. $y$) probability measure $\rho_\mathcal{Y}$ defined as Bernoulli measure with probability[4] $\frac{1}{2}$ for both events $y = 1$ ("$x$ belongs to $P$") and $y = -1$ ("$x$ belongs to $Q$"). Our analysis is based on the following underlying assumption.

**Assumption 1** (Zellinger et al. (2023))**.** *The data*

$$\mathbf{z} := (x_i, 1)_{i=1}^m \cup (x_i', -1)_{i=1}^n \in \mathcal{X} \times \mathcal{Y} \tag{3}$$

*is an i.i.d. sample of $\rho$.*

Assumption 1 allows us to introduce a binary classification problem for the accessible data $\mathbf{z}$ as follows. For a fixed *loss* function $\ell : \{-1, 1\} \times \mathbb{R} \to \mathbb{R}$, the goal of binary classification is to find, based on the given the data $\mathbf{z}$, a classifier $f : \mathcal{X} \to \mathbb{R}$ with a small *expected risk*

$$\mathcal{R}(f) := \mathbb{E}_{(x,y)\sim\rho}[\ell(y, f(x))] = \int_{\mathcal{X} \times \mathcal{Y}} \ell(y, f(x)) \, \mathrm{d}\rho(x, y). \tag{4}$$

The loss function $\ell$ is a central object in our study. In particular, we consider *composite* loss functions $\ell : \{-1, 1\} \times \mathbb{R} \to \mathbb{R}$ such that $\ell(y, \widehat{y}) := \lambda(y, \Psi^{-1}(\widehat{y}))$ is composed of a loss function $\lambda : \{-1, 1\} \times [0, 1] \to \mathbb{R}$ and an invertible *link* function $\Psi : [0, 1] \to \mathbb{R}$. Composite loss functions allow to estimate $\rho(y = 1|x)$ by functions $f : \mathcal{X} \to \mathbb{R}$ mapping outside of $[0, 1]$ (Reid & Williamson, 2010). A central object in the study of composite loss functions is the *conditional risk* $\mathrm{L} : [0, 1] \times \mathbb{R} \to \mathbb{R}$ defined by

$$\mathrm{L}(\eta, \widehat{y}) := \eta \ell_1(\widehat{y}) + (1 - \eta)\ell_{-1}(\widehat{y}),$$

---

[3]Existence follows from $\mathcal{X} \times \mathcal{Y}$ being Polish (cf. (Dudley, 2002, Theorem 10.2.1), the product case).

[4]Our analysis can be generalized to probabilities $\pi \neq \frac{1}{2}$ using (Menon & Ong, 2016, Appendix A).

where $\ell_1(\widehat{y}) := \ell(1, \widehat{y})$ and $\ell_{-1}(\widehat{y}) := \ell(-1, \widehat{y})$ denote the *partial losses* of $\ell$. A composite loss function with invertible link $\Psi : [0, 1] \to \mathbb{R}$ is called *proper*, if the conditional risk is minimized at $\widehat{y} = \Psi(\eta)$:

$$\mathrm{L}(\eta, \Psi(\eta)) = \underline{\mathrm{L}}(\eta) := \inf_{\widehat{y} \in \mathbb{R}} \mathrm{L}(\eta, \widehat{y})$$

and it is *strictly proper*, if the *Bayes risk* $\underline{\mathrm{L}}(\eta)$ is achieved at a unique $\eta$. We denote by $f^* : \mathcal{X} \to \mathcal{Y}$ the function defined by $f^*(x) := \Psi \circ \rho(y = 1 | x)$. Then, the theoretical starting point of our work is the following key lemma for density ratio estimation.

**Lemma 1** (Menon & Ong (2016, Proposition 3)). *Let $\ell : \{-1, 1\} \times \mathbb{R} \to \mathbb{R}$ be a strictly proper composite loss function with invertible link function $\Psi : [0, 1] \to \mathbb{R}$ and twice differentiable negative Bayes risk $-\underline{\mathrm{L}} : [0, 1] \to \mathbb{R}$. Then*

$$\mathcal{R}(f) - \mathcal{R}(f^*) = \frac{1}{2} B_{-\underline{\mathrm{L}}^\diamond}(\beta, g \circ f) \tag{5}$$

*with*

$$g(y) := \frac{\Psi^{-1}(y)}{1 - \Psi^{-1}(y)} \quad and \quad -\underline{\mathrm{L}}^\diamond(z) := -(1 + z)\underline{\mathrm{L}}\left(\frac{z}{1 + z}\right). \tag{6}$$

For convenience of the reader, a proof of Lemma 1 is provided in Subsection A.3. Lemma 1 specifies, for a given loss function, the asymptotic error $B_{-\underline{\mathrm{L}}^\diamond}(\beta, \widehat{\beta})$ of the model $\widehat{\beta} := g \circ \widehat{f}$ computed by Algorithm 1. However, the error $B_{-\underline{\mathrm{L}}^\diamond}$ is defined by the loss function $\ell$, while one might be interested in specifying the error measure $B_\phi(\beta, g \circ \widehat{f})$, i.e., specifying $\phi$ and $g$. The problem of this work is to answer the following question:

*Which convex loss functions satisfy $\frac{1}{2} B_\phi(\beta, g \circ f) = \mathcal{R}(f) - \mathcal{R}(f^*)$ for given $\phi$ and $g$?*

Using the characterization of such loss functions, we are interested in designing loss functions that lead to error measures $B_\phi$ which put higher weight to larger density ratio values than to smaller values, and, to test their effect in importance weighted parameter selection of deep neural networks.

## 4 CHARACTERIZATION OF LOSSES FOR DENSITY RATIO ESTIMATION

The following Theorem 1, see Subsection B.2 for a proof, answers our research question in Section 3 above for general (not necessarily convex) strictly proper composite loss functions.

**Theorem 1.** *Let $\phi : [0, \infty) \to \mathbb{R}$ be strictly convex and twice differentiable and let $g : \mathbb{R} \to \mathbb{R}$ be strictly monotonically increasing. Then, $\ell : \{-1, 1\} \times \mathbb{R} \to \mathbb{R}$ is strictly proper composite and satisfies*

$$\mathcal{R}(f) - \mathcal{R}(f^*) = \frac{1}{2} B_\phi(\beta, g \circ f), \quad \forall P, Q \in \mathcal{M}_1^+(\mathcal{X}) : P \ll Q, \forall f \in L^1(Q), \tag{7}$$

*if and only if*

$$\ell(y, \widehat{y}) = \begin{cases} \gamma\big(\Psi^{-1}(\widehat{y})\big) + \big(1 - \Psi^{-1}(\widehat{y})\big) \gamma'\big(\Psi^{-1}(\widehat{y})\big) & \text{if } y = 1 \\ \gamma\big(\Psi^{-1}(\widehat{y})\big) - \Psi^{-1}(\widehat{y})\gamma'\big(\Psi^{-1}(\widehat{y})\big) & \text{if } y = -1 \end{cases} \tag{8}$$

*with $\Psi^{-1}(\widehat{y}) := \frac{g(\widehat{y})}{1 + g(\widehat{y})}$ and $\gamma(\widehat{\eta}) := -\phi\left(\frac{\widehat{\eta}}{1 - \widehat{\eta}}\right)(1 - \widehat{\eta}) + \widehat{\eta}c_2 + c_1$ for some $c_1, c_2 \in \mathbb{R}$.*

**Remark 1** (on Novelty). *The novelty of Theorem 1 essentially lies in the fact that any strictly proper composite loss function satisfying Eq. 7 is necessarily of the form of Eq. 8. Our proof has four main components: (a) the inversion of the constructions developed in (Menon & Ong, 2016, Appendix B) done in Lemma 5, (b) the application of (Reid & Williamson, 2010, Corollary 13), (c) Savage's theorem (Savage, 1971) and (d) our technical Lemma 4.*

**Remark 2.** *The sufficiency of Eq. 8 can be easily seen from Menon & Ong (2016); Reid & Williamson (2010); Bao (2023) as follows: In Menon & Ong (2016, Proposition 1) it is proven that any strictly proper composite loss satisfies Eq. 7 and Reid & Williamson (2010, Theorem 7) (cf. Reid & Williamson (2011); Bao (2023)) proves that the loss in Eq. 8 constructed by any strictly concave $\gamma$ is strictly proper composite.*

Theorem 1 expresses the expected Bregman divergence as an excess risk. This allows us to estimate the Bregman divergence in a natural way by empirical risk minimization. However, the minimization of non-convex functions is hard. We therefore characterize convex loss functions by a corollary of Theorem 1 combined with Theorem 4 originally provided by Reid & Williamson (2010).

**Corollary 1.** *Let $\phi : [0, \infty) \to \mathbb{R}$ be three times differentiable such that $\phi''(x) > 0$ for all $x \in [0, \infty)$ and let $g : \mathbb{R} \to \mathbb{R}$ be strictly monotonically increasing. Then, the loss $\ell$ in Eq. 8 is convex in its second argument for all $y \in \{-1, 1\}$ if and only if*

$$-\frac{1}{x} \leq \frac{\phi'''(x)}{\phi''(x)} - \frac{\left(g^{-1}\right)''(x)}{\left(g^{-1}\right)'(x)} \leq 0, \quad \forall x \in [0, \infty). \tag{9}$$

Corollary 1 is proven in Subsection B.3. Corollary 1 suggests a *canonical*

$$g_{\text{can}}^{-1}(\widehat{y}) := \phi'(\widehat{y}) \tag{10}$$

for constructing strictly convex loss functions by Eq. 8.

**Remark 3.** *The canonical link $\Psi'_{\text{can}}(\widehat{\eta}) := w(\widehat{\eta}) = \widehat{\eta}\,\ell'_1(\Psi(\widehat{\eta}))$ of Buja et al. (2005) (i.e., the matching loss of Kivinen & Warmuth (1997); Helmbold et al. (1999) as discussed in (Reid & Williamson, 2010, Section 6.1)) and the canonical $g_{\text{can}}$ do not lead to the same loss function. To see this, note that the equation $\Psi_{\text{can}}^{-1}(\widehat{y}) = \frac{g(\widehat{y})}{1+g(\widehat{y})}$ in Theorem 1 leads to*

$$\left(g^{-1}\right)'(z) = \Psi'_{\text{can}}\left(\frac{z}{1+z}\right) = w\left(\frac{z}{1+z}\right).$$

*Buja et al. (2005) show that differentiating a second time Eq. 20 gives $w(\widehat{\eta}) = -\underline{L}''(\widehat{\eta})$ and therefore*

$$\left(g^{-1}\right)'(z) = -\underline{L}''\left(\frac{z}{1+z}\right) = -\gamma''\left(\frac{z}{1+z}\right) = \phi''(z)(1+z)^3$$

*which satisfies Eq. 9 but differs from $\left(g_{\text{can}}^{-1}\right)' = \phi''(z)$ by a factor $(1+z)^3$.*

Thanks to Theorem 1 and Corollary 1, given a strictly convex Bregman generator $\phi$, we are now able to express the Bregman divergence $B_\phi(\beta, g_{\text{can}} \circ \widehat{f})$ as excess risk of a *canonical* convex loss function $\ell_{\text{can}}$ defined by Eq. 8 and $g_{\text{can}}$. Instantiating Algorithm 1 by $\ell_{\text{can}}$ and $\Psi^{-1}(\widehat{y}) = \frac{g_{\text{can}}(\widehat{y})}{1+g_{\text{can}}(\widehat{y})}$, we arrive at minimizing $B_\phi(\beta, \widehat{\beta})$ for $\widehat{\beta} = g_{\text{can}} \circ \widehat{f}$.

# 5 LOSS FUNCTIONS FOR ESTIMATING LARGE DENSITY RATIOS

It has been observed by Menon & Ong (2016) that no loss function in Example 1 prioritizes an accurate estimation of large density ratio values over smaller values. We show this behavior in Subsection A.1 which uses the fact that the second derivative $\phi''$ of the Bregman generator $\phi$ satisfies the relation

$$B_\phi(\beta, \widehat{\beta}) = \mathbb{E}_{x \sim Q}\left[\int_0^\infty \phi''(c)\phi_c(\beta(x), \widehat{\beta}(x))\, \mathrm{d}c\right] \tag{11}$$

for the piecewise linear function $\phi_c$ defined in Eq. 16 that does not depend on $\phi$. It can be seen that the weight functions $\phi''(c) = \frac{1}{c(1+c)}$ (LR), $\phi''(c) = c^{-1}$ (KLest), $\phi''(c) = c^{-3/2}$ (Exp) and $\phi''(c) = 1$ (KuLSIF) of Example 1 are non-increasing and therefore do not realize a prioritization of estimating large values. In contrast, we propose the following novel Bregman divergences as error measures for Algorithm 1, which prioritize an accurate estimation of large values, see Figure 1 for an illustration.

The exponential weight (EW) $\phi''(c) := e^{2c}$ and $g_{\text{can}}^{-1} := \phi'$ realize the novel Bregman divergence

$$B_\phi(\beta, \widehat{\beta}) = \frac{1}{4}\mathbb{E}_{x \sim Q}\left[e^{2\beta(x)} + e^{2\widehat{\beta}(x)}(-1 - 2\beta(x) + 2\widehat{\beta}(x))\right]. \tag{12}$$

Plugging $\phi$ and $g_{\text{can}}$ in Eq. 8 gives the loss functions $\ell(1, y) = -y$, $\ell(-1, y) = \frac{1}{2}y(\log(2y) - 1)$ and $\widehat{\beta}(x) = \frac{1}{2}\log(2\widehat{f}(x))$, which satisfy Eq. 7 according to Theorem 1.

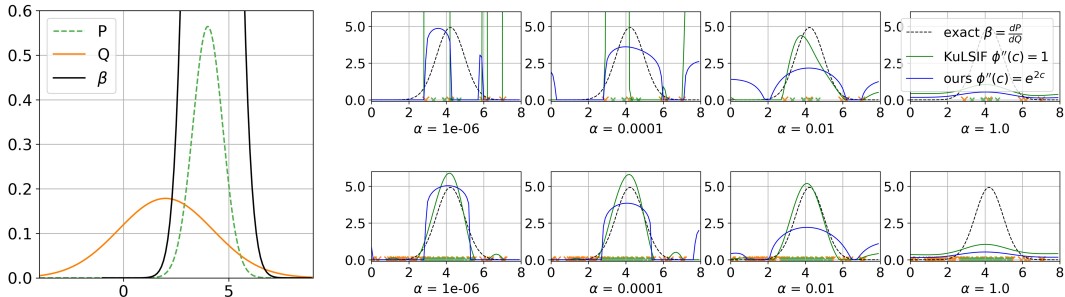

Figure 2: Estimation of density ratio (left, black) in Gaussian RKHS with Tikhonov penalty weighted by $\alpha$ for sample sizes $m + n = 10$ (top right) and $m + n = 100$ (lower right). Our loss function (blue) prioritizes accurate estimation of larger values and consequently produces flatter curves then KuLSIF (green).

The polynomial weight functions $\phi''(c) := c^{2k}$ for $k \in \{0, 1, 2, 3, \ldots\}$ and $g_{\mathrm{can}}^{-1} := \phi'$ in Eq. 10 realize the Bregman divergences

$$B_\phi(\beta, \widehat{\beta}) = \frac{1}{(1+k)(2+k)} \mathbb{E}_{x \sim Q} \left[ \beta(x)^{2+k} - (2+k)\beta(x)\widehat{\beta}(x)^{1+k} + (1+k)\widehat{\beta}(x)^{2+k} \right]. \quad (13)$$

Plugging $\phi$ and $g_{\mathrm{can}}$ in Eq. 8 gives the loss functions $\ell(1, y) = -y$ and $\ell(-1, y) = \frac{((1+k)y)^{\frac{2+k}{1+k}}}{2+k}$ and the estimators $\widehat{\beta}(x) = ((1+k)\widehat{f}(x))^{\frac{1}{1+k}}$. Note that the class of divergences in Eq. 13 can be interpreted to extend KuLSIF, which realizes Eq. 13 with $k = 0$, to larger exponents.

# 6 EMPIRICAL EVALUATIONS

## 6.1 NUMERICAL ILLUSTRATIONS

Before showing the potential of our novel loss functions on real world problems in Subsection 6.2, we illustrate the behavior of our methods on two experiments: Experiment (a) on the form of kernel estimators for a one-dimensional Gaussian density ratio and Experiment (b) on the behavior of the (analytically computed) density ratio estimators used as importance weights in kernel regression.

**Experiment (a):** Figure 2 shows the behavior of Algorithm 1 instantiated with the loss functions for KuLSIF (see Example 1) and our novel loss function corresponding to Eq. 12, for the numerical illustration of Zellinger et al. (2023). The function class $\mathcal{F}$ was fixed to a Gaussian RKHS with bandwidth parameter computed by the median heuristic. We also added a Tikhonov penalty $+\alpha\|f\|_{\mathcal{H}}^2$ to Eq. 2 and computed the solution for different values of $\alpha \in \{10^{-6}, 10^{-4}, 10^{-2}, 1\}$ by the Broyden-Fletcher-Goldfarb-Shanno (BFGS) algorithm, see, e.g., (Nocedal & Wright, 1999).

The equal weighting of all density ratio values in KuLSIF leads to exploding estimates for small sample size $m + n = 10$ with small regularization parameter $\alpha \in \{10^{-6}, 10^{-4}\}$. This problem does not appear for our loss function, which comes, however, at the cost of "flatter" estimates due to a higher penalization of errors in larger density ratio values.

**Experiment (b):** Figure 3 extends the setting of Figure 1 to unsupervised domain adaptation under covariate shift (Shimodaira, 2000): given iid *source* observations $(x_i')_{i=1}^m$ from $Q$ together with labels $y_i' = f_P(x_i') + \epsilon_i$ corrupted by iid Gaussian noise observations $(\epsilon_i)_{i=1}^n$ and unlabeled iid *target* observations $(x_i)_{i=1}^n$ from $P$, the goal is to estimate the unknown Bayes predictor $f_P(x) = \sin(3x^4)$. This is done by importance weighted kernel least squares regression

$$\widehat{f} := \arg\min_{f \in \mathcal{F}} \frac{1}{m+n} \sum_{i=1}^m \widehat{\beta}(x_i') \|y_i' - f(x_i')\|_2^2 + \alpha\|f\|_{\mathcal{F}}^2 \quad (14)$$

with $\widehat{\beta}$ of Figure 1 for LR and our method with exponential weight $\phi''(c) = e^{2c}$. We used a degree-five polynomial RKHS $\mathcal{F}$, a small $\alpha = 10^{-32}$ and computed the solution by BFGS optimizing the weights of the solution granted by the representer theorem Kimeldorf & Wahba (1970).

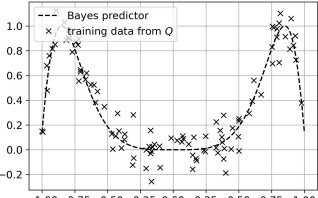 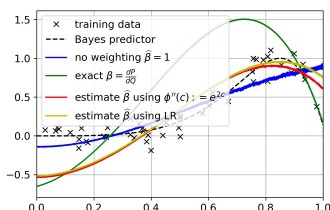 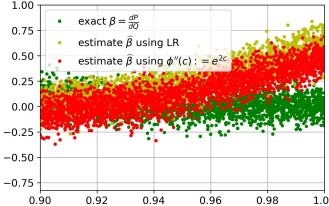

Figure 3: Using density ratio estimators from Figure 1 as sample weights for polynomial kernel least squares regression. Left: Bayes predictor (black, dashed) and observations ($\times$) from $Q$. Middle: regressors (blue: uniform weighting $\widehat{\beta} \equiv 1$, green: exact $\beta$, red: our estimate with $\phi''(c) = e^{2c}$ from Figure 1, yellow: LR estimate from Figure 1). Right: Pointwise errors for target observations from $P$. Exponential weight (ours) is higher for larger density ratio values; consequently achieves smaller errors on $[0.9, 1]$.

Our weight $\phi''(c) = e^{2c}$ is increasing, which leads to an error smaller than $0.040$ in squared $L^2(P)$-norm. This can be explained by the fact that the $L^2(P)$-norm assigns higher weight to regions where the density ratio is large and that these regions are exactly the ones where our weight leads to more accurate estimates. In contrast, the weight function of LR is decreasing, which leads to an error larger than $0.067$ in squared $L^2(P)$-norm. The higher performance of our method in squared $L^2(P)$-norm comes at the cost of a higher squared $L^2(Q)$-norm ($0.798$) than for LR ($0.778$). However, we note that in unsupervised domain adaptation we are interested in the target excess risk on $P$ which is nothing but the squared $L^2(P)$-norm. The corresponding pointwise errors are depicted in Figure 3.

## 6.2 REAL-WORLD DATASETS

In the following, we test the performance of our exponential weighted Bregman divergence in Eq. 12 compared to the performace of LR, KuLSIF and Boost (see Example 1) for calculating importance weights for parameter selection methods in unsupervised domain adaptation. In total, we test the performance on 484 parameter selection tasks; one for each combination of a parameter selection method, a domain adaptation method (from three datasets) and a domain adaptation tasks (originating from three datasets). We trained 9174 neural networks in total, see Section C for details.

**The unsupervised domain adaptation setting:** given iid *source* observations $(x_i')_{i=1}^m$ from $Q$ together with labels $y_i' = f_P(x_i') + \epsilon_i$ corrupted by *general* iid noise observations $(\epsilon_i)_{i=1}^n$ and unlabeled iid *target* observations $(x_i)_{i=1}^n$ from $P$, the goal is to estimate the *generally unknown* Bayes predictor $f_P : \mathcal{X} \to \mathcal{R}^{d_y}, d_y \in \mathbb{N}$.

**Domain adaptation methods:** The goal is to select parameters from the methods Adversarial Spectral Kernel Matching (AdvSKM) (Liu & Xue, 2021), Deep Domain Confusion (DDC) (Tzeng et al., 2014), Correlation Alignment (DCoral) (Sun et al., 2017; Nikzad-Langerodi et al., 2019), Central Moment Discrepancy (CMD) (Zellinger et al., 2017; 2019), Higher-order Moment Matching (HoMM) (Chen et al., 2020), Minimum Discrepancy Estimation for Deep Domain Adaptation (MMDA) (Rahman et al., 2020), Deep Subdomain Adaptation (DSAN) (Zhu et al., 2021), Domain-Adversarial Neural Networks (DANN) (Ganin et al., 2016), Conditional Adversarial Domain Adaptation (CDAN) (Long et al., 2018), DIRT-T (DIRT) (Shu et al., 2018) and Convolutional Deep Domain Adaptation specifically suitable for time-series (CoDATS) (Wilson et al., 2020).

**Parameter selection methods:** We select these parameters by Importance Weighted (Cross-)Validation (IWV) (Sugiyama et al., 2007) and Importance Weighted Aggregation (IWA) (Dinu et al., 2023). Both methods, IWV and IWA, first compute several models $\hat{f}_1, \ldots, \hat{f}_l : \mathcal{X} \to \mathbb{R}^{d_y}$, one for each parameter setting to try, and second compute the final model $\hat{f} : \mathcal{X} \to \mathbb{R}^{d_y}$ by Eq. 14 for $\mathcal{F} = \{f_1, \ldots, f_l\}$ and $\mathcal{F} = \{\sum_{k=1}^l c_k \cdot f_k | c_1, \ldots, c_k \in \mathbb{R}\}$ for IWV and IWA, respectively. Since Eq. 14 relies on an accurate estimate of $\beta = \frac{dP}{dQ}$ also IWV and IWA crucially depend on such an estimate.

**Datasets and tasks:** We rely on three domain adaptation datasets with several domains and tasks. The first dataset is the *AmazonReviews* dataset from Blitzer et al. (2006) which consists of text re-

views from four domains: books (B), DVDs (D), electronics (E), and kitchen appliances (K). The reviews are encoded in feature vectors of bag-of-words unigrams and bigrams with binary labels indicating the sentiment of the rankings. From the four domains we obtain twelve domain adaptation tasks (B→D, D→E,...) where each domain serves once as source domain and once as target domain. The second dataset is the *Heterogeneity Human Activity Recognition dataset* (HHAR) from Stisen et al. (2015) which investigates sensor-, device- and workload-specific heterogeneities from 36 smartphones and smartwatches, consisting of 13 different device models from four manufacturers. We follow Ragab et al. (2023) and obtain five domain adaptation tasks. Our third dataset is the *MiniDomainNet* dataset from Zellinger et al. (2021) which is a reduced version of the large scale *DomainNet-2019* from Peng et al. (2019) consisting of six different image domains (Quickdraw, Real, Clipart, Sketch, Infograph, and Painting). MiniDomainNet reduces the number of classes of DomainNet-2019 to the top-five largest representatives in the training set of each class across all six domains. MiniDomainNet is a multi-source single target dataset for which we obtain five tasks, one task for each domain chosen as target domain (and all others used as source domains).

**Density ratio estimation methods:** We estimate the density ratio by Algorithm 1 initialized by our loss function corresponding to Eq. 12 and the other loss functions from Example 1 excluding SQ which did not perform well in our experiments and also not in the one of Gruber et al. (2024). Eq. 2 is optimized with the BFGS algorithm (100 iterations) in a Gaussian RKHS with bandwidth chosen by the median heuristic and Tikhonov regularization with parameter $\alpha \in \{10, 0.1, 10^{-3}\}$ chosen by cross-validation following Gruber et al. (2024).

**Framework and benchmark protocol:** Our framework extends the Pytorch framework[5] of Gruber et al. (2024), by our exponential weight method in Eq. 12. The Pytorch framework of Gruber et al. (2024) itself extends the one[6] of Dinu et al. (2023) which relies on the AdaTime benchmark suite[7] of Ragab et al. (2023). In all our remaining parameters, we exactly follow the statistical setup and data splits of Dinu et al. (2023) together with the cross-validation based parameter choice for density ratio of Gruber et al. (2024), see Section C for details.

## 6.3 RESULTS OF EMPIRICAL EVALUATIONS

Beyond the obvious advantages of our novel methods illustrated in Figure 1, Experiment (a) and Experiment (b), we obtain the following results (see Subsections C.3, C.2 and C.3 for the full results):

- EW outperforms on average LR, KuLSIF and Boost for IWV and IWA on all three datasets, excluding IWV on MiniDomainNet for which all methods (Boost, KuLSIF, LR and EW) perform similarly.

- EW is among the best methods in 56% of parameter selection problems, i.e., among all cells in all tables in Subsections C.3, C.2 and C.3; clearly outperforming KuLSIF (23%), LR (13%) and Boost (17%).

## 7 CONCLUSION AND FUTURE WORK

A large class of methods for density ratio estimation minimizes a Bregman divergence between the true density ratio and an estimator. However, the Bregman divergence is only implicitly determined and depends on the choice of a binary loss function used to instantiate the algorithm that underlies this class of methods. In this work, we derived the *unique* formula of a (strictly proper composite convex) loss function for an explicitly given Bregman divergence that we want to minimize. Our formula allows us to construct novel loss functions with the property of focusing on estimating large density ratio values, which is in contrast to related approaches. We provide numerical illustrations and applications in deep domain adaptation. The sample complexity of our methods is, to the best of our knowledge, an open problem.

---

[5]https://github.com/lugruber/dre_iter_reg
[6]https://github.com/Xpitfire/iwa
[7]https://github.com/emadeldeen24/AdaTime

**AmazonReviews**

|  | SO | Importance Weighted Validation | | | | Importance Weighted Aggregation | | | | TB |
|---|---|---|---|---|---|---|---|---|---|---|
|  |  | Boost | KuLSIF | LR | EW (ours) | Boost | KuLSIF | LR | EW (ours) |  |
| AdvSKM | 0.766(±.009) | 0.766(±.010) | 0.767(±.010) | 0.767(±.013) | **0.770(±.010)** | 0.766(±.013) | 0.773(±.009) | 0.769(±.012) | **0.778(±.010)** | 0.770(±.010) |
| CDAN | 0.767(±.012) | **0.774(±.012)** | 0.773(±.011) | 0.772(±.011) | 0.773(±.009) | 0.768(±.025) | 0.782(±.014) | 0.775(±.019) | **0.789(±.011)** | 0.777(±.012) |
| CMD | 0.767(±.009) | 0.773(±.017) | 0.774(±.013) | 0.774(±.019) | **0.779(±.011)** | 0.745(±.046) | 0.782(±.013) | 0.772(±.021) | **0.791(±.010)** | 0.785(±.009) |
| CoDATS | 0.766(±.012) | **0.785(±.018)** | 0.783(±.017) | 0.781(±.020) | 0.784(±.017) | 0.777(±.020) | 0.790(±.012) | 0.784(±.018) | **0.797(±.010)** | 0.791(±.013) |
| DANN | 0.767(±.012) | 0.780(±.015) | 0.781(±.017) | 0.777(±.013) | **0.785(±.011)** | 0.772(±.024) | 0.789(±.011) | 0.779(±.020) | **0.799(±.008)** | 0.798(±.009) |
| DDC | 0.766(±.011) | 0.769(±.012) | **0.770(±.010)** | 0.770(±.011) | 0.769(±.012) | 0.765(±.022) | 0.772(±.014) | 0.769(±.015) | **0.779(±.011)** | 0.770(±.011) |
| DIRT | 0.764(±.013) | 0.775(±.020) | 0.780(±.016) | 0.777(±.017) | **0.783(±.017)** | 0.767(±.022) | 0.786(±.010) | 0.774(±.016) | **0.794(±.009)** | 0.786(±.015) |
| DSAN | 0.769(±.012) | 0.778(±.014) | 0.778(±.013) | 0.777(±.016) | **0.783(±.013)** | 0.769(±.037) | 0.788(±.014) | 0.782(±.017) | **0.798(±.008)** | 0.789(±.013) |
| DCoral | 0.766(±.012) | 0.769(±.012) | 0.770(±.012) | 0.769(±.011) | **0.771(±.012)** | 0.768(±.020) | 0.778(±.012) | 0.773(±.014) | **0.783(±.010)** | 0.776(±.011) |
| HoMM | 0.769(±.012) | 0.765(±.010) | 0.766(±.012) | 0.766(±.010) | **0.767(±.011)** | 0.762(±.021) | 0.771(±.013) | 0.765(±.015) | **0.776(±.009)** | 0.769(±.012) |
| MMDA | 0.767(±.009) | 0.768(±.014) | 0.772(±.014) | 0.769(±.012) | **0.775(±.012)** | 0.769(±.018) | 0.778(±.012) | 0.772(±.018) | **0.789(±.008)** | 0.782(±.012) |
| mean | 0.767(±.011) | 0.773(±.014) | 0.774(±.013) | 0.773(±.014) | **0.776(±.012)** | 0.766(±.024) | 0.781(±.012) | 0.774(±.017) | **0.789(±.010)** | 0.781(±.012) |

**HHAR**

|  | SO | Importance Weighted Validation | | | | Importance Weighted Aggregation | | | | TB |
|---|---|---|---|---|---|---|---|---|---|---|
|  |  | Boost | KuLSIF | LR | EW (ours) | Boost | KuLSIF | LR | EW (ours) |  |
| AdvSKM | 0.718(±.044) | **0.729(±.028)** | 0.726(±.025) | **0.729(±.028)** | **0.729(±.028)** | 0.682(±.042) | 0.683(±.053) | **0.684(±.034)** | 0.679(±.039) | 0.749(±.023) |
| CDAN | 0.728(±.042) | **0.789(±.035)** | 0.765(±.082) | 0.785(±.033) | 0.781(±.031) | 0.730(±.051) | 0.747(±.051) | 0.739(±.047) | **0.755(±.053)** | 0.790(±.025) |
| CMD | 0.748(±.026) | 0.760(±.040) | 0.757(±.043) | 0.757(±.043) | **0.768(±.027)** | 0.714(±.038) | **0.728(±.041)** | 0.722(±.040) | 0.723(±.033) | 0.794(±.015) |
| CoDATS | 0.710(±.063) | **0.741(±.052)** | 0.720(±.076) | **0.741(±.052)** | 0.739(±.048) | 0.724(±.042) | 0.731(±.039) | 0.721(±.028) | **0.732(±.038)** | 0.785(±.027) |
| DANN | 0.757(±.027) | 0.796(±.030) | 0.793(±.081) | **0.798(±.027)** | **0.798(±.027)** | 0.752(±.044) | 0.759(±.057) | 0.765(±.037) | **0.767(±.040)** | 0.807(±.023) |
| DDC | 0.716(±.036) | **0.713(±.052)** | 0.701(±.046) | 0.705(±.039) | 0.712(±.036) | **0.713(±.042)** | 0.702(±.035) | 0.703(±.043) | 0.703(±.040) | 0.729(±.035) |
| DIRT | 0.728(±.030) | 0.748(±.035) | **0.814(±.060)** | 0.748(±.035) | 0.749(±.035) | 0.718(±.082) | 0.730(±.066) | 0.725(±.073) | **0.747(±.047)** | 0.820(±.039) |
| DSAN | 0.721(±.046) | **0.727(±.066)** | 0.717(±.068) | 0.725(±.062) | 0.723(±.062) | 0.731(±.030) | **0.736(±.037)** | 0.728(±.034) | 0.731(±.033) | 0.826(±.023) |
| DCoral | 0.745(±.039) | 0.747(±.040) | 0.744(±.038) | **0.748(±.038)** | **0.748(±.038)** | 0.673(±.054) | 0.687(±.035) | 0.682(±.032) | **0.687(±.031)** | 0.776(±.038) |
| HoMM | 0.739(±.057) | 0.720(±.041) | **0.730(±.061)** | 0.728(±.037) | 0.728(±.037) | 0.671(±.061) | 0.690(±.044) | 0.681(±.053) | **0.703(±.037)** | 0.764(±.020) |
| MMDA | 0.738(±.053) | **0.726(±.040)** | 0.723(±.031) | 0.726(±.044) | 0.722(±.044) | 0.685(±.060) | 0.697(±.039) | 0.690(±.048) | **0.710(±.029)** | 0.785(±.046) |
| mean | 0.732(±.042) | 0.745(±.042) | 0.744(±.056) | 0.745(±.040) | **0.745(±.038)** | 0.709(±.050) | 0.717(±.045) | 0.713(±.043) | **0.722(±.038)** | 0.784(±.028) |

**MiniDomainNet**

|  | SO | Importance Weighted Validation | | | | Importance Weighted Aggregation | | | | TB |
|---|---|---|---|---|---|---|---|---|---|---|
|  |  | Boost | KuLSIF | LR | EW (ours) | Boost | KuLSIF | LR | EW (ours) |  |
| AdvSKM | 0.509(±.018) | 0.515(±.014) | 0.514(±.014) | 0.515(±.014) | **0.516(±.015)** | 0.512(±.015) | **0.513(±.013)** | 0.511(±.015) | 0.512(±.012) | 0.522(±.020) |
| CDAN | 0.514(±.015) | 0.511(±.013) | 0.513(±.019) | 0.510(±.021) | **0.514(±.018)** | 0.519(±.031) | 0.530(±.014) | **0.531(±.019)** | 0.530(±.021) | 0.542(±.017) |
| CMD | 0.509(±.022) | **0.525(±.022)** | 0.522(±.023) | 0.522(±.021) | 0.520(±.021) | 0.517(±.025) | 0.516(±.025) | **0.519(±.029)** | 0.518(±.031) | 0.533(±.020) |
| CoDATS | 0.502(±.032) | 0.514(±.023) | **0.515(±.020)** | 0.512(±.023) | **0.515(±.020)** | 0.527(±.023) | 0.526(±.015) | 0.525(±.017) | **0.527(±.016)** | 0.529(±.019) |
| DANN | 0.496(±.019) | 0.511(±.020) | 0.514(±.018) | 0.512(±.021) | **0.515(±.018)** | 0.515(±.013) | 0.516(±.014) | **0.516(±.010)** | 0.513(±.014) | 0.532(±.024) |
| DDC | 0.510(±.021) | 0.513(±.010) | **0.514(±.010)** | 0.513(±.010) | 0.512(±.012) | 0.516(±.015) | 0.515(±.018) | 0.515(±.016) | **0.518(±.018)** | 0.521(±.029) |
| DIRT | 0.499(±.026) | 0.491(±.033) | 0.495(±.032) | **0.500(±.039)** | 0.484(±.024) | 0.519(±.031) | **0.523(±.025)** | 0.519(±.020) | 0.519(±.021) | 0.525(±.042) |
| DSAN | 0.509(±.022) | 0.504(±.020) | 0.510(±.028) | 0.502(±.020) | **0.516(±.013)** | 0.518(±.018) | 0.520(±.014) | 0.522(±.015) | **0.524(±.018)** | 0.563(±.024) |
| DCoral | 0.505(±.028) | **0.522(±.012)** | 0.522(±.014) | 0.521(±.013) | 0.522(±.015) | 0.520(±.012) | 0.520(±.013) | **0.521(±.013)** | **0.521(±.013)** | 0.535(±.017) |
| HoMM | 0.509(±.023) | 0.518(±.021) | **0.521(±.025)** | 0.516(±.020) | 0.514(±.019) | 0.521(±.019) | 0.526(±.013) | 0.523(±.016) | **0.527(±.014)** | 0.537(±.014) |
| MMDA | 0.509(±.022) | **0.522(±.020)** | 0.518(±.018) | 0.518(±.018) | 0.519(±.020) | **0.530(±.012)** | 0.529(±.013) | 0.530(±.016) | 0.528(±.013) | 0.531(±.013) |
| mean | 0.507(±.022) | 0.513(±.019) | **0.514(±.020)** | 0.513(±.020) | 0.513(±.018) | 0.519(±.019) | 0.521(±.016) | 0.521(±.017) | **0.522(±.017)** | 0.534(±.022) |

Table 1: Mean (± standard deviation) for three random initializations of model weights averaged over 12 domain adaptation tasks for AmazonReviews and five tasks for HHAR and MiniDomainNet; full tables in Subsections C.3, C.2 and C.3.

## ACKNOWLEDGMENTS

Part of this work was done while Werner Zellinger was affiliated with the Johann Radon Institute for Computational and Applied Mathematics of the Austrian Academy of Sciences.

I thank five anonymous reviewers and an anonymous area chair for helpful discussions. I also thank Lukas Gruber and Marius-Constantin Dinu for helpful discussions on their Pytorch frameworks.

The ELLIS Unit Linz, the LIT AI Lab, the Institute for Machine Learning, are supported by the Federal State Upper Austria. We thank the projects Medical Cognitive Computing Center (MC3), INCONTROL-RL (FFG-881064), PRIMAL (FFG-873979), S3AI (FFG-872172), DL for Gran- ularFlow (FFG-871302), EPILEPSIA (FFG-892171), AIRI FG 9-N (FWF-36284, FWF-36235), AI4GreenHeatingGrids (FFG- 899943), INTEGRATE (FFG-892418), ELISE (H2020-ICT-2019-3 ID: 951847), Stars4Waters (HORIZON-CL6-2021-CLIMATE-01-01). We thank Audi.JKU Deep Learning Center, TGW LOGISTICS GROUP GMBH, Silicon Austria Labs (SAL), FILL Gesellschaft mbH, Anyline GmbH, Google, ZF Friedrichshafen AG, Robert Bosch GmbH, UCB Biopharma SRL, Merck Healthcare KGaA, Verbund AG, Software Competence Center Hagenberg GmbH, Borealis AG, TÜV Austria, Frauscher Sensonic, TRUMPF and the NVIDIA Corporation.

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

# Appendix

## Table of Contents

## A AUXILIARIES

### A.1 WEIGHT REPRESENTATION OF BREGMAN DIVERGENCES

The second derivative $\phi''$ of the Bregman generator $\phi$ of Eq. 1 can be seen as a weight of the Bregman divergence as follows.

**Lemma 2** (Reid & Williamson (2009)). *Let $\phi : \mathbb{R} \to \mathbb{R}$ be strictly convex and twice differentiable. Then*

$$B_\phi(\beta, \widehat{\beta}) = \mathbb{E}_{x \sim Q} \left[ \int_0^\infty \phi''(c) \phi_c(\beta(x), \widehat{\beta}(x)) \, dc \right] \tag{15}$$

*with*

$$\phi_c(r, \widehat{r}) := \begin{cases} (r - c) & \text{if } \widehat{r} < c \leq r \\ (c - r) & \text{if } r < c \leq \widehat{r} \\ 0 & \text{otherwise} \end{cases}. \tag{16}$$

For the convenience of the reader, we provide a proof of Lemma 2 based on Taylor's theorem.

*Proof of Lemma 2.* For any $s_0, s \in [a, b] \subset \mathbb{R}$ and $\phi : [a, b] \to \mathbb{R}$, Taylor's theorem grants us the expansion (cf. Reid & Williamson (2009, Corollary 7))

$$\phi(s) = \phi(s_0) + \phi'(s_0)(s - s_0) + \int_a^b \phi_c(s, s_0) \phi''(c) \, dc. \tag{17}$$

It follows that

$$\begin{aligned} B_\phi(\beta, \widehat{\beta}) &= \mathbb{E}_{x \sim Q} \left[ \phi(\beta(x)) - \phi(\widehat{\beta}(x)) - \phi'(\widehat{\beta}(x))(\beta(x) - \widehat{\beta}(x)) \right] \\ &= \mathbb{E}_{x \sim Q} \left[ \int_{\widehat{\beta}(x) \wedge \beta(x)}^{\widehat{\beta}(x) \vee \beta(x)} \phi_c(\beta(x), \widehat{\beta}(x)) \phi''(c) \, dc \right] \\ &= \mathbb{E}_{x \sim Q} \left[ \int_0^\infty \phi_c(\beta(x), \widehat{\beta}(x)) \phi''(c) \, dc \right]. \end{aligned}$$

$\square$

Lemma 2 illustrates a prioritization of large values $r_2 \geq r_1$ over smaller ones for increasing weight function $\phi''$:

$$d_\phi(r_2, r_2 - \Delta) \geq d_\phi(r_1, r_1 - \Delta) \tag{18}$$

for the distance

$$d_\phi(s, s_0) := \phi(s) - \phi(s_0) - \phi'(s_0)(s - s_0)$$

that determines $B_\phi(\beta, \widehat{\beta}) = \mathbb{E}_{x \sim Q}[d_\phi(\beta(x), \widehat{\beta}(x))]$. Eq. 18 follows from Lemma 2 and the substitution $\varphi(x) := x + r_1 - r_2$,

$$\begin{aligned}
d_\phi(r_1, r_1 - \Delta) &= \int_0^\infty \phi_c(r_1, r_1 - \Delta)\phi''(c)\,\mathrm{d}c \\
&= \int_{r_1-\Delta}^{r_1} (r_1 - c)\phi''(c)\,\mathrm{d}c \\
&= \int_{\varphi(r_2-\Delta)}^{\varphi(r_2)} (r_1 - c)\phi''(c)\,\mathrm{d}c \\
&= \int_{r_2-\Delta}^{r_2} \phi''(\varphi(c))(r_1 - \varphi(c))\varphi'(c)\,\mathrm{d}c \\
&= \int_{r_2-\Delta}^{r_2} \phi''(c + r_1 - r_2)(r_2 - c)\,\mathrm{d}c \\
&\leq \int_{r_2-\Delta}^{r_2} \phi''(c)(r_2 - c)\,\mathrm{d}c \\
&= d_\phi(r_2, r_2 - \Delta),
\end{aligned}$$

where the inequality holds for increasing $\phi''(c - (r_2 - r_1)) \leq \phi''(c)$.

## A.2 CHARACTERIZATION OF PROPER LOSSES

For convenience of the reader, in the following, we provide self-inclusive proofs of all mathematical statements used to prove our results.

Menon & Ong (2016) discovered a connection[8] between $\phi''$ in Lemma 2 and the weight in a fundamental characterization of proper loss functions due to Shuford Jr et al. (1966, Theorem 1) (cf. Aczél & Pfanzagl (1967); Staël von Holstein (1970)), which we re-state here following Buja et al. (2005); Reid & Williamson (2010).

**Theorem 2** (Shuford Jr et al. (1966, Theorem 1)). *A loss function* $\lambda : \{-1, 1\} \times [0, 1] \to \mathbb{R}$ *with differentiable partial losses* $\lambda_1, \lambda_{-1}$ *is strictly proper if and only if for all* $\eta \in (0, 1)$

$$\frac{\lambda_1'(\eta)}{\eta - 1} = \frac{\lambda_{-1}'(\eta)}{\eta} = w(\eta) \tag{19}$$

*for some weight function* $w : (0, 1) \to [0, \infty)$ *such that* $\int_\epsilon^{1-\epsilon} w(c)\,\mathrm{d}c < \infty$ *for all* $\epsilon > 0$.

*Proof of Theorem 2.* If $\lambda$ is proper, then $\mathrm{L}(\eta, \eta) = \inf_{\widehat{\eta} \in (0,1)} \mathrm{L}(\eta, \widehat{\eta})$, which implies that

$$0 = \left.\frac{\partial}{\partial \widehat{\eta}} \mathrm{L}(\eta, \widehat{\eta})\right|_{\widehat{\eta}=\eta} = (1 - \eta)\lambda_{-1}'(\eta) + \eta\lambda_1'(\eta)$$

and consequently also Eq. 19.

Eq. 19 gives

$$\begin{aligned}
\frac{\partial}{\partial \widehat{\eta}} \mathrm{L}(\eta, \widehat{\eta}) &= (1 - \eta)\lambda_{-1}'(\widehat{\eta}) + \eta\lambda_1'(\widehat{\eta}) \\
&= \eta(\widehat{\eta} - 1)w(\widehat{\eta}) + (1 - \eta)\widehat{\eta}w(\widehat{\eta}) \\
&= (\widehat{\eta} - \eta)w(\widehat{\eta}), \tag{20}
\end{aligned}$$

which provides us a unique minimum $\mathrm{L}(\eta, \eta) = \inf_{\widehat{\eta} \in (0,1)} \mathrm{L}(\eta, \widehat{\eta})$ since $w(\eta) > 0$. $\square$

---

[8]The relation $\phi''(c) = \frac{1}{(1+c)^3} w\left(\frac{c}{1+c}\right)$ is proven in (Menon & Ong, 2016, Lemma 4) with $w$ as defined in our Theorem 2.

Another fundamental characterization of proper loss functions is due to Savage (1971), which we re-state here in the notation of (Reid & Williamson, 2010, Theorem 4).

**Theorem 3** (Savage (1971)). *A loss function $\lambda : \{-1, 1\} \times [0, 1] \to \mathbb{R}$ is proper if and only if its pointwise Bayes risk $\underline{L}$ is concave and for each $\eta \in [0, 1]$ and $\widehat{\eta} \in (0, 1)$*

$$L(\eta, \widehat{\eta}) = \underline{L}(\widehat{\eta}) + (\eta - \widehat{\eta}) \underline{L}'(\widehat{\eta}). \tag{21}$$

The next proof of Theorem 3 is given by Reid & Williamson (2009).

*Proof of Theorem 3.* To prove the first direction, note that the properness of the loss $\lambda$ means that

$$L(\eta, \eta) = \underline{L}(\eta) := \inf_{\widehat{\eta} \in [0,1]} L(\eta, \widehat{\eta})$$

for all $\eta \in [0, 1]$. Expanding the derivative of the pointwise Bayes risk gives

$$
\begin{aligned}
\underline{L}'(\eta) &= \frac{\partial}{\partial \eta} L(\eta, \eta) \\
&= \frac{\partial}{\partial \eta} \left( (1 - \eta)\lambda(-1, \eta) + \eta\lambda(1, \eta) \right) \\
&= -\lambda(-1, \eta) + (1 - \eta)\lambda'(-1, \eta) + \lambda(1, \eta) + \eta\lambda'(1, \eta) \\
&= -\lambda(-1, \eta) + \lambda(1, \eta),
\end{aligned}
$$

where the last equality follows from the fact that

$$0 = \frac{\partial}{\partial \widehat{\eta}} L(\eta, \widehat{\eta}) \Big|_{\widehat{\eta}=\eta} = (1 - \eta)\lambda'(-1, \eta) + \eta\lambda'(1, \eta).$$

Eq. 21 can now be obtained as follows:

$$
\begin{aligned}
\underline{L}(\widehat{\eta}) + (\eta - \widehat{\eta}) \underline{L}'(\widehat{\eta}) &= (1 - \widehat{\eta})\lambda(-1, \widehat{\eta}) + \widehat{\eta}\lambda(1, \widehat{\eta}) + (\eta - \widehat{\eta})(-\lambda(-1, \widehat{\eta}) + \lambda(1, \widehat{\eta})) \\
&= (1 - \eta)\lambda(-1, \widehat{\eta}) + \eta\lambda(1, \widehat{\eta})
\end{aligned}
$$

where the last line is just the definition of $L(\eta, \widehat{\eta})$.

To prove the other direction, assume that $\phi : [0, 1] \to \mathbb{R}$ is a concave function and

$$L(\eta, \widehat{\eta}) = \phi(\widehat{\eta}) + (\eta - \widehat{\eta})\phi'(\widehat{\eta}).$$

Taylor's theorem grants us the expansion (cf. (Reid & Williamson, 2009, Corollary 7))

$$\phi(\eta) = \phi(\widehat{\eta}) + \phi'(\widehat{\eta})(\eta - \widehat{\eta}) + \int_0^1 \phi_c(\eta, \widehat{\eta})\phi''(c) \, \mathrm{d}c.$$

for $\phi_c$ as defined in Lemma 2. It follows that

$$L(\eta, \widehat{\eta}) = \phi(\eta) - \int_0^1 \phi_c(\eta, \widehat{\eta})\phi''(c) \, \mathrm{d}c \geq \phi(\widehat{\eta}) \tag{22}$$

since $\phi$ is concave and $\phi''(\eta) \leq 0$. The integral in Eq. 22 is minimized to zero in the case of $\widehat{\eta} = \eta$. $\qquad\square$

### A.3 BREGMAN DIVERGENCE REPRESENTATION OF EXCESS RISK

To prove Lemma 1, we will need the following Bregman identity.

**Lemma 3** (Menon & Ong (2016, Lemma 2)). *Let $\phi : [0, 1] \to \mathbb{R}$ be twice differentiable strictly convex and denote by $d_\phi(y, \widehat{y}) := \phi(y) - \phi(\widehat{y}) - \phi'(\widehat{y})(y - \widehat{y})$. Then, for all $x, y \in [0, \infty)$, it holds that*

$$(1 + x) \cdot d_\phi\left( \frac{x}{1 + x}, \frac{y}{1 + y} \right) = d_{\phi^\diamond}(x, y), \tag{23}$$

*where $\phi^\diamond(z) := (1 + z) \cdot \phi\left( \frac{z}{1+z} \right)$.*

*Proof of Lemma 3 (Menon & Ong, 2016, Section 5.1).* Eq. 17 gives, for any $s_0, s \in \mathbb{R}$:

$$d_\phi(s, s_0) = \int_0^\infty \phi_c(s, s_0) \phi''(c) \, \mathrm{d}c$$

with $\phi_c(s, s_0) = |s - c| \cdot [\![(s - c)(s_0 - c) < 0]\!]$, where $[\![\circ]\!]$ is 1 if the statement $\circ$ is true and it is 0 otherwise. For $s_0 := \frac{y}{1+y}$ and $s := \frac{x}{1+x}$, we obtain

$$d_\phi\left(\frac{x}{1+x}, \frac{y}{1+y}\right) = \int_{\frac{y \wedge x}{1+y \wedge x}}^{\frac{x \vee y}{1+x \vee y}} \left|\frac{x}{1+x} - z\right| \cdot \phi''(z) \, \mathrm{d}z.$$

Applying the substitution $z = \frac{u}{1+u}$ with $\mathrm{d}z = \frac{\mathrm{d}u}{(1+u)^2}$ gives

$$\begin{aligned}
d_\phi\left(\frac{x}{1+x}, \frac{y}{1+y}\right) &= \int_{y \wedge x}^{x \vee y} \left|\frac{x}{1+x} - \frac{u}{1+u}\right| \cdot \phi''\left(\frac{u}{1+u}\right) \frac{1}{(1+u)^2} \, \mathrm{d}u \\
&= \int_{y \wedge x}^{x \vee y} \frac{|x - u|}{(1+x)(1+u)} \cdot \phi''\left(\frac{u}{1+u}\right) \frac{1}{(1+u)^2} \, \mathrm{d}u \\
&= \frac{1}{1+x} \int_{y \wedge x}^{x \vee y} |x - u| \cdot \phi''\left(\frac{u}{1+u}\right) \frac{1}{(1+u)^3} \, \mathrm{d}u \\
&= \frac{1}{1+x} d_{\phi^\diamond}(x, y),
\end{aligned}$$

where the last line uses

$$(\phi^\diamond)'(z) = \frac{1}{1+x} \phi'\left(\frac{z}{1+z}\right) + \phi\left(\frac{z}{1+z}\right) \text{ and } (\phi^\diamond)''(z) = \phi''\left(\frac{z}{1+z}\right) \frac{1}{(1+z)^3}.$$

$\square$

We are now ready to provide a proof of Lemma 1. For convenience of the reader, we don't just repeat the arguments of Menon & Ong (2016, Proposition 3), but we also include the required basis arguments of Reid & Williamson (2009) and Reid & Williamson (2010).

*Proof of Lemma 1.* The properness of $\ell$ and Theorem 3 imply

$$L(\eta, \hat{y}) = \underline{L}(\Psi^{-1}(\hat{y})) + (\eta - \Psi^{-1}(\hat{y})) \underline{L}'(\Psi^{-1}(\hat{y})). \tag{24}$$

Denoting by $\eta(x) := \rho(y = 1|x)$ and following (Reid & Williamson, 2010, Corollary 13), the excess risk can be expressed as an expected Bregman divergence as follows:

$$\begin{aligned}
\mathcal{R}(f) - \mathcal{R}(f^*) &= \\
&= \mathbb{E}_{(x,y) \sim \rho}[\ell(y, f(x))] - \mathbb{E}_{(x,y) \sim \rho}[\ell(y, f^*(x))] \\
&= \mathbb{E}_{x \sim \rho_\mathcal{X}}[L(\eta(x), f(x))] - \mathbb{E}_{x \sim \rho_\mathcal{X}}[\underline{L}(\eta(x))] \\
&= \mathbb{E}_{x \sim \rho_\mathcal{X}}\left[\underline{L}(\Psi^{-1}(f(x))) + (\eta(x) - \Psi^{-1}(f(x))) \underline{L}'(\Psi^{-1}(f(x))) - \underline{L}(\eta(x))\right] \\
&= \mathbb{E}_{x \sim \rho_\mathcal{X}}\left[d_{-\underline{L}}(\eta(x), \Psi^{-1}(f(x)))\right] \tag{25}
\end{aligned}$$

with $d_\phi$ as defined in Lemma 3. Note that the Bregman generator $-\underline{L}$ is strictly convex, since $\underline{L}$ is concave due to Theorem 3 and it is strictly concave due to the uniqueness of the minimum $\underline{L}(\eta) = \min_{\hat{y} \in \mathbb{R}} L(\eta, \hat{y})$.

Following the proof of Menon & Ong (2016, Proposition 3), in the next step, we apply the Radon-Nikodým theorem to obtain:

$$\begin{aligned}
&\mathbb{E}_{x \sim \rho_\mathcal{X}}\left[d_{-\underline{L}}(\eta(x), \Psi^{-1}(f(x)))\right] \\
&= \frac{1}{2} \mathbb{E}_{x \sim P}\left[d_{-\underline{L}}(\eta(x), \Psi^{-1}(f(x)))\right] + \frac{1}{2} \mathbb{E}_{x \sim Q}\left[d_{-\underline{L}}(\eta(x), \Psi^{-1}(f(x)))\right] \\
&= \frac{1}{2} \mathbb{E}_{x \sim Q}\left[\beta(x) d_{-\underline{L}}(\eta(x), \Psi^{-1}(f(x)))\right] + \frac{1}{2} \mathbb{E}_{x \sim Q}\left[d_{-\underline{L}}(\eta(x), \Psi^{-1}(f(x)))\right] \\
&= \frac{1}{2} \mathbb{E}_{x \sim Q}\left[(1 + \beta(x)) d_{-\underline{L}}(\eta(x), \Psi^{-1}(f(x)))\right].
\end{aligned}$$

Bayes' theorem gives

$$
\begin{aligned}
\beta(x) &= \frac{\rho(x|y=1)}{\rho(x|y=-1)} = \frac{\rho(x,y=1)\rho_{\mathcal{Y}}(y=-1)}{\rho(x,y=-1)\rho_{\mathcal{Y}}(y=1)} = \frac{\rho(x,y=1)\rho_{\mathcal{X}}(x)}{\rho(x,y=-1)\rho_{\mathcal{X}}(x)} \\
&= \frac{\rho(y=1|x)}{\rho(y=-1|x)} = \frac{\rho(y=1|x)}{1-\rho(y=1|x)} = \frac{\eta(x)}{1-\eta(x)}
\end{aligned}
$$

which implies $\eta(x) = \frac{\beta(x)}{1+\beta(x)}$ (Bickel et al., 2009). Applying Lemma 3 yields the desired result. □

### A.4 CHARACTERIZATION OF CONVEX LOSSES

A characterization of *convex* proper composite loss functions can be stated as follows.

**Theorem 4** (Reid & Williamson (2010, Theorem 24)). *Consider a strictly proper composite loss function $\ell : \{-1,1\} \times \mathbb{R} \to \mathbb{R}$ with invertible link $\Psi : [0,1] \to \mathbb{R}$ such that $(\Psi^{-1})'(z) > 0$ for all $z \in \mathbb{R}$ and denote by $w(\widehat{\eta}) := \widehat{\eta}\,\ell_1'(\Psi(\widehat{\eta}))$. Then $\ell_1$ and $\ell_{-1}$ are convex if and only if*

$$
-\frac{1}{\widehat{\eta}} \leq \frac{w'(\widehat{\eta})}{w(\widehat{\eta})} - \frac{\Psi''(\widehat{\eta})}{\Psi'(\widehat{\eta})} \leq \frac{1}{1-\widehat{\eta}}, \quad \forall \widehat{\eta} \in (0,1). \tag{26}
$$

Theorem 4 suggests a *canonical link* (Buja et al., 2005) for convex losses is $\Psi_{\text{can}}'(\widehat{\eta}) := w(\widehat{\eta})$, which corresponds to the notion of "matching loss" studied by Kivinen & Warmuth (1997); Helmbold et al. (1999).

The following proof from Reid & Williamson (2010) reads in our notation as follows.

*Proof of Theorem 4.* Let us denote by $q(\widehat{y}) := \Psi^{-1}(\widehat{y})$. Then, the definition of L and Thm. 2, give

$$
\begin{aligned}
\frac{\partial}{\partial z} \mathrm{L}(\eta, z)\Big|_{z=\widehat{y}} &= (1-\eta)\ell_{-1}'(q(z))q'(z) + \eta\ell_1'(q(z))q'(z)\Big|_{z=\widehat{y}} \\
&= (1-\eta)w(q(\widehat{y}))q(\widehat{y})q'(z) + \eta(q(\widehat{y})-1)w(q(\widehat{y}))q'(z) \\
&= (q(\widehat{y})-\eta)w(q(\widehat{y}))q'(\widehat{y}). 
\end{aligned} \tag{27}
$$

A necessary and sufficient condition for $\ell_y$ for being convex for $\eta \in \{0,1\}$ is that

$$
\frac{\partial^2}{\partial z^2} \mathrm{L}(\eta, z)\Big|_{z=\widehat{y}} \geq 0, \quad \forall \widehat{y} \in \mathbb{R}.
$$

From Eq. 27, we obtain

$$
[w(q(\widehat{y}))q'(\widehat{y})]' \, (q(\widehat{y}) - [\![\eta=1]\!]) + w(q(\widehat{y}))q'(\widehat{y})q'(\widehat{y}) \geq 0, \quad \forall \widehat{y} \in \mathbb{R}, \tag{28}
$$

where

$$
[w(q(\widehat{y}))q'(\widehat{y})]' := \frac{\partial}{\partial \widehat{y}} w(q(\widehat{y}))q'(\widehat{y}).
$$

As it is noted in (Reid & Williamson, 2010), Eq. 28 can be found in (Buja et al., 2005, Eq. (39)). To proceed further, note that Eq. 28 leads, for $y \in \{-1,1\}$ to the pair of inequalities

$$
\begin{aligned}
w(q(\widehat{y}))(q'(\widehat{y}))^2 &\geq -q(\widehat{y}) \, [w(q(\widehat{y}))q'(\widehat{y})]' \\
w(q(\widehat{y}))(q'(\widehat{y}))^2 &\geq (1-q(\widehat{y})) \, [w(q(\widehat{y}))q'(\widehat{y})]'
\end{aligned} \tag{29}
$$

which holds for all $\widehat{y} \in \mathbb{R}$. Note that Eq. 29 holds for $q(\cdot) = 0$ and $1-q(\cdot) = 0$ because $q'(\cdot) > 0$ and $w(\cdot) > 0$. Without loss of generality, in the following, we restrict our attention to the set $\Psi((0,1)) = \{x \in \mathbb{R} \mid q(x) \neq 0, 1-q(x) \neq 0\}$. From Eq. 29 we equivalently have

$$
\frac{(q'(\widehat{y}))^2}{1-q(\widehat{y})} \geq \frac{[w(q(\widehat{y}))q'(\widehat{y})]'}{w(q(\widehat{y}))} \geq \frac{-(q'(\widehat{y}))^2}{q(\widehat{y})}, \quad \forall \widehat{y} \in \Psi((0,1)), \tag{30}
$$

where we have used the fact that $-q(\widehat{y}) = -\Psi^{-1}(\widehat{y})$ is always negative, that $q(\widehat{y}), 1-q(\widehat{y})$ are non-zero for $\widehat{y} \in \Psi((0,1))$ and $w(\cdot) > 0$ for *strictly* proper loss function $\ell$. Using that

$$
[w(q(\cdot))q'(\cdot)]' = w'(q(\cdot))q'(\cdot)q'(\cdot) + w(q(\cdot))q''(\cdot)
$$

we obtain, equivalently to Eq. 30,

$$\frac{1}{1 - q(\widehat{y})} \geq \frac{w'(q(\widehat{y}))}{w(q(\widehat{y}))} + \frac{q''(\widehat{y})}{(q'(\widehat{y}))^2} \geq \frac{-1}{q(\widehat{y})}, \quad \forall \widehat{y} \in \Psi((0, 1)). \tag{31}$$

By substituting $\widehat{\eta} := q(\widehat{y})$ so that $\widehat{y} = q^{-1}(\widehat{\eta}) = \Psi(\widehat{\eta})$, we get

$$\frac{1}{1 - \widehat{\eta}} \geq \frac{w'(\widehat{\eta})}{w(\widehat{\eta})} + \frac{q''(\Psi(\widehat{\eta}))}{(q'(\Psi(\widehat{\eta})))^2} \geq \frac{-1}{\widehat{\eta}}, \quad \forall \widehat{\eta} \in (0, 1). \tag{32}$$

Note that $\frac{1}{q'(\Psi(\widehat{\eta}))} = \frac{1}{q'(q^{-1}(\widehat{\eta}))} = (q^{-1})'(\widehat{\eta}) = \Psi'(\widehat{\eta})$ and Eq. 32 is equivalent to

$$\frac{1}{1 - \widehat{\eta}} \geq \frac{w'(\widehat{\eta})}{w(\widehat{\eta})} + q''(\Psi(\widehat{\eta})) \left( \Psi'(\widehat{\eta}) \right)^2 \geq \frac{-1}{\widehat{\eta}}, \quad \forall \widehat{\eta} \in (0, 1). \tag{33}$$

Observe that $q'(\cdot) = (\psi^{-1})'(\cdot) = \frac{1}{\Psi'(\Psi^{-1}(\cdot))}$ and therefore

$$q''(\cdot) = (\Psi^{-1})''(\cdot) = \left( \frac{1}{\Psi'(\Psi^{-1}(\cdot))} \right)' = \frac{-1}{(\Psi'(\Psi^{-1}(\cdot)))^3} \Psi''(\Psi^{-1}(\cdot)),$$

which implies that

$$q''(\Psi(\widehat{\eta})) \left( \Psi'(\widehat{\eta}) \right)^2 = \frac{-1}{(\Psi'(\widehat{\eta}))^3} \Psi''(\widehat{\eta})(\Psi'(\widehat{\eta}))^2 = -\frac{\Psi''(\widehat{\eta})}{\Psi'(\widehat{\eta})}.$$

Plugging this into Eq. 33, which is equivalent to the convexity of $\ell$ in its second argument, completes the proof. $\qquad \square$

## B   PROOFS OF NOVEL RESULTS

### B.1   KEY LEMMA ON CONDITIONAL BAYES RISK

Before proving our main theorem, we characterize the conditional Bayes risk $\underline{\mathsf{L}}$ of a strictly proper composite loss function $\ell : \{-1, 1\} \times \mathbb{R} \to \mathbb{R}$ that satisfies the questioned Bregman representation $\mathcal{R}(f) - \mathcal{R}(f^*) = \frac{1}{2} B_\phi(\beta, g \circ f)$. We start with the following technical lemma.

**Lemma 4.** *Let $\phi : [0, \infty) \to \mathbb{R}$ be convex and twice differentiable and denote by*

$$\zeta(z) := -\phi \left( \frac{z}{1 - z} \right) (1 - z).$$

*If it holds that*

$$\mathbb{E}_{x \sim \rho_{\mathcal{X}}} \left[ d_{-\underline{\mathsf{L}}}(\rho(y = 1|x), \widehat{u}) \right] = \mathbb{E}_{x \sim \rho_{\mathcal{X}}} \left[ d_{-\zeta}(\rho(y = 1|x), \widehat{u}) \right]$$

*for all $\widehat{u} \in [0, 1]$ and all $P, Q \in \mathcal{M}_1^+(\mathcal{X})$ with $P \ll Q$, then there exist $c_1, c_2 \in \mathbb{R}$ such that*

$$\underline{\mathsf{L}}(b) = \zeta(b) + c_2 b + c_1, \quad \forall b \in [0, 1]. \tag{34}$$

*Proof.* With $u := \rho(y = 1|x)$ it holds that

$$\mathbb{E}_{x \sim \rho_{\mathcal{X}}} \left[ d_{-\underline{\mathsf{L}}}(u, \widehat{u}) \right] = \mathbb{E}_{x \sim \rho_{\mathcal{X}}} \left[ d_{-\zeta}(u, \widehat{u}) \right]$$

$$\mathbb{E}_{x \sim \rho_{\mathcal{X}}} \left[ -\underline{\mathsf{L}}(u) + \underline{\mathsf{L}}(\widehat{u}) + \underline{\mathsf{L}}'(\widehat{u})(u - \widehat{u}) \right] = \mathbb{E}_{x \sim \rho_{\mathcal{X}}} \left[ -\zeta(u) + \zeta(\widehat{u}) + \zeta'(\widehat{u})(u - \widehat{u}) \right]$$

for all $P, Q \in \mathcal{M}_1^+(\mathcal{X})$ with $P \ll Q$ and for all $\widehat{u} \in [0, 1]$. In particular, for $\widehat{u} := 1$ we have

$$\mathbb{E}_{x \sim \rho_{\mathcal{X}}} \left[ \zeta(u) - \underline{\mathsf{L}}(u) \right] =$$
$$= \mathbb{E}_{x \sim \rho_{\mathcal{X}}} \left[ \zeta(1) - \underline{\mathsf{L}}(1) \right] + \mathbb{E}_{x \sim \rho_{\mathcal{X}}} \left[ (\zeta'(1) - \underline{\mathsf{L}}'(1))(u - 1) \right]$$
$$= \mathbb{E}_{x \sim \rho_{\mathcal{X}}} \left[ \zeta(1) - \underline{\mathsf{L}}(1) \right] + \mathbb{E}_{x \sim \rho_{\mathcal{X}}} \left[ u(\zeta'(1) - \underline{\mathsf{L}}'(1)) \right] + \mathbb{E}_{x \sim \rho_{\mathcal{X}}} \left[ -1(\zeta'(1) - \underline{\mathsf{L}}'(1)) \right]$$
$$= \mathbb{E}_{x \sim \rho_{\mathcal{X}}} \left[ \zeta(1) - \underline{\mathsf{L}}(1) \right] + \mathbb{E}_{x \sim \rho_{\mathcal{X}}} \left[ u \right] \left( \zeta'(1) - \underline{\mathsf{L}}'(1) \right) - 1(\zeta'(1) - \underline{\mathsf{L}}'(1)) = c,$$

where

$$\mathbb{E}_{x \sim \rho_{\mathcal{X}}}[u] = \mathbb{E}_{x \sim \rho_{\mathcal{X}}}[\rho(y = 1|x)] = \frac{1}{2}$$

and

$$c := \mathbb{E}_{x \sim \rho_{\mathcal{X}}}\left[\zeta(1) - \underline{L}(1)\right] + \frac{1}{2}\left(\zeta'(1) - \underline{L}'(1)\right) - 1(\zeta'(1) - \underline{L}'(1))$$

is constant. To summarize the steps above, we may state with $\varphi := \zeta - \underline{L}$ that

$$\mathbb{E}_{x \sim \rho_{\mathcal{X}}}[\varphi(\rho(y = 1|x))] = c \tag{35}$$

for all $P, Q \in \mathcal{M}_1^+(\mathcal{X})$ with $P \ll Q$.

In the following, we will construct probability measures witnessing the affinity of $\varphi$. Let us therefore assume without loss of generality[9] that $0 \in \mathcal{X}$ is in the interior of $\mathcal{X}$ such that we can fix some $0 < \epsilon \leq 1$ satisfying $[0, \epsilon]^d \subseteq \mathcal{X}$.

We now distinguish between two cases. For the first case, let $r \geq 1$ and define two measures $P_r, Q \in \mathcal{M}_+^1(\mathcal{X})$ by

$$P_r(x) := \begin{cases} \frac{r}{\epsilon} \cdot \delta_0(x^{(2)}) \cdots \delta_0(x^{(d)}) & \text{if } x^{(1)} \in [0, \frac{\epsilon}{r}] \\ 0 & \text{otherwise} \end{cases}$$

$$Q(x) := \begin{cases} \frac{1}{\epsilon} \cdot \delta_0(x^{(2)}) \cdots \delta_0(x^{(d)}) & \text{if } x^{(1)} \in [0, \epsilon] \\ 0 & \text{otherwise} \end{cases},$$

where $x^{(i)}$ denotes the $i$-th element of the vector $x \in \mathcal{X}$ and $\delta_z(x)$ is the Dirac measure being equal to 0 whenever $x \neq z$. Both measures $P_r$ and $Q$ are probability measures since

$$\int_{\mathcal{X}} \mathrm{d}P_r(x) = \int_0^\epsilon \cdots \int_0^\epsilon \int_0^{\frac{\epsilon}{r}} \frac{r}{\epsilon} \, \mathrm{d}x^{(1)} \, \mathrm{d}\delta_0(x^{(2)}) \ldots \mathrm{d}\delta_0(x^{(d)}) = 1$$

and $\int_{\mathcal{X}} \mathrm{d}Q(x) = 1$ analogously. It holds that

$$\rho(y = 1|x) = \frac{\rho(x|y = 1)\rho_{\mathcal{Y}}(y = 1)}{\rho_{\mathcal{X}}(x)}$$

$$= \frac{P_r(x)}{P_r(x) + Q(x)}$$

$$= \begin{cases} \frac{r}{r+1} & \text{if } x^{(1)} \in [0, \frac{\epsilon}{r}] \text{ and } x^{(2)} = \ldots = x^{(d)} = 0 \\ 0 & \text{otherwise} \end{cases}.$$

It follows that

$$\mathbb{E}_{x \sim \rho_{\mathcal{X}}}[\varphi(\rho(y = 1|x))] =$$

$$= \frac{1}{2} \int_{\mathcal{X}} \varphi(\rho(y = 1|x)) \, \mathrm{d}P_r(x) + \frac{1}{2} \int_{\mathcal{X}} \varphi(\rho(y = 1|x)) \, \mathrm{d}Q(x)$$

$$= \frac{1}{2} \int_0^\epsilon \cdots \int_0^\epsilon \int_0^{\frac{\epsilon}{r}} \frac{r}{\epsilon} \varphi\left(\frac{r}{r+1}\right) \mathrm{d}x^{(1)} \, \mathrm{d}\delta_0(x^{(2)}) \ldots \mathrm{d}\delta_0(x^{(d)})$$

$$+ \frac{1}{2} \int_0^\epsilon \cdots \int_0^\epsilon \int_0^{\frac{\epsilon}{r}} \frac{1}{\epsilon} \varphi\left(\frac{r}{r+1}\right) \mathrm{d}x^{(1)} \, \mathrm{d}\delta_0(x^{(2)}) \ldots \mathrm{d}\delta_0(x^{(d)})$$

$$+ \frac{1}{2} \int_0^\epsilon \cdots \int_0^\epsilon \int_{\frac{\epsilon}{r}}^\epsilon \frac{1}{\epsilon} \varphi(0) \, \mathrm{d}x^{(1)} \, \mathrm{d}\delta_0(x^{(2)}) \ldots \mathrm{d}\delta_0(x^{(d)})$$

$$= \frac{1}{2}\varphi\left(\frac{r}{r+1}\right) + \frac{1}{2r}\varphi\left(\frac{r}{r+1}\right) + \frac{1}{2}\varphi(0)\left(1 - \frac{1}{r}\right)$$

$$= \varphi(b)\frac{1}{2b} + \varphi(0)\frac{2b - 1}{2b}$$

---

[9]We can always choose some arbitrary but fixed inner point $\overline{x} \in \mathcal{X}$ and shift all our constructions by choosing $x - \overline{x}$ instead of $x$.

for $b := \frac{r}{r+1} \in [\frac{1}{2}, 1]$ with $b = 1$ in the limit. As a consequence, we get from Eq. 35

$$f(b) = b(2c - 2\varphi(0)) + \varphi(0) \tag{36}$$

for all $b \in [\frac{1}{2}, 1]$.

For the second case, let $0 \le r \le 1$ and define the probability measures

$$P_r(x) := \begin{cases} \frac{r}{\epsilon} \cdot \delta_0(x^{(2)}) \cdots \delta_0(x^{(d)}) & \text{if } x^{(1)} \in [0, \frac{\epsilon}{2}] \\ \frac{2-r}{\epsilon} \cdot \delta_0(x^{(2)}) \cdots \delta_0(x^{(d)}) & \text{if } x^{(1)} \in [\frac{\epsilon}{2}, \epsilon] \\ 0 & \text{otherwise} \end{cases}$$

$$Q(x) := \begin{cases} \frac{2-r}{\epsilon} \cdot \delta_0(x^{(2)}) \cdots \delta_0(x^{(d)}) & \text{if } x^{(1)} \in [0, \frac{\epsilon}{2}] \\ \frac{r}{\epsilon} \cdot \delta_0(x^{(2)}) \cdots \delta_0(x^{(d)}) & \text{if } x^{(1)} \in [\frac{\epsilon}{2}, \epsilon] \\ 0 & \text{otherwise} \end{cases}.$$

It holds that

$$\rho(y = 1|x) = \begin{cases} \frac{r}{2} & \text{if } x^{(1)} \in [0, \frac{\epsilon}{2}] \text{ and } x^{(2)} = \ldots = x^{(d)} = 0 \\ 1 - \frac{r}{2} & \text{if } x^{(1)} \in [\frac{\epsilon}{2}, \epsilon] \text{ and } x^{(2)} = \ldots = x^{(d)} = 0 \\ 0 & \text{otherwise} \end{cases}$$

which gives

$$\mathbb{E}_{x \sim \rho_{\mathcal{X}}}[\varphi(\rho(y = 1|x))] =$$
$$= \frac{r}{4}\varphi\left(\frac{r}{2}\right) + \frac{2-r}{4}\varphi\left(1 - \frac{r}{2}\right) + \frac{2-r}{4}\varphi\left(\frac{r}{2}\right) + \frac{r}{2}\varphi\left(1 - \frac{r}{2}\right)$$
$$= \frac{1}{2}\varphi\left(\frac{r}{2}\right) + \frac{1}{2}\varphi\left(1 - \frac{r}{2}\right). \tag{37}$$

Applying Eq. 36 with $\widetilde{b} := 1 - \frac{r}{2} \in [\frac{1}{2}, 1]$ gives

$$\varphi(\widetilde{b}) = \varphi\left(1 - \frac{r}{2}\right) = \left(1 - \frac{r}{2}\right)(2c - 2\varphi(0)) + \varphi(0)$$

which yields, together with Eq. 37,

$$c = \frac{1}{2}\varphi\left(\frac{r}{2}\right) + \frac{1}{2}\left(1 - \frac{r}{2}\right)(2c - 2\varphi(0)) + \frac{1}{2}\varphi(0)$$
$$-\varphi\left(\frac{r}{2}\right) = (2c - 2\varphi(0))\left(1 - \frac{r}{2}\right) + 2\varphi(0) - 2c - \varphi(0)$$
$$\varphi\left(\frac{r}{2}\right) = (2c - 2\varphi(0))\left(\frac{r}{2} - 1\right) + (2c - 2\varphi(0)) + \varphi(0)$$
$$\varphi\left(\frac{r}{2}\right) = \frac{r}{2}(2c - 2\varphi(0)) + \varphi(0).$$

Choosing $b := \frac{r}{2} \in [0, \frac{1}{2}]$ proves, as in the first case, that $\varphi$ is an affine function:

$$\varphi(b) = b(2c - 2\varphi(0)) + \varphi(0). \tag{38}$$

The statement of the lemma follows for $c_1 := \varphi(0)$ and $c_2 := (2c - 2\varphi(0))$. $\qquad \square$

With our technical result Lemma 4 above, we are now able to characterize the conditional Bayes risk of loss functions satisfy the questioned Bregman representation $\mathcal{R}(f) - \mathcal{R}(f^*) = \frac{1}{2}B_\phi(\beta, g \circ f)$.

**Lemma 5.** *Let $\ell : \{-1, 1\} \times \mathbb{R} \to \mathbb{R}$ be strictly proper composite with invertible link function $\Psi : [0, 1] \to \mathbb{R}$ and denote by $g(\widehat{y}) := \frac{\Psi^{-1}(\widehat{y})}{1 - \Psi^{-1}(\widehat{y})}$. If there is a strictly convex twice differentiable $\phi : [0, \infty) \to \mathbb{R}$ such that*

$$\mathcal{R}(f) - \mathcal{R}(f^*) = \frac{1}{2}B_\phi(\beta, g \circ f), \quad \forall P, Q \in \mathcal{M}_1^+(\mathcal{X}) : P \ll Q, \forall f \in L^1(Q), \tag{39}$$

*then*

$$\underline{L}(\widehat{y}) = -\phi\left(\frac{\Psi^{-1}(\widehat{y})}{1 - \Psi^{-1}(\widehat{y})}\right)(1 - \Psi^{-1}(\widehat{y})) + c_2\Psi^{-1}(\widehat{y}) + c_1, \quad \forall \widehat{y} \in \mathbb{R}, \tag{40}$$

*for some $c_1, c_2 \in \mathbb{R}$.*

The following proof of Lemma 5 has three parts. The first part inverts the constructions developed in (Menon & Ong, 2016, Appendix B), the second part is to apply (Reid & Williamson, 2010, Corollary 13), and, the third part is to apply Lemma 4.

*Proof.* From Eq. 39 and Bayes' theorem, we know that

$$2(\mathcal{R}(f) - \mathcal{R}(f^*)) = B_\phi(\beta, g \circ f) = \mathbb{E}_{x \sim Q}\left[d_\phi\left(\frac{\Psi^{-1}(f^*(x))}{1 - \Psi^{-1}(f^*(x))}, \frac{\Psi^{-1}(f(x))}{1 - \Psi^{-1}(f(x))}\right)\right].$$

Applying Lemma 3 with $\widetilde{\phi}^\diamond := \phi$, $\widetilde{x} := \frac{\Psi^{-1}(f^*(x))}{1-\Psi^{-1}(f^*(x))}$ and $\widetilde{y} := \frac{\Psi^{-1}(f(x))}{1-\Psi^{-1}(f(x))}$, such that $\frac{\widetilde{x}}{1+\widetilde{x}} = \Psi^{-1}(f^*(x))$ and $\frac{\widetilde{y}}{1+\widetilde{y}} = \Psi^{-1}(f(x))$, gives

$$d_\phi\left(\frac{\Psi^{-1}(f^*(x))}{1 - \Psi^{-1}(f^*(x))}, \frac{\Psi^{-1}(f(x))}{1 - \Psi^{-1}(f(x))}\right) = (1 + \beta(x))d_{-\zeta}\left(\Psi^{-1}(f^*(x)), \Psi^{-1}(f(x))\right)$$

for

$$\zeta(z) := -\phi\left(\frac{z}{1-z}\right)(1 - z),$$

where we have substituted $z = \frac{u}{1-u}$ in the definition of $\widetilde{\phi}^\diamond(z)$ in Lemma 3 to obtain

$$\widetilde{\phi}(u) = (1 - u)\widetilde{\phi}^\diamond\left(\frac{u}{1-u}\right) = (1 - u)\phi\left(\frac{u}{1-u}\right) = -\zeta(u),$$

which is twice differentiable and strictly convex as $\phi$ is twice differentiable and strictly convex. Denoting by $u := \Psi^{-1}(f^*(x))$ and $\widehat{u} := \Psi^{-1}(f(x))$ and summarizing the steps above, we have

$$2(\mathcal{R}(f) - \mathcal{R}(f^*)) = \mathbb{E}_{x \sim Q}\left[(1 + \beta(x))d_{-\zeta}(u, \widehat{u})\right].$$

The transformation used in the proof of (Menon & Ong, 2016, Proposition 3) is the following:

$$\begin{aligned}
\mathbb{E}_{x \sim Q}\left[(1 + \beta(x))d_{-\zeta}(u, \widehat{u})\right] &= \frac{1}{2}\mathbb{E}_{x \sim Q}\left[\beta(x)d_{-\zeta}(u, \widehat{u})\right] + \frac{1}{2}\mathbb{E}_{x \sim Q}\left[d_{-\zeta}(u, \widehat{u})\right] \\
&= \frac{1}{2}\mathbb{E}_{x \sim P}\left[d_{-\zeta}(u, \widehat{u})\right] + \frac{1}{2}\mathbb{E}_{x \sim Q}\left[d_{-\zeta}(u, \widehat{u})\right] \\
&= \mathbb{E}_{x \sim \rho_{\mathcal{X}}}\left[d_{-\zeta}(u, \widehat{u})\right],
\end{aligned}$$

which gives

$$2(\mathcal{R}(f) - \mathcal{R}(f^*)) = \mathbb{E}_{x \sim \rho_{\mathcal{X}}}\left[d_{-\zeta}(u, \widehat{u})\right]. \tag{41}$$

At the same time, we know from Reid & Williamson (2010, Corollary 13) that (see Eq. 25 for a proof)

$$2(\mathcal{R}(f) - \mathcal{R}(f^*)) = \mathbb{E}_{x \sim \rho_{\mathcal{X}}}\left[d_{-\underline{L}}(u, \widehat{u})\right]. \tag{42}$$

From the strict properness of $\ell$ we get that $u = \rho(y = 1|x)$ and Eq. 40 follows from Lemma 4. $\square$

## B.2 Main Characterization of Losses

Next, we prove our main characterization result.

*Proof of Theorem 1.* To prove sufficiency of Eq. 8, note that the conditional Bayes risk of the loss function in Eq. 8 satisfies

$$\begin{aligned}
\mathrm{L}(\eta, \widehat{y}) &= \eta\ell(1, \widehat{y}) + (1 - \eta)\ell(-1, \widehat{y}) \\
&= \gamma(\Psi^{-1}(\widehat{y})) + \left(\eta - \Psi^{-1}(\widehat{y})\right)\gamma'(\Psi^{-1}(\widehat{y}))
\end{aligned}$$

with strictly concave $\gamma$ defined by the strictly convex function $\phi$. It follows from Theorem 3 that $\ell$ with $\underline{L} = \gamma \circ \Psi^{-1}$ is proper composite with link $\Psi : [0, 1] \to \mathbb{R}$. Moreover, $\ell$ is strictly proper composite since $\underline{L}''(\widehat{y}) = \gamma(\Psi^{-1}(\widehat{y})) < 0$ for all $\widehat{y} \in \mathbb{R}$. Applying Lemma 1 gives

$$\mathcal{R}(f) - \mathcal{R}(f^*) = \frac{1}{2}B_{-\underline{L}^\diamond}(\beta, g \circ f) = \frac{1}{2}B_{-\gamma^\diamond}(\beta, g \circ f) = \frac{1}{2}B_\phi(\beta, g \circ f)$$

because $\gamma^\diamond(z) = \phi(z) - c_2 z - c_1$, the fact that Bregman divergences are unique up to affine terms of the generator and the equality $g(y) = \frac{\Psi^{-1}(y)}{1-\Psi^{-1}(y)}$ which follows from the definition of $\Psi^{-1}$.

For proving necessity of Eq. 8 assume $\ell$ is a strictly proper composite loss function with invertible link function $\Psi : [0,1] \to \mathbb{R}$ satisfying Eq. 7. From $\Psi^{-1}(\widehat{y}) = \frac{g(\widehat{y})}{1+g(\widehat{y})}$ we obtain $g(\widehat{y}) = \frac{\Psi^{-1}(\widehat{y})}{1-\Psi^{-1}(\widehat{y})}$. That is, all requirements for Lemma 5 are satisfied and it's application gives, for all $\widehat{\eta} \in [0,1]$,

$$\underline{L}(\widehat{y}) = \gamma\big(\Psi^{-1}(\widehat{y})\big). \tag{43}$$

Next, we construct a strictly proper loss function in the same way as it is done in (Reid & Williamson, 2010, Theorem 7) and (Bao, 2023, Corollary 3). We use Theorem 3 to obtain

$$L(\eta, \widehat{y}) = \gamma(\Psi^{-1}(\widehat{y})) + \big(\eta - \Psi^{-1}(\widehat{y})\big)\, \gamma'\big(\Psi^{-1}(\widehat{y})\big)$$

for all $\eta \in [0,1]$ and $\widehat{y} \in \mathbb{R}$, which can be applied since $\Psi^{-1}(\widehat{y}) \in (0,1)$ as $g$ is strictly monotonically increasing. The form of the loss function in Eq. 8 follows now from the definition

$$L(\eta, \widehat{y}) = \eta\ell(1, \widehat{y}) + (1-\eta)\ell(-1, \widehat{y}),$$

for $\eta = 0$ and $\eta = 1$. $\qquad\square$

### B.3 CHARACTERIZATION OF CONVEX LOSSES

Next, we prove our characterization of convex losses by properties of $\phi, g$.

*Proof of Corollary 1.* Our aim is to apply Theorem 4. In the following denote by $\widehat{\eta} := \Psi^{-1}(\widehat{y})$. Applying Theorem 2 with $w$ as defined in Eq. 19 gives

$$\begin{aligned}
\frac{\partial}{\partial \widehat{y}} L(\eta, \widehat{y}) &= (1-\eta)\ell'_{-1}(\widehat{y}) + \eta\ell'_1(\widehat{y}) \\
&= \eta\,(\widehat{\eta}-1)\,w(\widehat{\eta}) + (1-\eta)\,\widehat{\eta}\,w(\widehat{\eta}) \\
&= (\widehat{\eta}-\eta)w(\widehat{\eta})
\end{aligned}$$

and, together with Theorem 3, we have

$$\begin{aligned}
w(\widehat{\eta}) &= \frac{1}{\eta - \widehat{\eta}}\frac{\partial}{\partial \widehat{y}} L(\eta, \widehat{y}) \\
&= \frac{1}{\eta - \widehat{\eta}}\frac{\partial}{\partial \widehat{y}}\big(\underline{L}(\widehat{\eta}) + (\eta - \widehat{\eta})\,\underline{L}'(\widehat{\eta})\big) \\
&= \frac{1}{\eta - \widehat{\eta}}\frac{\partial}{\partial \widehat{y}}\big(\underline{L}'(\widehat{\eta}) - \underline{L}'(\widehat{\eta}) + (\eta - \widehat{\eta})\,\underline{L}''(\widehat{\eta})\big) \\
&= -\underline{L}''(\widehat{\eta}) \\
&= -\gamma''(\widehat{\eta}),
\end{aligned}$$

where the last equality follows from Lemma 5. Note that

$$\gamma'(\widehat{\eta}) = \phi\left(\frac{\widehat{\eta}}{1-\widehat{\eta}}\right) + \frac{1}{\widehat{\eta}-1}\phi'\left(\frac{\widehat{\eta}}{1-\widehat{\eta}}\right) + c_1$$

$$\gamma''(\widehat{\eta}) = \frac{1}{(\widehat{\eta}-1)^3}\phi''\left(\frac{\widehat{\eta}}{1-\widehat{\eta}}\right)$$

$$\gamma'''(\widehat{\eta}) = \frac{1}{(\widehat{\eta}-1)^5}\phi'''\left(\frac{\widehat{\eta}}{1-\widehat{\eta}}\right) - \frac{3}{(\widehat{\eta}-1)^4}\phi''\left(\frac{\widehat{\eta}}{1-\widehat{\eta}}\right)$$

and

$$\Psi(\widehat{\eta}) = g^{-1}\left(\frac{\widehat{\eta}}{1-\widehat{\eta}}\right)$$

$$\Psi'(\widehat{\eta}) = \frac{1}{(1-\widehat{\eta})^2}\,(g^{-1})'\left(\frac{\widehat{\eta}}{1-\widehat{\eta}}\right)$$

$$\Psi''(\widehat{\eta}) = \frac{1}{(1-\widehat{\eta})^4}\,(g^{-1})''\left(\frac{\widehat{\eta}}{1-\widehat{\eta}}\right) - \frac{2}{(1-\widehat{\eta})^3}\,(g^{-1})'\left(\frac{\widehat{\eta}}{1-\widehat{\eta}}\right),$$

which gives

$$\frac{w'(\widehat{\eta})}{w(\widehat{\eta})} = \frac{\phi'''\left(\frac{\widehat{\eta}}{1-\widehat{\eta}}\right)}{\phi''\left(\frac{\widehat{\eta}}{1-\widehat{\eta}}\right)(\widehat{\eta}-1)^2} - \frac{3}{\widehat{\eta}-1}$$

$$\frac{\Psi''(\widehat{\eta})}{\Psi'(\widehat{\eta})} = \frac{\left(g^{-1}\right)''\left(\frac{\widehat{\eta}}{1-\widehat{\eta}}\right)}{\left(g^{-1}\right)'\left(\frac{\widehat{\eta}}{1-\widehat{\eta}}\right)(\widehat{\eta}-1)^2} - \frac{2}{\widehat{\eta}-1}.$$

It holds that $\Psi^{-1}(z) = \frac{g^{-1}(z)}{g^{-1}(z)+1}$ for $g : \mathbb{R} \to \mathbb{R}$ being strictly monotonically increasing. That is, $\left(\Psi^{-1}\right)'(z) > 0$ and Theorem 4 grants convexity of losses if and only if

$$-\frac{1}{\widehat{\eta}} \leq \frac{\phi'''\left(\frac{\widehat{\eta}}{1-\widehat{\eta}}\right)}{\phi''\left(\frac{\widehat{\eta}}{1-\widehat{\eta}}\right)(\widehat{\eta}-1)^2} - \frac{\left(g^{-1}\right)''\left(\frac{\widehat{\eta}}{1-\widehat{\eta}}\right)}{\left(g^{-1}\right)'\left(\frac{\widehat{\eta}}{1-\widehat{\eta}}\right)(\widehat{\eta}-1)^2} - \frac{1}{\widehat{\eta}-1} \leq \frac{1}{1-\widehat{\eta}}, \quad \forall \widehat{\eta} \in (0,1).$$

Substituting $x := \frac{\widehat{\eta}}{1-\widehat{\eta}}$ such that $\widehat{\eta} = \frac{x}{1+x}$, $-\frac{1}{\widehat{\eta}} = -\frac{1+x}{x}$ and $-\frac{1}{\widehat{\eta}-1} = 1+x$ gives

$$-\frac{1+x}{x} \leq \frac{\phi'''(x)}{\phi''(x)}(1+x)^2 - \frac{\left(g^{-1}\right)''(x)}{\left(g^{-1}\right)'(x)}(1+x)^2 + (1+x) \leq 1+x, \quad \forall x \in [0,\infty)$$

and consequently

$$-\frac{(1+x)^2}{x} \leq \frac{\phi'''(x)}{\phi''(x)}(1+x)^2 - \frac{\left(g^{-1}\right)''(x)}{\left(g^{-1}\right)'(x)}(1+x)^2 \leq 0, \quad \forall x \in [0,\infty).$$

$\square$

## C  DETAILS ON EMPIRICAL EVALUATIONS

The computation of the results for the domain adaptation benchmark experiment is based on gradient-based training of overall 9174 models which we built following Gruber et al. (2024) which uses the code-base of Dinu et al. (2023). We obtain the number of trained models as follows.

AmazonReviews (text data): 11 methods $\times$ 14 parameters $\times$ 12 domain adaptation tasks $\times$ 3 seeds = 5544 trained models

MiniDomainNet (image data): 11 methods $\times$ 8 parameters $\times$ 5 domain adaptation tasks $\times$ 3 seeds = 1320 trained models

HHAR (sensory data): 11 methods $\times$ 14 parameters $\times$ 5 domain adaptation tasks $\times$ 3 seeds = 2310 trained models

Following Dinu et al. (2023) we used 11 domain adaptation methods from the AdaTime (Ragab et al., 2023) benchmark. For each of these methods and domain adaptation modalities (text, image, sensory) we evaluate 4 density ratio estimators (KuLSIF, LR, Boost and EW). For our experiments we follow the model implementations and experimental setup of Dinu et al. (2023) which results in the usage of fully connected networks for Amazon Reviews and a pretrained ResNet-18 backbone for MiniDomainNet.

For training and selecting the density ratio estimation methods within this pipeline we follow Gruber et al. (2024) perform an additional train/val split of $80/20$ on the datasets that are used for training the domain adaption methods. The regularization parameter $\lambda$ is selected from $\{10, 10^{-1}, 10^{-3}\}$ and the number of iterations of the BFGS algorithm is fixed with 100. We follow Kanamori et al. (2012) in using the Gaussian kernel with kernel width set according to the median heuristic for all compared density ratio estimation methods. Each experiment is run 3 times. To test the performance of the compared methods the classification accuracy on the respective test sets of the target distribution is evaluated. The rcond parameter for inverting the Gram matrix in (Dinu et al., 2023), we used a value of $10^{-3}$.

## C.1 AMAZON REVIEWS

| | SO | Importance Weighted Validation | | | | Importance Weighted Aggregation | | | | TB |
|---|---|---|---|---|---|---|---|---|---|---|
| | | Boost (IWV) | KuLSIF (IWV) | LR (IWV) | EW (IWV) | Boost (Agg) | KuLSIF (Agg) | LR (Agg) | EW (Agg) | |
| B→D | 0.784(±.004) | **0.773(±.008)** | 0.773(±.008) | 0.772(±.008) | 0.771(±.002) | 0.772(±.011) | 0.782(±.003) | 0.779(±.005) | **0.786(±.007)** | 0.790(±.018) |
| B→E | 0.754(±.009) | 0.751(±.008) | 0.752(±.007) | **0.754(±.011)** | 0.752(±.018) | 0.758(±.012) | 0.755(±.007) | 0.749(±.012) | **0.762(±.012)** | 0.756(±.023) |
| B→K | 0.769(±.021) | 0.768(±.016) | 0.770(±.011) | 0.770(±.017) | **0.771(±.012)** | 0.766(±.032) | 0.771(±.016) | 0.774(±.017) | **0.781(±.014)** | 0.774(±.014) |
| D→B | 0.779(±.004) | **0.783(±.004)** | 0.779(±.009) | 0.776(±.004) | 0.782(±.006) | 0.781(±.010) | 0.789(±.002) | 0.788(±.007) | **0.792(±.001)** | 0.789(±.007) |
| D→E | 0.766(±.013) | 0.772(±.006) | 0.766(±.011) | 0.766(±.011) | **0.772(±.004)** | 0.773(±.005) | 0.776(±.008) | 0.772(±.012) | **0.780(±.006)** | 0.776(±.005) |
| D→K | 0.777(±.015) | 0.783(±.009) | 0.780(±.010) | 0.782(±.008) | **0.789(±.015)** | 0.789(±.007) | 0.788(±.012) | 0.786(±.009) | **0.792(±.011)** | 0.786(±.009) |
| E→B | 0.688(±.015) | 0.694(±.021) | 0.695(±.019) | 0.694(±.021) | **0.701(±.009)** | 0.696(±.020) | 0.703(±.020) | 0.691(±.019) | **0.714(±.020)** | 0.707(±.017) |
| E→D | 0.714(±.005) | 0.717(±.004) | 0.717(±.004) | 0.725(±.017) | **0.735(±.008)** | 0.715(±.021) | 0.726(±.007) | 0.722(±.013) | **0.732(±.011)** | 0.737(±.011) |
| E→K | 0.858(±.004) | 0.859(±.010) | 0.859(±.010) | 0.863(±.014) | **0.865(±.007)** | 0.861(±.007) | 0.866(±.013) | 0.869(±.012) | **0.873(±.012)** | 0.865(±.008) |
| K→B | 0.713(±.002) | **0.710(±.009)** | 0.710(±.009) | 0.710(±.009) | 0.708(±.014) | 0.695(±.009) | 0.722(±.006) | 0.708(±.014) | **0.722(±.004)** | 0.718(±.007) |
| K→D | 0.741(±.010) | 0.728(±.015) | **0.753(±.020)** | 0.742(±.033) | 0.748(±.021) | 0.732(±.014) | **0.752(±.006)** | 0.741(±.014) | 0.751(±.016) | 0.756(±.014) |
| K→E | 0.848(±.008) | **0.851(±.005)** | 0.848(±.003) | 0.850(±.007) | 0.846(±.003) | 0.849(±.012) | 0.849(±.010) | 0.847(±.011) | **0.856(±.007)** | 0.850(±.002) |
| mean | 0.766(±.009) | 0.766(±.010) | 0.767(±.010) | 0.767(±.013) | **0.770(±.010)** | 0.766(±.013) | 0.773(±.009) | 0.769(±.012) | **0.778(±.010)** | 0.775(±.011) |

Table 2: AdvSKM

| | SO | Importance Weighted Validation | | | | Importance Weighted Aggregation | | | | TB |
|---|---|---|---|---|---|---|---|---|---|---|
| | | Boost (IWV) | KuLSIF (IWV) | LR (IWV) | EW (IWV) | Boost (Agg) | KuLSIF (Agg) | LR (Agg) | EW (Agg) | |
| B→D | 0.784(±.022) | 0.786(±.007) | **0.793(±.013)** | 0.786(±.007) | 0.782(±.016) | 0.794(±.015) | 0.793(±.016) | 0.785(±.007) | **0.797(±.010)** | 0.797(±.001) |
| B→E | 0.758(±.013) | **0.770(±.019)** | 0.759(±.009) | 0.758(±.010) | 0.762(±.007) | 0.772(±.014) | 0.777(±.008) | 0.766(±.017) | **0.782(±.009)** | 0.775(±.008) |
| B→K | 0.769(±.019) | 0.782(±.007) | 0.782(±.007) | 0.782(±.007) | **0.783(±.006)** | 0.777(±.042) | 0.787(±.026) | 0.788(±.027) | **0.792(±.012)** | 0.796(±.011) |
| D→B | 0.786(±.010) | 0.785(±.006) | 0.785(±.006) | 0.785(±.006) | **0.787(±.007)** | 0.780(±.008) | 0.791(±.010) | 0.787(±.011) | **0.791(±.005)** | 0.794(±.001) |
| D→E | 0.761(±.003) | 0.779(±.015) | 0.775(±.007) | **0.786(±.023)** | 0.773(±.004) | 0.788(±.012) | 0.790(±.007) | 0.791(±.010) | **0.803(±.007)** | 0.794(±.008) |
| D→K | 0.778(±.018) | 0.785(±.013) | 0.785(±.013) | 0.789(±.007) | **0.794(±.015)** | 0.805(±.012) | 0.801(±.016) | 0.799(±.013) | **0.807(±.014)** | 0.801(±.010) |
| E→B | 0.692(±.014) | 0.695(±.029) | **0.702(±.022)** | 0.690(±.024) | 0.693(±.013) | 0.706(±.020) | 0.705(±.023) | 0.698(±.022) | **0.719(±.031)** | 0.711(±.029) |
| E→D | 0.720(±.018) | 0.723(±.015) | 0.728(±.012) | **0.728(±.012)** | 0.718(±.006) | 0.683(±.061) | 0.726(±.019) | 0.705(±.045) | **0.741(±.003)** | 0.735(±.013) |
| E→K | 0.860(±.006) | 0.857(±.013) | 0.857(±.013) | 0.854(±.010) | **0.860(±.008)** | 0.859(±.025) | 0.868(±.014) | 0.861(±.020) | **0.875(±.012)** | 0.865(±.011) |
| K→B | 0.706(±.004) | **0.721(±.005)** | 0.718(±.009) | 0.721(±.005) | 0.719(±.011) | 0.697(±.050) | 0.728(±.009) | 0.719(±.020) | **0.733(±.013)** | 0.724(±.013) |
| K→D | 0.748(±.013) | **0.755(±.018)** | 0.755(±.018) | 0.746(±.013) | 0.752(±.008) | 0.735(±.022) | 0.757(±.018) | 0.757(±.016) | **0.772(±.012)** | 0.752(±.018) |
| K→E | 0.845(±.007) | 0.848(±.002) | 0.843(±.006) | 0.843(±.006) | **0.849(±.007)** | 0.825(±.017) | 0.853(±.006) | 0.842(±.023) | **0.861(±.005)** | 0.850(±.001) |
| mean | 0.767(±.012) | **0.774(±.012)** | 0.773(±.011) | 0.772(±.011) | 0.773(±.009) | 0.768(±.025) | 0.782(±.014) | 0.775(±.019) | **0.789(±.011)** | 0.783(±.010) |

Table 3: CDAN

| | SO | Importance Weighted Validation | | | | Importance Weighted Aggregation | | | | TB |
|---|---|---|---|---|---|---|---|---|---|---|
| | | Boost (IWV) | KuLSIF (IWV) | LR (IWV) | EW (IWV) | Boost (Agg) | KuLSIF (Agg) | LR (Agg) | EW (Agg) | |
| B→D | 0.772(±.009) | **0.789(±.025)** | 0.789(±.025) | 0.789(±.025) | 0.772(±.012) | 0.791(±.017) | **0.797(±.013)** | 0.784(±.035) | 0.794(±.008) | 0.789(±.018) |
| B→E | 0.745(±.012) | 0.761(±.025) | 0.759(±.021) | 0.761(±.025) | **0.770(±.005)** | 0.739(±.028) | 0.759(±.020) | 0.756(±.018) | **0.780(±.009)** | 0.780(±.009) |
| B→K | 0.763(±.015) | 0.754(±.048) | 0.778(±.012) | 0.758(±.052) | **0.787(±.013)** | 0.743(±.085) | 0.792(±.011) | 0.790(±.012) | **0.796(±.012)** | 0.793(±.009) |
| D→B | 0.788(±.010) | 0.784(±.007) | 0.793(±.007) | 0.783(±.007) | **0.793(±.007)** | 0.715(±.115) | 0.788(±.002) | 0.786(±.010) | **0.807(±.007)** | 0.794(±.006) |
| D→E | 0.768(±.003) | 0.776(±.016) | 0.774(±.016) | 0.783(±.008) | **0.784(±.022)** | 0.794(±.016) | 0.792(±.009) | 0.795(±.013) | **0.806(±.011)** | 0.798(±.016) |
| D→K | 0.777(±.012) | 0.787(±.011) | 0.777(±.012) | 0.797(±.027) | **0.802(±.019)** | 0.757(±.049) | 0.792(±.014) | 0.799(±.016) | **0.804(±.008)** | 0.811(±.007) |
| E→B | 0.699(±.014) | 0.700(±.010) | 0.705(±.009) | 0.698(±.014) | **0.713(±.014)** | 0.685(±.014) | **0.727(±.014)** | 0.691(±.013) | 0.726(±.015) | 0.712(±.011) |
| E→D | 0.722(±.008) | **0.742(±.010)** | 0.733(±.020) | 0.742(±.010) | 0.742(±.010) | 0.713(±.021) | 0.735(±.009) | 0.724(±.015) | **0.742(±.004)** | 0.738(±.007) |
| E→K | 0.860(±.012) | **0.872(±.013)** | 0.859(±.005) | 0.867(±.012) | 0.860(±.006) | 0.857(±.012) | 0.867(±.020) | 0.872(±.016) | **0.875(±.012)** | 0.871(±.014) |
| K→B | 0.718(±.005) | 0.725(±.011) | 0.733(±.004) | **0.734(±.006)** | 0.713(±.005) | 0.646(±.068) | 0.727(±.018) | 0.698(±.042) | **0.734(±.011)** | 0.740(±.007) |
| K→D | 0.748(±.003) | 0.739(±.010) | 0.739(±.010) | 0.727(±.026) | **0.755(±.015)** | 0.694(±.081) | 0.753(±.016) | 0.723(±.061) | **0.762(±.023)** | 0.761(±.026) |
| K→E | 0.842(±.005) | 0.847(±.016) | 0.847(±.016) | 0.844(±.025) | **0.855(±.007)** | 0.812(±.051) | 0.854(±.004) | 0.851(±.004) | **0.867(±.004)** | 0.858(±.005) |
| mean | 0.767(±.009) | 0.773(±.017) | 0.774(±.013) | 0.774(±.019) | **0.779(±.011)** | 0.745(±.046) | 0.782(±.013) | 0.772(±.021) | **0.791(±.010)** | 0.787(±.011) |

Table 4: CMD

| | SO | Importance Weighted Validation | | | | Importance Weighted Aggregation | | | | TB |
|---|---|---|---|---|---|---|---|---|---|---|
| | | Boost (IWV) | KuLSIF (IWV) | LR (IWV) | EW (IWV) | Boost (Agg) | KuLSIF (Agg) | LR (Agg) | EW (Agg) | |
| B→D | 0.783(±.013) | **0.793(±.014)** | 0.793(±.014) | 0.793(±.014) | 0.788(±.008) | 0.794(±.013) | 0.799(±.001) | 0.803(±.006) | **0.806(±.006)** | 0.801(±.014) |
| B→E | 0.755(±.005) | 0.766(±.009) | 0.766(±.009) | **0.780(±.018)** | 0.764(±.019) | 0.763(±.024) | 0.774(±.020) | 0.773(±.020) | **0.785(±.007)** | 0.802(±.009) |
| B→K | 0.771(±.022) | **0.817(±.008)** | 0.794(±.015) | 0.803(±.008) | 0.798(±.011) | 0.793(±.021) | **0.811(±.001)** | 0.807(±.008) | 0.807(±.005) | 0.817(±.007) |
| D→B | 0.774(±.001) | 0.777(±.006) | 0.777(±.006) | 0.781(±.011) | **0.791(±.011)** | 0.785(±.021) | 0.799(±.008) | 0.795(±.014) | **0.806(±.009)** | 0.800(±.011) |
| D→E | 0.769(±.003) | 0.806(±.022) | 0.788(±.015) | 0.806(±.022) | **0.811(±.018)** | 0.785(±.019) | 0.788(±.010) | 0.784(±.009) | **0.807(±.010)** | 0.817(±.011) |
| D→K | 0.782(±.021) | 0.806(±.041) | 0.806(±.041) | 0.806(±.041) | **0.823(±.019)** | 0.795(±.020) | 0.808(±.018) | 0.811(±.016) | **0.824(±.012)** | 0.828(±.010) |
| E→B | 0.687(±.013) | 0.722(±.045) | **0.732(±.022)** | 0.722(±.045) | 0.718(±.023) | 0.713(±.049) | **0.725(±.028)** | 0.720(±.045) | 0.721(±.028) | 0.723(±.033) |
| E→D | 0.720(±.013) | 0.721(±.020) | **0.725(±.026)** | 0.725(±.026) | 0.725(±.026) | 0.732(±.010) | 0.750(±.027) | 0.741(±.027) | **0.751(±.020)** | 0.736(±.012) |
| E→K | 0.859(±.016) | 0.862(±.014) | **0.867(±.002)** | 0.860(±.010) | 0.865(±.010) | 0.837(±.035) | 0.868(±.011) | 0.864(±.022) | **0.879(±.008)** | 0.870(±.012) |
| K→B | 0.712(±.012) | 0.731(±.022) | **0.732(±.042)** | 0.703(±.016) | 0.722(±.043) | 0.735(±.005) | 0.739(±.005) | 0.721(±.023) | **0.745(±.005)** | 0.751(±.016) |
| K→D | 0.730(±.022) | 0.774(±.005) | **0.774(±.005)** | 0.748(±.024) | 0.749(±.017) | 0.741(±.015) | 0.754(±.010) | 0.736(±.021) | **0.762(±.008)** | 0.764(±.010) |
| K→E | 0.845(±.000) | 0.849(±.009) | 0.847(±.009) | 0.846(±.008) | **0.853(±.002)** | 0.855(±.004) | 0.862(±.004) | 0.858(±.009) | **0.868(±.008)** | 0.856(±.003) |
| mean | 0.766(±.012) | **0.785(±.018)** | 0.783(±.017) | 0.781(±.020) | 0.784(±.017) | 0.777(±.020) | 0.790(±.012) | 0.784(±.018) | **0.797(±.010)** | 0.797(±.012) |

Table 5: CoDATS

| | SO | Importance Weighted Validation | | | | Importance Weighted Aggregation | | | | TB |
|---|---|---|---|---|---|---|---|---|---|---|
| | | Boost (IWV) | KuLSIF (IWV) | LR (IWV) | EW (IWV) | Boost (Agg) | KuLSIF (Agg) | LR (Agg) | EW (Agg) | |
| B→D | 0.783(±.004) | 0.789(±.020) | 0.798(±.009) | 0.789(±.020) | **0.799(±.008)** | 0.787(±.015) | 0.795(±.004) | **0.801(±.009)** | 0.801(±.007) | 0.813(±.017) |
| B→E | 0.752(±.011) | 0.770(±.018) | 0.770(±.018) | 0.772(±.016) | **0.800(±.005)** | 0.739(±.044) | 0.769(±.010) | 0.747(±.022) | **0.790(±.004)** | 0.800(±.005) |
| B→K | 0.767(±.022) | 0.791(±.020) | 0.784(±.013) | **0.795(±.001)** | 0.784(±.029) | 0.779(±.029) | 0.798(±.013) | 0.785(±.018) | **0.804(±.007)** | 0.820(±.004) |
| D→B | 0.779(±.002) | 0.786(±.012) | 0.784(±.015) | 0.787(±.007) | **0.798(±.001)** | 0.785(±.010) | 0.787(±.006) | 0.772(±.028) | **0.806(±.006)** | 0.799(±.001) |
| D→E | 0.767(±.016) | **0.802(±.009)** | 0.794(±.009) | 0.794(±.009) | 0.789(±.006) | 0.800(±.022) | 0.806(±.007) | 0.795(±.017) | **0.814(±.006)** | 0.815(±.007) |
| D→K | 0.784(±.013) | 0.805(±.007) | 0.800(±.002) | 0.793(±.024) | **0.807(±.007)** | 0.774(±.011) | 0.806(±.007) | 0.780(±.028) | **0.823(±.007)** | 0.827(±.005) |
| E→B | 0.701(±.019) | 0.715(±.035) | 0.714(±.035) | 0.714(±.035) | **0.719(±.030)** | 0.719(±.018) | 0.720(±.025) | 0.713(±.036) | **0.723(±.014)** | 0.721(±.010) |
| E→D | 0.736(±.005) | 0.727(±.003) | 0.723(±.018) | 0.728(±.002) | **0.736(±.008)** | 0.718(±.025) | 0.747(±.014) | 0.741(±.012) | **0.749(±.003)** | 0.757(±.032) |
| E→K | 0.854(±.016) | 0.855(±.019) | 0.862(±.009) | 0.853(±.016) | **0.866(±.013)** | 0.856(±.012) | 0.868(±.012) | 0.867(±.018) | **0.878(±.014)** | 0.875(±.008) |
| K→B | 0.711(±.014) | **0.742(±.017)** | 0.742(±.017) | 0.721(±.004) | 0.729(±.004) | 0.720(±.030) | 0.751(±.011) | 0.748(±.005) | **0.752(±.007)** | 0.747(±.021) |
| K→D | 0.738(±.006) | 0.729(±.018) | 0.747(±.049) | 0.735(±.013) | **0.751(±.018)** | 0.741(±.050) | 0.772(±.021) | 0.754(±.044) | **0.786(±.013)** | 0.762(±.021) |
| K→E | 0.837(±.013) | 0.853(±.004) | **0.853(±.004)** | 0.850(±.007) | 0.849(±.005) | 0.848(±.016) | 0.854(±.003) | 0.849(±.005) | **0.866(±.007)** | 0.859(±.013) |
| mean | 0.767(±.012) | 0.780(±.015) | 0.781(±.017) | 0.777(±.013) | **0.785(±.011)** | 0.772(±.024) | 0.789(±.011) | 0.779(±.020) | **0.799(±.008)** | 0.800(±.012) |

Table 6: DANN

| | SO | Importance Weighted Validation | | | | Importance Weighted Aggregation | | | | TB |
|---|---|---|---|---|---|---|---|---|---|---|
| | | Boost (IWV) | KuLSIF (IWV) | LR (IWV) | EW (IWV) | Boost (Agg) | KuLSIF (Agg) | LR (Agg) | EW (Agg) | |
| B→D | 0.781(±.016) | 0.776(±.008) | 0.780(±.009) | 0.779(±.009) | **0.782(±.005)** | 0.775(±.015) | 0.783(±.004) | 0.785(±.009) | **0.790(±.005)** | 0.786(±.004) |
| B→E | 0.752(±.007) | 0.754(±.008) | 0.754(±.007) | **0.754(±.008)** | 0.753(±.012) | 0.741(±.013) | 0.753(±.010) | 0.746(±.015) | **0.766(±.013)** | 0.756(±.012) |
| B→K | 0.766(±.015) | 0.773(±.019) | 0.773(±.019) | 0.773(±.019) | **0.774(±.018)** | 0.767(±.026) | 0.777(±.026) | 0.772(±.032) | **0.782(±.017)** | 0.780(±.014) |
| D→B | 0.781(±.008) | 0.783(±.007) | **0.786(±.011)** | 0.783(±.007) | 0.779(±.009) | 0.776(±.023) | 0.782(±.016) | 0.779(±.016) | **0.790(±.011)** | 0.788(±.009) |
| D→E | 0.767(±.015) | 0.772(±.006) | 0.774(±.011) | 0.772(±.006) | **0.776(±.004)** | 0.771(±.014) | 0.767(±.009) | 0.770(±.018) | **0.784(±.002)** | 0.778(±.010) |
| D→K | 0.782(±.013) | **0.784(±.010)** | 0.782(±.013) | 0.782(±.013) | 0.783(±.014) | 0.771(±.031) | 0.781(±.018) | 0.781(±.013) | **0.793(±.017)** | 0.788(±.008) |
| E→B | 0.693(±.019) | 0.685(±.014) | 0.685(±.014) | 0.685(±.014) | **0.691(±.019)** | **0.708(±.018)** | 0.705(±.027) | 0.702(±.011) | 0.702(±.023) | 0.704(±.018) |
| E→D | 0.725(±.006) | 0.737(±.028) | 0.727(±.017) | **0.740(±.024)** | 0.736(±.026) | 0.737(±.024) | **0.742(±.007)** | 0.739(±.009) | 0.737(±.011) | 0.736(±.028) |
| E→K | 0.857(±.011) | 0.860(±.001) | **0.860(±.002)** | 0.860(±.002) | 0.860(±.008) | 0.841(±.034) | 0.857(±.016) | 0.854(±.010) | **0.873(±.009)** | 0.864(±.009) |
| K→B | 0.713(±.002) | **0.716(±.004)** | 0.716(±.004) | 0.713(±.015) | 0.707(±.006) | 0.706(±.023) | 0.719(±.011) | 0.709(±.010) | **0.722(±.004)** | 0.719(±.005) |
| K→D | 0.741(±.018) | 0.745(±.025) | **0.754(±.016)** | 0.751(±.016) | 0.739(±.016) | 0.737(±.019) | 0.747(±.018) | 0.744(±.014) | **0.751(±.011)** | 0.748(±.019) |
| K→E | 0.841(±.001) | **0.849(±.010)** | 0.844(±.004) | 0.849(±.010) | 0.846(±.005) | 0.845(±.018) | 0.854(±.005) | 0.843(±.014) | **0.857(±.006)** | 0.849(±.011) |
| mean | 0.766(±.011) | 0.769(±.012) | 0.770(±.010) | **0.770(±.011)** | 0.769(±.012) | 0.765(±.022) | 0.772(±.014) | 0.769(±.015) | **0.779(±.011)** | 0.775(±.012) |

Table 7: DDC

| | SO | Importance Weighted Validation | | | | Importance Weighted Aggregation | | | | TB |
|---|---|---|---|---|---|---|---|---|---|---|
| | | Boost (IWV) | KuLSIF (IWV) | LR (IWV) | EW (IWV) | Boost (Agg) | KuLSIF (Agg) | LR (Agg) | EW (Agg) | |
| B→D | 0.778(±.018) | 0.790(±.011) | **0.790(±.011)** | 0.789(±.014) | 0.781(±.009) | 0.786(±.008) | 0.796(±.005) | 0.798(±.009) | **0.799(±.009)** | 0.794(±.020) |
| B→E | 0.749(±.013) | 0.755(±.015) | 0.755(±.015) | 0.755(±.015) | **0.761(±.011)** | 0.745(±.008) | 0.754(±.013) | 0.750(±.021) | **0.762(±.014)** | 0.767(±.006) |
| B→K | 0.769(±.011) | 0.775(±.015) | **0.776(±.012)** | 0.774(±.012) | 0.773(±.021) | 0.768(±.028) | **0.783(±.018)** | 0.771(±.030) | 0.782(±.015) | 0.779(±.016) |
| D→B | 0.783(±.009) | 0.780(±.003) | 0.777(±.002) | **0.780(±.003)** | 0.780(±.012) | 0.775(±.025) | 0.791(±.002) | 0.782(±.009) | **0.794(±.003)** | 0.786(±.004) |
| D→E | 0.760(±.013) | 0.762(±.014) | 0.771(±.011) | 0.762(±.014) | **0.775(±.008)** | 0.767(±.021) | 0.771(±.015) | 0.768(±.015) | **0.784(±.003)** | 0.785(±.005) |
| D→K | 0.780(±.016) | 0.780(±.016) | 0.779(±.015) | 0.780(±.016) | **0.789(±.017)** | 0.761(±.054) | 0.782(±.027) | 0.774(±.023) | **0.797(±.012)** | 0.789(±.017) |
| E→B | 0.696(±.009) | 0.695(±.008) | **0.702(±.020)** | 0.696(±.009) | 0.702(±.020) | 0.710(±.017) | **0.711(±.021)** | 0.700(±.013) | 0.709(±.021) | 0.708(±.009) |
| E→D | 0.722(±.009) | 0.731(±.008) | 0.726(±.015) | **0.734(±.005)** | 0.729(±.006) | 0.735(±.008) | **0.745(±.004)** | 0.744(±.005) | 0.740(±.003) | 0.739(±.006) |
| E→K | 0.859(±.010) | 0.859(±.010) | 0.859(±.010) | 0.859(±.010) | **0.861(±.009)** | 0.841(±.044) | 0.870(±.012) | 0.867(±.015) | **0.874(±.008)** | 0.864(±.005) |
| K→B | 0.720(±.002) | 0.721(±.008) | **0.721(±.008)** | 0.721(±.008) | 0.717(±.001) | 0.729(±.006) | 0.731(±.011) | 0.725(±.014) | **0.737(±.006)** | 0.740(±.005) |
| K→D | 0.733(±.011) | 0.734(±.032) | **0.743(±.018)** | 0.735(±.020) | 0.738(±.023) | 0.755(±.005) | 0.750(±.018) | 0.751(±.012) | **0.760(±.010)** | 0.751(±.008) |
| K→E | 0.841(±.010) | 0.844(±.006) | 0.844(±.006) | 0.844(±.006) | **0.846(±.006)** | 0.841(±.020) | 0.854(±.002) | 0.848(±.007) | **0.859(±.008)** | 0.849(±.009) |
| mean | 0.766(±.012) | 0.769(±.012) | 0.770(±.012) | 0.769(±.011) | **0.771(±.012)** | 0.768(±.020) | 0.778(±.012) | 0.773(±.014) | **0.783(±.010)** | 0.779(±.009) |

Table 8: DCoral

| | SO | Importance Weighted Validation | | | | Importance Weighted Aggregation | | | | TB |
|---|---|---|---|---|---|---|---|---|---|---|
| | | Boost (IWV) | KuLSIF (IWV) | LR (IWV) | EW (IWV) | Boost (Agg) | KuLSIF (Agg) | LR (Agg) | EW (Agg) | |
| B→D | 0.777(±.010) | 0.796(±.014) | 0.796(±.014) | 0.796(±.014) | **0.804(±.009)** | 0.816(±.003) | **0.819(±.001)** | 0.819(±.007) | 0.805(±.008) | 0.809(±.010) |
| B→E | 0.756(±.013) | **0.779(±.022)** | 0.765(±.010) | 0.748(±.014) | 0.753(±.010) | 0.770(±.007) | 0.777(±.004) | 0.757(±.018) | **0.777(±.012)** | 0.804(±.005) |
| B→K | 0.770(±.013) | 0.772(±.031) | 0.796(±.020) | 0.789(±.012) | **0.796(±.020)** | 0.745(±.049) | 0.798(±.005) | 0.784(±.017) | **0.802(±.009)** | 0.814(±.006) |
| D→B | 0.773(±.008) | 0.799(±.013) | 0.810(±.017) | **0.813(±.015)** | 0.811(±.016) | 0.767(±.042) | 0.796(±.019) | 0.780(±.020) | **0.808(±.006)** | 0.811(±.017) |
| D→E | 0.757(±.020) | 0.768(±.032) | 0.768(±.032) | 0.768(±.032) | **0.778(±.024)** | 0.776(±.038) | 0.782(±.030) | 0.778(±.028) | **0.801(±.008)** | 0.812(±.006) |
| D→K | 0.775(±.016) | 0.820(±.015) | **0.820(±.015)** | 0.820(±.015) | 0.814(±.021) | 0.822(±.008) | 0.826(±.002) | 0.814(±.007) | **0.828(±.010)** | 0.828(±.010) |
| E→B | 0.702(±.018) | 0.689(±.025) | **0.700(±.011)** | 0.689(±.025) | 0.697(±.011) | 0.691(±.014) | 0.698(±.017) | 0.687(±.013) | **0.708(±.021)** | 0.708(±.042) |
| E→D | 0.716(±.008) | 0.715(±.014) | 0.715(±.014) | 0.715(±.014) | **0.718(±.010)** | 0.704(±.010) | 0.721(±.010) | 0.698(±.010) | **0.722(±.016)** | 0.730(±.001) |
| E→K | 0.855(±.013) | 0.871(±.020) | 0.872(±.020) | 0.871(±.020) | **0.879(±.019)** | 0.848(±.037) | 0.868(±.018) | 0.859(±.031) | **0.889(±.005)** | 0.889(±.004) |
| K→B | 0.709(±.012) | 0.689(±.037) | 0.719(±.016) | 0.719(±.016) | **0.735(±.020)** | 0.713(±.018) | 0.735(±.009) | 0.726(±.027) | **0.745(±.009)** | 0.723(±.007) |
| K→D | 0.736(±.012) | **0.750(±.011)** | 0.749(±.009) | 0.741(±.005) | 0.748(±.022) | 0.704(±.028) | 0.745(±.005) | 0.728(±.009) | **0.765(±.000)** | 0.756(±.007) |
| K→E | 0.842(±.009) | 0.857(±.012) | 0.855(±.015) | 0.853(±.019) | **0.861(±.019)** | 0.854(±.008) | 0.865(±.000) | 0.857(±.003) | **0.873(±.005)** | 0.863(±.013) |
| mean | 0.764(±.013) | 0.775(±.020) | 0.780(±.016) | 0.777(±.017) | **0.783(±.017)** | 0.767(±.022) | 0.786(±.010) | 0.774(±.016) | **0.794(±.009)** | 0.795(±.011) |

Table 9: DIRT

|  | SO | Importance Weighted Validation | | | | Importance Weighted Aggregation | | | | TB |
|---|---|---|---|---|---|---|---|---|---|---|
|  |  | Boost (IWV) | KuLSIF (IWV) | LR (IWV) | EW (IWV) | Boost (Agg) | KuLSIF (Agg) | LR (Agg) | EW (Agg) |  |
| B→D | 0.779(±.013) | 0.791(±.007) | 0.794(±.008) | 0.791(±.008) | **0.798(±.013)** | 0.785(±.017) | 0.795(±.008) | 0.794(±.008) | **0.805(±.004)** | 0.805(±.006) |
| B→E | 0.752(±.005) | 0.766(±.027) | 0.761(±.019) | 0.766(±.027) | **0.785(±.002)** | 0.780(±.012) | 0.782(±.023) | 0.780(±.027) | **0.797(±.005)** | 0.796(±.012) |
| B→K | 0.768(±.012) | 0.778(±.033) | **0.794(±.006)** | 0.786(±.031) | 0.783(±.027) | 0.795(±.028) | 0.806(±.013) | 0.798(±.036) | **0.811(±.010)** | 0.816(±.005) |
| D→B | 0.782(±.008) | 0.784(±.005) | 0.789(±.009) | 0.784(±.005) | **0.790(±.006)** | 0.782(±.027) | 0.795(±.007) | 0.793(±.009) | **0.798(±.008)** | 0.802(±.007) |
| D→E | 0.771(±.002) | 0.788(±.018) | 0.798(±.014) | 0.802(±.009) | **0.804(±.009)** | 0.796(±.011) | 0.795(±.009) | 0.796(±.005) | **0.808(±.003)** | 0.813(±.002) |
| D→K | 0.786(±.013) | 0.800(±.002) | 0.793(±.012) | 0.783(±.017) | **0.808(±.015)** | 0.657(±.273) | 0.790(±.038) | 0.788(±.033) | **0.820(±.014)** | 0.830(±.016) |
| E→B | 0.702(±.021) | **0.710(±.010)** | 0.704(±.019) | 0.706(±.007) | 0.706(±.008) | 0.713(±.022) | **0.724(±.018)** | 0.717(±.024) | 0.719(±.015) | 0.711(±.010) |
| E→D | 0.725(±.007) | **0.736(±.015)** | 0.732(±.019) | 0.725(±.026) | 0.731(±.019) | 0.743(±.018) | **0.744(±.014)** | 0.744(±.014) | 0.744(±.011) | 0.732(±.019) |
| E→K | 0.865(±.013) | 0.860(±.005) | 0.857(±.011) | 0.857(±.011) | **0.876(±.006)** | 0.875(±.006) | 0.874(±.008) | 0.871(±.007) | **0.885(±.009)** | 0.876(±.006) |
| K→B | 0.716(±.016) | 0.729(±.020) | 0.712(±.014) | **0.730(±.020)** | 0.720(±.021) | 0.710(±.000) | 0.729(±.012) | 0.706(±.033) | **0.748(±.004)** | 0.739(±.022) |
| K→D | 0.739(±.023) | 0.738(±.016) | **0.744(±.017)** | 0.744(±.017) | 0.743(±.026) | 0.741(±.027) | 0.765(±.019) | 0.744(±.015) | **0.775(±.013)** | 0.763(±.026) |
| K→E | 0.840(±.010) | 0.853(±.007) | 0.854(±.007) | 0.848(±.011) | **0.855(±.009)** | 0.852(±.005) | 0.858(±.004) | 0.848(±.005) | **0.869(±.005)** | 0.857(±.004) |
| mean | 0.769(±.012) | 0.778(±.014) | 0.778(±.013) | 0.777(±.016) | **0.783(±.013)** | 0.769(±.037) | 0.788(±.014) | 0.782(±.017) | **0.798(±.008)** | 0.795(±.011) |

Table 10: DSAN

|  | SO | Importance Weighted Validation | | | | Importance Weighted Aggregation | | | | TB |
|---|---|---|---|---|---|---|---|---|---|---|
|  |  | Boost (IWV) | KuLSIF (IWV) | LR (IWV) | EW (IWV) | Boost (Agg) | KuLSIF (Agg) | LR (Agg) | EW (Agg) |  |
| B→D | 0.778(±.010) | 0.778(±.011) | **0.781(±.015)** | 0.780(±.013) | 0.781(±.012) | 0.779(±.012) | **0.793(±.013)** | 0.786(±.013) | 0.792(±.009) | 0.789(±.009) |
| B→E | 0.745(±.014) | **0.753(±.018)** | 0.749(±.017) | 0.749(±.017) | 0.735(±.014) | 0.747(±.010) | 0.750(±.024) | 0.750(±.022) | **0.761(±.012)** | 0.758(±.017) |
| B→K | 0.770(±.018) | 0.766(±.017) | 0.760(±.012) | 0.769(±.022) | **0.774(±.014)** | 0.767(±.040) | 0.770(±.023) | 0.767(±.037) | **0.779(±.014)** | 0.776(±.016) |
| D→B | 0.785(±.006) | 0.769(±.009) | 0.778(±.016) | **0.781(±.011)** | 0.771(±.013) | 0.765(±.025) | 0.780(±.006) | 0.776(±.012) | **0.789(±.007)** | 0.787(±.007) |
| D→E | 0.772(±.012) | 0.767(±.004) | 0.759(±.011) | 0.768(±.006) | **0.768(±.005)** | 0.764(±.012) | 0.768(±.010) | 0.763(±.008) | **0.777(±.001)** | 0.774(±.004) |
| D→K | 0.783(±.015) | 0.781(±.005) | **0.788(±.012)** | 0.783(±.007) | 0.784(±.012) | 0.764(±.051) | 0.780(±.021) | 0.779(±.020) | **0.787(±.018)** | 0.786(±.013) |
| E→B | 0.702(±.026) | 0.687(±.017) | 0.686(±.017) | 0.686(±.016) | **0.696(±.020)** | 0.697(±.010) | 0.694(±.020) | 0.694(±.024) | **0.699(±.015)** | 0.703(±.030) |
| E→D | 0.733(±.003) | 0.723(±.013) | 0.732(±.009) | 0.725(±.009) | **0.735(±.013)** | 0.730(±.005) | 0.734(±.016) | 0.728(±.006) | **0.734(±.009)** | 0.735(±.015) |
| E→K | 0.862(±.009) | 0.855(±.008) | **0.862(±.013)** | 0.856(±.005) | 0.859(±.005) | 0.844(±.024) | 0.866(±.007) | 0.861(±.011) | **0.873(±.006)** | 0.863(±.004) |
| K→B | 0.715(±.008) | 0.711(±.003) | **0.712(±.010)** | 0.710(±.005) | 0.708(±.003) | 0.707(±.022) | 0.715(±.006) | 0.700(±.016) | **0.723(±.006)** | 0.717(±.005) |
| K→D | 0.745(±.011) | 0.745(±.003) | 0.741(±.005) | 0.741(±.005) | **0.746(±.009)** | 0.738(±.022) | **0.748(±.009)** | 0.738(±.005) | 0.748(±.002) | 0.745(±.011) |
| K→E | 0.844(±.009) | 0.848(±.008) | **0.849(±.008)** | 0.847(±.011) | 0.846(±.010) | 0.839(±.016) | 0.852(±.006) | 0.843(±.009) | **0.855(±.007)** | 0.849(±.007) |
| mean | 0.769(±.012) | 0.765(±.010) | 0.766(±.012) | 0.766(±.010) | **0.767(±.011)** | 0.762(±.021) | 0.771(±.013) | 0.765(±.015) | **0.776(±.009)** | 0.774(±.011) |

Table 11: HoMM

|  | SO | Importance Weighted Validation | | | | Importance Weighted Aggregation | | | | TB |
|---|---|---|---|---|---|---|---|---|---|---|
|  |  | Boost (IWV) | KuLSIF (IWV) | LR (IWV) | EW (IWV) | Boost (Agg) | KuLSIF (Agg) | LR (Agg) | EW (Agg) |  |
| B→D | 0.775(±.005) | 0.783(±.016) | 0.788(±.009) | 0.788(±.009) | **0.789(±.009)** | 0.776(±.014) | 0.786(±.002) | 0.783(±.008) | **0.800(±.001)** | 0.792(±.013) |
| B→E | 0.752(±.006) | **0.768(±.013)** | 0.758(±.008) | 0.754(±.011) | 0.758(±.011) | 0.754(±.032) | 0.766(±.024) | 0.758(±.037) | **0.771(±.019)** | 0.781(±.006) |
| B→K | 0.765(±.009) | 0.787(±.012) | **0.791(±.017)** | 0.783(±.010) | 0.784(±.021) | 0.796(±.011) | 0.793(±.014) | 0.794(±.019) | **0.797(±.013)** | 0.802(±.019) |
| D→B | 0.783(±.007) | 0.783(±.006) | **0.786(±.010)** | 0.779(±.014) | 0.777(±.009) | 0.769(±.034) | 0.787(±.004) | 0.784(±.008) | **0.795(±.003)** | 0.790(±.001) |
| D→E | 0.764(±.003) | 0.767(±.017) | 0.769(±.020) | 0.774(±.009) | **0.774(±.009)** | 0.771(±.018) | 0.781(±.016) | 0.788(±.008) | **0.789(±.004)** | 0.799(±.006) |
| D→K | 0.775(±.012) | 0.777(±.005) | 0.777(±.005) | 0.777(±.005) | **0.811(±.007)** | 0.785(±.023) | 0.789(±.012) | 0.787(±.015) | **0.810(±.006)** | 0.800(±.013) |
| E→B | 0.701(±.015) | **0.703(±.013)** | 0.698(±.005) | 0.698(±.005) | 0.699(±.009) | 0.707(±.008) | 0.711(±.008) | 0.692(±.018) | **0.719(±.013)** | 0.707(±.013) |
| E→D | 0.738(±.010) | 0.709(±.022) | **0.733(±.011)** | 0.732(±.014) | 0.729(±.008) | 0.732(±.017) | **0.741(±.019)** | 0.737(±.022) | 0.739(±.003) | 0.738(±.020) |
| E→K | 0.855(±.015) | 0.857(±.011) | 0.857(±.011) | 0.857(±.011) | **0.858(±.010)** | 0.855(±.009) | 0.869(±.003) | 0.864(±.008) | **0.878(±.005)** | 0.867(±.004) |
| K→B | 0.715(±.008) | 0.710(±.017) | 0.729(±.032) | 0.710(±.005) | **0.735(±.023)** | 0.726(±.009) | 0.726(±.020) | 0.717(±.005) | **0.743(±.010)** | 0.730(±.007) |
| K→D | 0.736(±.014) | 0.740(±.031) | 0.740(±.031) | 0.740(±.031) | **0.747(±.009)** | 0.737(±.023) | 0.747(±.016) | 0.735(±.042) | **0.766(±.007)** | 0.750(±.017) |
| K→E | 0.842(±.008) | 0.831(±.009) | 0.834(±.010) | 0.831(±.008) | **0.843(±.021)** | 0.823(±.024) | 0.839(±.006) | 0.823(±.021) | **0.858(±.009)** | 0.847(±.010) |
| mean | 0.767(±.009) | 0.768(±.014) | 0.772(±.014) | 0.769(±.012) | **0.775(±.012)** | 0.769(±.018) | 0.778(±.012) | 0.772(±.018) | **0.789(±.008)** | 0.784(±.011) |

Table 12: MMDA

## C.2 HHAR

|  | SO | Importance Weighted Validation | | | | Importance Weighted Aggregation | | | | TB |
|---|---|---|---|---|---|---|---|---|---|---|
|  |  | Boost (IWV) | KuLSIF (IWV) | LR (IWV) | EW (IWV) | Boost (Agg) | KuLSIF (Agg) | LR (Agg) | EW (Agg) |  |
| 0→6 | 0.661(±.035) | 0.692(±.019) | **0.696(±.019)** | 0.692(±.019) | 0.692(±.019) | 0.646(±.033) | 0.642(±.036) | **0.650(±.041)** | 0.604(±.046) | 0.731(±.013) |
| 1→6 | 0.821(±.046) | 0.806(±.026) | **0.806(±.026)** | 0.806(±.026) | 0.806(±.026) | 0.737(±.045) | 0.732(±.046) | **0.740(±.040)** | 0.735(±.049) | 0.838(±.014) |
| 2→7 | 0.455(±.070) | 0.497(±.067) | 0.482(±.062) | **0.497(±.067)** | 0.497(±.067) | 0.490(±.053) | 0.484(±.099) | 0.475(±.022) | **0.496(±.034)** | 0.574(±.112) |
| 3→8 | 0.790(±.011) | 0.799(±.005) | 0.794(±.008) | 0.799(±.005) | **0.799(±.005)** | 0.768(±.034) | **0.771(±.049)** | 0.768(±.037) | 0.770(±.026) | 0.818(±.012) |
| 4→5 | 0.862(±.055) | 0.850(±.025) | **0.852(±.010)** | 0.850(±.025) | 0.850(±.025) | 0.771(±.043) | 0.785(±.035) | 0.785(±.028) | **0.790(±.039)** | 0.895(±.014) |
| mean | 0.718(±.044) | 0.729(±.028) | 0.726(±.025) | **0.729(±.028)** | 0.729(±.028) | 0.682(±.042) | 0.683(±.053) | **0.684(±.034)** | 0.679(±.039) | 0.771(±.033) |

Table 13: AdvSKM

|  | SO | Importance Weighted Validation | | | | Importance Weighted Aggregation | | | | TB |
|---|---|---|---|---|---|---|---|---|---|---|
|  |  | Boost (IWV) | KuLSIF (IWV) | LR (IWV) | EW (IWV) | Boost (Agg) | KuLSIF (Agg) | LR (Agg) | EW (Agg) |  |
| 0→6 | 0.717(±.004) | **0.726(±.032)** | 0.643(±.159) | 0.726(±.032) | 0.726(±.032) | 0.690(±.096) | **0.692(±.101)** | 0.683(±.082) | 0.690(±.086) | 0.718(±.012) |
| 1→6 | 0.742(±.090) | 0.942(±.011) | **0.942(±.011)** | 0.942(±.011) | 0.942(±.011) | 0.789(±.064) | 0.821(±.034) | 0.807(±.051) | **0.829(±.036)** | 0.946(±.000) |
| 2→7 | 0.554(±.031) | **0.600(±.041)** | 0.564(±.064) | 0.583(±.029) | 0.562(±.019) | 0.490(±.014) | 0.542(±.056) | 0.507(±.044) | **0.567(±.066)** | 0.624(±.014) |
| 3→8 | 0.770(±.069) | 0.801(±.004) | 0.771(±.069) | 0.801(±.004) | **0.801(±.004)** | 0.803(±.063) | 0.812(±.058) | **0.815(±.035)** | 0.814(±.049) | 0.987(±.006) |
| 4→5 | 0.859(±.014) | 0.875(±.087) | **0.904(±.107)** | 0.875(±.087) | 0.875(±.087) | 0.880(±.016) | 0.870(±.006) | **0.882(±.022)** | 0.876(±.026) | 0.982(±.002) |
| mean | 0.728(±.042) | **0.789(±.035)** | 0.765(±.082) | 0.785(±.033) | 0.781(±.031) | 0.730(±.051) | 0.747(±.051) | 0.739(±.047) | **0.755(±.053)** | 0.851(±.007) |

Table 14: CDAN

| | | Importance Weighted Validation | | | | Importance Weighted Aggregation | | | | |
|---|---|---|---|---|---|---|---|---|---|---|
| | SO | Boost (IWV) | KuLSIF (IWV) | LR (IWV) | EW (IWV) | Boost (Agg) | KuLSIF (Agg) | LR (Agg) | EW (Agg) | TB |
| 0→6 | 0.703(±.040) | 0.700(±.085) | 0.743(±.021) | **0.743(±.021)** | **0.743(±.021)** | 0.537(±.029) | **0.618(±.031)** | 0.550(±.059) | 0.572(±.046) | 0.724(±.049) |
| 1→6 | 0.861(±.035) | 0.868(±.023) | **0.868(±.023)** | **0.868(±.023)** | **0.868(±.023)** | 0.793(±.080) | 0.844(±.042) | 0.857(±.013) | **0.861(±.010)** | 0.925(±.004) |
| 2→7 | 0.573(±.018) | 0.577(±.011) | **0.586(±.009)** | 0.577(±.011) | 0.577(±.011) | **0.632(±.034)** | 0.591(±.030) | 0.609(±.041) | 0.610(±.017) | 0.603(±.016) |
| 3→8 | 0.799(±.016) | **0.799(±.016)** | 0.790(±.022) | 0.799(±.016) | 0.799(±.016) | **0.814(±.005)** | 0.794(±.030) | 0.790(±.036) | 0.753(±.040) | 0.822(±.010) |
| 4→5 | 0.806(±.024) | **0.854(±.063)** | 0.799(±.143) | 0.799(±.143) | 0.854(±.063) | 0.796(±.045) | 0.792(±.043) | 0.803(±.051) | **0.819(±.053)** | 0.961(±.024) |
| mean | 0.748(±.026) | 0.760(±.040) | 0.757(±.043) | 0.757(±.043) | **0.768(±.027)** | 0.714(±.038) | **0.728(±.041)** | 0.722(±.040) | 0.723(±.033) | 0.807(±.021) |

Table 15: CMD

| | | Importance Weighted Validation | | | | Importance Weighted Aggregation | | | | |
|---|---|---|---|---|---|---|---|---|---|---|
| | SO | Boost (IWV) | KuLSIF (IWV) | LR (IWV) | EW (IWV) | Boost (Agg) | KuLSIF (Agg) | LR (Agg) | EW (Agg) | TB |
| 0→6 | 0.640(±.096) | 0.689(±.069) | 0.615(±.129) | 0.689(±.069) | **0.689(±.069)** | 0.640(±.092) | 0.638(±.069) | 0.638(±.078) | **0.642(±.073)** | 0.735(±.010) |
| 1→6 | 0.775(±.091) | 0.896(±.069) | 0.858(±.119) | 0.896(±.069) | **0.896(±.069)** | 0.839(±.017) | **0.846(±.022)** | 0.835(±.017) | 0.835(±.025) | 0.947(±.009) |
| 2→7 | 0.527(±.076) | 0.472(±.019) | **0.472(±.019)** | 0.472(±.019) | 0.472(±.019) | 0.501(±.016) | **0.515(±.029)** | 0.491(±.009) | 0.501(±.014) | 0.558(±.041) |
| 3→8 | 0.783(±.015) | 0.789(±.010) | 0.783(±.015) | 0.789(±.010) | **0.789(±.010)** | 0.794(±.052) | 0.796(±.043) | 0.806(±.048) | **0.845(±.051)** | 0.987(±.002) |
| 4→5 | 0.827(±.039) | 0.861(±.094) | **0.871(±.098)** | 0.861(±.094) | 0.849(±.074) | 0.846(±.033) | **0.859(±.031)** | 0.837(±.027) | 0.837(±.025) | 0.979(±.006) |
| mean | 0.710(±.063) | **0.741(±.052)** | 0.720(±.076) | 0.741(±.052) | 0.739(±.048) | 0.724(±.042) | 0.731(±.039) | 0.721(±.028) | **0.732(±.038)** | 0.841(±.014) |

Table 16: CoDATS

| | | Importance Weighted Validation | | | | Importance Weighted Aggregation | | | | |
|---|---|---|---|---|---|---|---|---|---|---|
| | SO | Boost (IWV) | KuLSIF (IWV) | LR (IWV) | EW (IWV) | Boost (Agg) | KuLSIF (Agg) | LR (Agg) | EW (Agg) | TB |
| 0→6 | 0.704(±.037) | 0.718(±.006) | **0.719(±.005)** | 0.718(±.006) | 0.718(±.006) | 0.694(±.069) | **0.737(±.027)** | 0.717(±.037) | 0.732(±.013) | 0.711(±.006) |
| 1→6 | 0.833(±.036) | 0.929(±.004) | 0.833(±.159) | 0.929(±.004) | **0.929(±.004)** | 0.839(±.013) | 0.839(±.038) | **0.849(±.021)** | 0.819(±.049) | 0.939(±.015) |
| 2→7 | 0.591(±.023) | 0.618(±.029) | 0.586(±.080) | 0.626(±.018) | **0.626(±.018)** | 0.597(±.030) | 0.609(±.055) | 0.606(±.046) | **0.615(±.060)** | 0.635(±.025) |
| 3→8 | 0.809(±.020) | 0.796(±.030) | **0.909(±.088)** | 0.796(±.030) | 0.796(±.030) | 0.772(±.064) | 0.805(±.031) | 0.796(±.037) | **0.807(±.028)** | 0.983(±.002) |
| 4→5 | 0.846(±.022) | **0.922(±.078)** | 0.919(±.076) | 0.922(±.078) | 0.922(±.078) | **0.859(±.045)** | 0.805(±.135) | 0.859(±.045) | 0.859(±.050) | 0.980(±.000) |
| mean | 0.757(±.027) | 0.796(±.030) | 0.793(±.081) | 0.798(±.027) | **0.798(±.027)** | 0.752(±.044) | 0.759(±.057) | 0.765(±.037) | **0.767(±.040)** | 0.850(±.010) |

Table 17: DANN

| | | Importance Weighted Validation | | | | Importance Weighted Aggregation | | | | |
|---|---|---|---|---|---|---|---|---|---|---|
| | SO | Boost (IWV) | KuLSIF (IWV) | LR (IWV) | EW (IWV) | Boost (Agg) | KuLSIF (Agg) | LR (Agg) | EW (Agg) | TB |
| 0→6 | 0.575(±.083) | 0.649(±.012) | 0.619(±.038) | 0.649(±.012) | **0.650(±.015)** | 0.615(±.031) | 0.604(±.014) | 0.617(±.033) | **0.622(±.045)** | 0.651(±.010) |
| 1→6 | 0.875(±.014) | **0.856(±.067)** | 0.844(±.084) | 0.856(±.067) | 0.856(±.067) | **0.842(±.040)** | 0.826(±.037) | 0.836(±.042) | 0.825(±.044) | 0.899(±.016) |
| 2→7 | 0.487(±.016) | 0.442(±.136) | **0.455(±.054)** | 0.405(±.073) | 0.438(±.054) | **0.501(±.045)** | 0.464(±.039) | 0.443(±.051) | 0.464(±.041) | 0.533(±.107) |
| 3→8 | 0.815(±.019) | **0.827(±.025)** | 0.806(±.019) | 0.827(±.025) | 0.827(±.025) | 0.794(±.023) | 0.803(±.033) | 0.806(±.025) | **0.807(±.035)** | 0.822(±.029) |
| 4→5 | 0.831(±.048) | 0.792(±.020) | 0.777(±.033) | 0.792(±.020) | 0.792(±.020) | 0.810(±.070) | 0.811(±.049) | **0.814(±.064)** | 0.797(±.037) | 0.888(±.037) |
| mean | 0.716(±.036) | **0.713(±.052)** | 0.701(±.046) | 0.705(±.039) | 0.712(±.036) | **0.713(±.042)** | 0.702(±.035) | 0.703(±.043) | 0.703(±.040) | 0.758(±.040) |

Table 18: DDC

| | | Importance Weighted Validation | | | | Importance Weighted Aggregation | | | | |
|---|---|---|---|---|---|---|---|---|---|---|
| | SO | Boost (IWV) | KuLSIF (IWV) | LR (IWV) | EW (IWV) | Boost (Agg) | KuLSIF (Agg) | LR (Agg) | EW (Agg) | TB |
| 0→6 | 0.692(±.026) | 0.703(±.038) | 0.703(±.038) | 0.708(±.029) | **0.708(±.029)** | **0.618(±.025)** | 0.600(±.018) | 0.574(±.048) | 0.586(±.049) | 0.703(±.030) |
| 1→6 | 0.862(±.018) | 0.864(±.017) | 0.849(±.034) | 0.864(±.017) | **0.864(±.017)** | 0.785(±.032) | **0.806(±.040)** | 0.790(±.016) | 0.789(±.024) | 0.911(±.017) |
| 2→7 | 0.509(±.097) | 0.499(±.079) | 0.487(±.019) | **0.499(±.079)** | 0.499(±.079) | 0.491(±.051) | **0.522(±.028)** | 0.518(±.035) | 0.515(±.035) | 0.565(±.117) |
| 3→8 | 0.798(±.016) | 0.799(±.013) | 0.796(±.018) | 0.799(±.013) | **0.799(±.013)** | 0.743(±.011) | 0.737(±.013) | 0.745(±.006) | **0.758(±.008)** | 0.822(±.010) |
| 4→5 | 0.862(±.036) | 0.868(±.053) | **0.884(±.083)** | 0.868(±.053) | 0.868(±.053) | 0.728(±.148) | 0.771(±.074) | 0.784(±.054) | **0.789(±.041)** | 0.960(±.010) |
| mean | 0.745(±.039) | 0.747(±.040) | 0.744(±.038) | 0.748(±.038) | **0.748(±.038)** | 0.673(±.054) | 0.687(±.035) | 0.682(±.032) | **0.687(±.031)** | 0.792(±.037) |

Table 19: DCoral

| | | Importance Weighted Validation | | | | Importance Weighted Aggregation | | | | |
|---|---|---|---|---|---|---|---|---|---|---|
| | SO | Boost (IWV) | KuLSIF (IWV) | LR (IWV) | EW (IWV) | Boost (Agg) | KuLSIF (Agg) | LR (Agg) | EW (Agg) | TB |
| 0→6 | 0.708(±.011) | **0.708(±.011)** | 0.708(±.011) | 0.708(±.011) | 0.708(±.011) | 0.608(±.154) | 0.649(±.114) | 0.636(±.119) | **0.717(±.055)** | 0.739(±.016) |
| 1→6 | 0.756(±.058) | 0.814(±.056) | **0.878(±.104)** | 0.814(±.056) | 0.814(±.056) | 0.768(±.073) | 0.786(±.032) | 0.785(±.049) | **0.824(±.047)** | 0.942(±.004) |
| 2→7 | 0.531(±.057) | 0.573(±.083) | **0.637(±.072)** | 0.573(±.083) | 0.579(±.083) | 0.519(±.067) | **0.542(±.086)** | 0.504(±.092) | 0.499(±.048) | 0.688(±.004) |
| 3→8 | 0.807(±.008) | 0.807(±.008) | **0.866(±.106)** | 0.807(±.008) | 0.807(±.008) | 0.832(±.081) | 0.836(±.051) | 0.845(±.075) | **0.849(±.047)** | 0.911(±.070) |
| 4→5 | 0.839(±.016) | 0.839(±.016) | **0.982(±.008)** | 0.839(±.016) | 0.839(±.016) | **0.861(±.034)** | 0.837(±.049) | 0.854(±.029) | 0.845(±.039) | 0.984(±.004) |
| mean | 0.728(±.030) | 0.748(±.035) | **0.814(±.060)** | 0.748(±.035) | 0.749(±.035) | 0.718(±.082) | 0.730(±.066) | 0.725(±.073) | **0.747(±.047)** | 0.853(±.020) |

Table 20: DIRT

| | | Importance Weighted Validation | | | | Importance Weighted Aggregation | | | | |
|---|---|---|---|---|---|---|---|---|---|---|
| | SO | Boost (IWV) | KuLSIF (IWV) | LR (IWV) | EW (IWV) | Boost (Agg) | KuLSIF (Agg) | LR (Agg) | EW (Agg) | TB |
| 0→6 | 0.653(±.033) | **0.572(±.103)** | 0.557(±.078) | 0.572(±.103) | 0.572(±.103) | **0.683(±.019)** | 0.683(±.023) | 0.675(±.019) | 0.678(±.024) | 0.664(±.010) |
| 1→6 | 0.806(±.083) | 0.908(±.065) | 0.894(±.057) | **0.908(±.065)** | 0.908(±.065) | 0.817(±.072) | 0.814(±.081) | 0.815(±.075) | **0.818(±.081)** | 0.940(±.005) |
| 2→7 | 0.490(±.053) | **0.494(±.030)** | 0.476(±.056) | 0.482(±.012) | 0.476(±.009) | 0.497(±.024) | **0.522(±.054)** | 0.493(±.030) | 0.503(±.025) | 0.586(±.083) |
| 3→8 | 0.797(±.016) | 0.816(±.010) | **0.857(±.091)** | 0.816(±.010) | 0.816(±.010) | 0.828(±.014) | 0.829(±.008) | 0.832(±.010) | **0.836(±.004)** | 0.982(±.010) |
| 4→5 | 0.861(±.043) | **0.844(±.122)** | 0.799(±.060) | 0.844(±.122) | 0.844(±.122) | 0.829(±.022) | **0.832(±.018)** | 0.826(±.035) | 0.822(±.032) | 0.982(±.005) |
| mean | 0.721(±.046) | **0.727(±.066)** | 0.717(±.068) | 0.725(±.062) | 0.723(±.062) | 0.731(±.030) | **0.736(±.037)** | 0.728(±.034) | 0.731(±.033) | 0.831(±.023) |

Table 21: DSAN

| | SO | Importance Weighted Validation | | | | Importance Weighted Aggregation | | | | TB |
|---|---|---|---|---|---|---|---|---|---|---|
| | | Boost (IWV) | KuLSIF (IWV) | LR (IWV) | EW (IWV) | Boost (Agg) | KuLSIF (Agg) | LR (Agg) | EW (Agg) | |
| 0→6 | 0.719(±.041) | 0.704(±.011) | 0.701(±.006) | **0.715(±.019)** | 0.715(±.019) | 0.583(±.120) | 0.618(±.070) | 0.586(±.089) | **0.622(±.037)** | 0.733(±.007) |
| 1→6 | 0.787(±.110) | 0.749(±.060) | **0.768(±.082)** | 0.749(±.060) | 0.749(±.060) | 0.771(±.012) | 0.753(±.035) | 0.774(±.006) | **0.774(±.005)** | 0.914(±.010) |
| 2→7 | 0.528(±.103) | 0.497(±.105) | 0.516(±.089) | 0.530(±.079) | **0.530(±.079)** | 0.454(±.058) | 0.493(±.045) | 0.469(±.069) | **0.497(±.067)** | 0.546(±.071) |
| 3→8 | 0.797(±.007) | **0.805(±.010)** | 0.770(±.086) | 0.805(±.010) | 0.805(±.010) | 0.729(±.080) | 0.763(±.049) | 0.755(±.063) | **0.792(±.043)** | 0.831(±.022) |
| 4→5 | 0.861(±.022) | 0.844(±.016) | **0.895(±.041)** | 0.844(±.016) | 0.844(±.016) | 0.816(±.037) | 0.822(±.040) | 0.822(±.036) | **0.831(±.033)** | 0.940(±.040) |
| mean | 0.739(±.057) | 0.720(±.041) | **0.730(±.061)** | 0.728(±.037) | 0.728(±.037) | 0.671(±.061) | 0.690(±.048) | 0.681(±.053) | **0.703(±.037)** | 0.793(±.030) |

Table 22: HoMM

| | SO | Importance Weighted Validation | | | | Importance Weighted Aggregation | | | | TB |
|---|---|---|---|---|---|---|---|---|---|---|
| | | Boost (IWV) | KuLSIF (IWV) | LR (IWV) | EW (IWV) | Boost (Agg) | KuLSIF (Agg) | LR (Agg) | EW (Agg) | |
| 0→6 | 0.706(±.042) | 0.699(±.055) | 0.685(±.028) | **0.699(±.055)** | 0.699(±.055) | 0.571(±.076) | 0.597(±.025) | 0.574(±.040) | **0.636(±.037)** | 0.732(±.024) |
| 1→6 | 0.792(±.098) | 0.732(±.009) | 0.722(±.009) | **0.732(±.009)** | 0.732(±.009) | 0.757(±.030) | 0.751(±.010) | 0.758(±.027) | **0.765(±.016)** | 0.914(±.028) |
| 2→7 | 0.552(±.073) | 0.549(±.051) | 0.530(±.071) | **0.549(±.072)** | 0.530(±.071) | 0.494(±.090) | 0.516(±.048) | 0.496(±.083) | **0.531(±.035)** | 0.552(±.073) |
| 3→8 | 0.790(±.016) | 0.802(±.018) | **0.820(±.018)** | 0.802(±.018) | 0.802(±.018) | 0.779(±.090) | 0.801(±.061) | **0.801(±.075)** | 0.796(±.040) | 0.932(±.005) |
| 4→5 | 0.852(±.036) | 0.846(±.066) | **0.859(±.031)** | 0.846(±.066) | 0.846(±.066) | **0.826(±.016)** | 0.820(±.051) | 0.824(±.014) | 0.824(±.018) | 0.947(±.022) |
| mean | 0.738(±.053) | 0.726(±.040) | 0.723(±.031) | **0.726(±.044)** | 0.722(±.044) | 0.685(±.060) | 0.697(±.039) | 0.690(±.048) | **0.710(±.029)** | 0.815(±.030) |

Table 23: MMDA

## C.3 MINIDOMAINNET

| | SO | Importance Weighted Validation | | | | Importance Weighted Aggregation | | | | TB |
|---|---|---|---|---|---|---|---|---|---|---|
| | | Boost (IWV) | KuLSIF (IWV) | LR (IWV) | EW (IWV) | Boost (Agg) | KuLSIF (Agg) | LR (Agg) | EW (Agg) | |
| R→C | 0.544(±.017) | **0.546(±.031)** | 0.546(±.031) | 0.546(±.031) | 0.546(±.031) | 0.546(±.017) | 0.546(±.017) | 0.546(±.017) | **0.555(±.017)** | 0.582(±.046) |
| R→I | 0.380(±.018) | **0.394(±.021)** | 0.394(±.021) | 0.394(±.021) | 0.394(±.021) | 0.400(±.009) | 0.403(±.024) | 0.400(±.021) | 0.395(±.003) | 0.395(±.020) |
| R→P | 0.723(±.009) | 0.711(±.015) | 0.711(±.015) | 0.711(±.015) | **0.712(±.017)** | **0.707(±.018)** | 0.705(±.021) | 0.705(±.019) | 0.705(±.021) | 0.723(±.009) |
| R→Q | 0.322(±.027) | 0.340(±.004) | 0.333(±.003) | **0.340(±.004)** | 0.340(±.004) | 0.316(±.015) | **0.317(±.011)** | 0.314(±.015) | 0.315(±.013) | 0.335(±.004) |
| R→S | 0.579(±.017) | 0.586(±.001) | **0.586(±.001)** | 0.586(±.001) | 0.586(±.001) | 0.592(±.003) | **0.592(±.000)** | 0.591(±.001) | 0.588(±.004) | 0.592(±.009) |
| mean | 0.509(±.018) | 0.515(±.014) | 0.514(±.014) | 0.515(±.014) | **0.516(±.015)** | 0.512(±.012) | **0.513(±.013)** | 0.511(±.015) | 0.512(±.012) | 0.525(±.018) |

Table 24: AdvSKM

| | SO | Importance Weighted Validation | | | | Importance Weighted Aggregation | | | | TB |
|---|---|---|---|---|---|---|---|---|---|---|
| | | Boost (IWV) | KuLSIF (IWV) | LR (IWV) | EW (IWV) | Boost (Agg) | KuLSIF (Agg) | LR (Agg) | EW (Agg) | |
| R→C | 0.557(±.008) | **0.574(±.008)** | 0.549(±.036) | 0.549(±.036) | 0.549(±.036) | 0.593(±.026) | **0.598(±.028)** | 0.598(±.030) | 0.598(±.038) | 0.615(±.008) |
| R→I | 0.386(±.010) | 0.357(±.027) | 0.370(±.014) | 0.357(±.027) | **0.374(±.015)** | 0.326(±.079) | 0.383(±.003) | 0.382(±.011) | **0.388(±.007)** | 0.386(±.010) |
| R→P | 0.709(±.022) | 0.688(±.014) | 0.708(±.025) | 0.708(±.025) | **0.710(±.023)** | 0.717(±.020) | 0.714(±.012) | **0.719(±.016)** | 0.709(±.025) | 0.716(±.055) |
| R→Q | 0.342(±.006) | **0.338(±.008)** | 0.338(±.008) | 0.338(±.008) | 0.338(±.008) | 0.351(±.016) | **0.352(±.013)** | 0.351(±.024) | 0.351(±.024) | 0.442(±.114) |
| R→S | 0.576(±.027) | 0.595(±.008) | **0.599(±.011)** | 0.599(±.011) | 0.599(±.011) | **0.608(±.012)** | 0.605(±.013) | 0.606(±.015) | 0.605(±.013) | 0.651(±.056) |
| mean | 0.514(±.015) | 0.511(±.013) | 0.513(±.019) | 0.510(±.021) | **0.514(±.018)** | 0.519(±.031) | 0.530(±.014) | **0.531(±.019)** | 0.530(±.021) | 0.562(±.049) |

Table 25: CDAN

| | SO | Importance Weighted Validation | | | | Importance Weighted Aggregation | | | | TB |
|---|---|---|---|---|---|---|---|---|---|---|
| | | Boost (IWV) | KuLSIF (IWV) | LR (IWV) | EW (IWV) | Boost (Agg) | KuLSIF (Agg) | LR (Agg) | EW (Agg) | |
| R→C | 0.555(±.042) | **0.557(±.038)** | 0.555(±.042) | 0.555(±.042) | 0.549(±.036) | 0.541(±.030) | 0.525(±.028) | 0.541(±.030) | **0.544(±.037)** | 0.617(±.025) |
| R→I | 0.377(±.009) | **0.395(±.010)** | 0.383(±.010) | 0.382(±.009) | 0.379(±.007) | 0.388(±.021) | 0.403(±.024) | **0.406(±.030)** | 0.397(±.029) | 0.389(±.032) |
| R→P | 0.704(±.020) | **0.727(±.023)** | 0.727(±.023) | 0.727(±.023) | 0.727(±.023) | **0.743(±.014)** | 0.740(±.018) | 0.735(±.021) | 0.739(±.023) | 0.734(±.014) |
| R→Q | 0.332(±.015) | **0.355(±.035)** | 0.355(±.035) | 0.355(±.035) | 0.355(±.035) | **0.316(±.041)** | 0.312(±.040) | 0.315(±.045) | 0.316(±.046) | 0.378(±.008) |
| R→S | 0.578(±.022) | 0.592(±.005) | **0.592(±.005)** | 0.592(±.005) | 0.592(±.005) | 0.599(±.019) | **0.599(±.013)** | 0.599(±.017) | 0.594(±.018) | 0.590(±.026) |
| mean | 0.509(±.022) | **0.525(±.022)** | 0.522(±.023) | 0.522(±.023) | 0.520(±.021) | 0.517(±.025) | 0.516(±.025) | **0.519(±.029)** | 0.518(±.031) | 0.542(±.021) |

Table 26: CMD

| | SO | Importance Weighted Validation | | | | Importance Weighted Aggregation | | | | TB |
|---|---|---|---|---|---|---|---|---|---|---|
| | | Boost (IWV) | KuLSIF (IWV) | LR (IWV) | EW (IWV) | Boost (Agg) | KuLSIF (Agg) | LR (Agg) | EW (Agg) | |
| R→C | 0.538(±.072) | **0.566(±.025)** | 0.557(±.028) | 0.557(±.028) | 0.557(±.028) | **0.579(±.033)** | 0.566(±.014) | 0.557(±.014) | **0.560(±.005)** | 0.617(±.066) |
| R→I | 0.365(±.037) | 0.362(±.034) | **0.377(±.015)** | 0.362(±.034) | 0.377(±.015) | 0.360(±.041) | 0.368(±.025) | 0.368(±.027) | **0.371(±.032)** | 0.406(±.021) |
| R→P | 0.689(±.024) | 0.723(±.020) | **0.723(±.020)** | 0.723(±.020) | 0.723(±.020) | 0.726(±.016) | **0.728(±.016)** | 0.727(±.019) | 0.726(±.020) | 0.732(±.017) |
| R→Q | 0.322(±.022) | **0.329(±.023)** | 0.329(±.023) | 0.329(±.023) | 0.329(±.023) | 0.359(±.022) | 0.357(±.015) | 0.364(±.020) | **0.364(±.018)** | 0.417(±.025) |
| R→S | 0.597(±.004) | **0.588(±.012)** | 0.588(±.012) | 0.588(±.012) | 0.588(±.012) | 0.611(±.004) | 0.611(±.006) | 0.609(±.006) | **0.614(±.008)** | 0.627(±.008) |
| mean | 0.502(±.032) | 0.514(±.023) | **0.515(±.020)** | 0.512(±.023) | 0.515(±.020) | 0.527(±.023) | 0.526(±.015) | 0.525(±.017) | **0.527(±.016)** | 0.560(±.027) |

Table 27: CoDATS

| | | Importance Weighted Validation | | | | Importance Weighted Aggregation | | | | |
|---|---|---|---|---|---|---|---|---|---|---|
| | SO | Boost (IWV) | KuLSIF (IWV) | LR (IWV) | EW (IWV) | Boost (Agg) | KuLSIF (Agg) | LR (Agg) | EW (Agg) | TB |
| R→C | 0.519(±.013) | **0.552(±.017)** | 0.552(±.017) | 0.552(±.017) | 0.546(±.017) | 0.560(±.005) | **0.563(±.005)** | 0.557(±.000) | 0.560(±.017) | 0.631(±.038) |
| R→I | 0.376(±.016) | 0.379(±.036) | 0.388(±.025) | 0.379(±.036) | **0.389(±.024)** | 0.370(±.023) | 0.379(±.026) | **0.380(±.016)** | 0.371(±.012) | 0.400(±.025) |
| R→P | 0.705(±.011) | 0.699(±.006) | 0.703(±.011) | 0.703(±.011) | **0.710(±.015)** | 0.718(±.004) | 0.717(±.004) | **0.719(±.000)** | 0.712(±.006) | 0.713(±.013) |
| R→Q | 0.300(±.043) | 0.332(±.015) | **0.332(±.015)** | 0.332(±.015) | 0.329(±.013) | 0.317(±.023) | 0.315(±.025) | 0.318(±.022) | **0.322(±.023)** | 0.392(±.005) |
| R→S | 0.581(±.014) | **0.596(±.024)** | 0.596(±.024) | 0.596(±.024) | 0.596(±.024) | **0.608(±.010)** | 0.607(±.011) | 0.605(±.011) | 0.601(±.013) | 0.657(±.015) |
| mean | 0.496(±.019) | 0.511(±.020) | 0.514(±.018) | 0.512(±.021) | **0.515(±.018)** | 0.515(±.013) | 0.516(±.014) | **0.516(±.010)** | 0.513(±.014) | 0.559(±.019) |

Table 28: DANN

| | | Importance Weighted Validation | | | | Importance Weighted Aggregation | | | | |
|---|---|---|---|---|---|---|---|---|---|---|
| | SO | Boost (IWV) | KuLSIF (IWV) | LR (IWV) | EW (IWV) | Boost (Agg) | KuLSIF (Agg) | LR (Agg) | EW (Agg) | TB |
| R→C | 0.555(±.045) | 0.560(±.017) | 0.563(±.013) | **0.563(±.013)** | 0.546(±.025) | 0.563(±.025) | 0.563(±.033) | 0.555(±.029) | **0.571(±.034)** | 0.585(±.076) |
| R→I | 0.374(±.014) | 0.379(±.009) | 0.379(±.009) | 0.379(±.009) | **0.386(±.009)** | 0.397(±.017) | 0.394(±.023) | **0.398(±.020)** | 0.394(±.023) | 0.416(±.014) |
| R→P | 0.709(±.025) | 0.706(±.012) | **0.707(±.013)** | 0.706(±.012) | 0.706(±.012) | 0.717(±.002) | 0.717(±.002) | 0.719(±.003) | **0.724(±.005)** | 0.717(±.023) |
| R→Q | 0.334(±.002) | **0.333(±.005)** | 0.333(±.005) | 0.333(±.005) | 0.333(±.005) | 0.323(±.023) | **0.324(±.022)** | 0.324(±.025) | 0.324(±.015) | 0.337(±.004) |
| R→S | 0.576(±.022) | **0.587(±.008)** | 0.586(±.009) | 0.586(±.009) | 0.586(±.009) | **0.580(±.006)** | 0.579(±.009) | 0.577(±.008) | 0.577(±.005) | 0.585(±.021) |
| mean | 0.510(±.021) | 0.513(±.010) | **0.514(±.010)** | 0.513(±.010) | 0.512(±.012) | 0.516(±.015) | 0.515(±.018) | 0.515(±.016) | **0.518(±.018)** | 0.528(±.027) |

Table 29: DDC

| | | Importance Weighted Validation | | | | Importance Weighted Aggregation | | | | |
|---|---|---|---|---|---|---|---|---|---|---|
| | SO | Boost (IWV) | KuLSIF (IWV) | LR (IWV) | EW (IWV) | Boost (Agg) | KuLSIF (Agg) | LR (Agg) | EW (Agg) | TB |
| R→C | 0.555(±.045) | 0.566(±.008) | 0.566(±.008) | 0.566(±.008) | **0.579(±.021)** | 0.546(±.005) | **0.552(±.009)** | 0.546(±.013) | 0.549(±.008) | 0.601(±.056) |
| R→I | 0.376(±.012) | **0.377(±.015)** | 0.373(±.007) | 0.373(±.007) | 0.373(±.007) | 0.379(±.014) | **0.379(±.018)** | 0.379(±.011) | 0.376(±.005) | 0.386(±.007) |
| R→P | 0.711(±.021) | 0.719(±.011) | 0.719(±.011) | 0.719(±.011) | 0.709(±.009) | 0.732(±.003) | 0.728(±.006) | 0.734(±.004) | **0.739(±.011)** | 0.734(±.014) |
| R→Q | 0.316(±.041) | 0.363(±.023) | **0.363(±.023)** | 0.363(±.023) | 0.363(±.023) | 0.334(±.026) | 0.333(±.021) | **0.337(±.029)** | 0.332(±.032) | 0.388(±.002) |
| R→S | 0.569(±.020) | 0.585(±.014) | **0.585(±.014)** | 0.585(±.014) | 0.585(±.014) | 0.611(±.014) | 0.608(±.010) | 0.609(±.009) | **0.611(±.012)** | 0.658(±.035) |
| mean | 0.505(±.028) | 0.522(±.012) | **0.522(±.014)** | 0.521(±.013) | 0.522(±.015) | 0.520(±.012) | 0.520(±.013) | 0.521(±.013) | **0.521(±.013)** | 0.554(±.023) |

Table 30: DCoral

| | | Importance Weighted Validation | | | | Importance Weighted Aggregation | | | | |
|---|---|---|---|---|---|---|---|---|---|---|
| | SO | Boost (IWV) | KuLSIF (IWV) | LR (IWV) | EW (IWV) | Boost (Agg) | KuLSIF (Agg) | LR (Agg) | EW (Agg) | TB |
| R→C | 0.530(±.013) | 0.538(±.005) | 0.555(±.031) | **0.566(±.050)** | 0.541(±.025) | **0.574(±.022)** | 0.568(±.026) | 0.566(±.008) | 0.563(±.013) | 0.587(±.037) |
| R→I | 0.373(±.038) | 0.363(±.038) | **0.379(±.021)** | 0.379(±.021) | 0.360(±.011) | 0.397(±.046) | **0.421(±.021)** | 0.410(±.013) | 0.418(±.020) | 0.431(±.045) |
| R→P | 0.704(±.031) | 0.663(±.030) | **0.663(±.030)** | 0.663(±.030) | 0.663(±.030) | 0.711(±.027) | **0.712(±.028)** | 0.711(±.026) | 0.708(±.029) | 0.715(±.065) |
| R→Q | 0.340(±.020) | **0.326(±.086)** | 0.315(±.069) | 0.326(±.086) | 0.286(±.049) | **0.323(±.028)** | 0.323(±.023) | 0.319(±.022) | 0.316(±.029) | 0.382(±.027) |
| R→S | 0.550(±.028) | 0.566(±.008) | 0.566(±.008) | 0.566(±.008) | **0.569(±.004)** | 0.589(±.033) | **0.592(±.033)** | 0.591(±.032) | 0.592(±.016) | 0.588(±.033) |
| mean | 0.499(±.026) | 0.491(±.033) | 0.495(±.032) | **0.500(±.039)** | 0.484(±.024) | 0.519(±.031) | **0.523(±.025)** | 0.519(±.020) | 0.519(±.021) | 0.541(±.041) |

Table 31: DIRT

| | | Importance Weighted Validation | | | | Importance Weighted Aggregation | | | | |
|---|---|---|---|---|---|---|---|---|---|---|
| | SO | Boost (IWV) | KuLSIF (IWV) | LR (IWV) | EW (IWV) | Boost (Agg) | KuLSIF (Agg) | LR (Agg) | EW (Agg) | TB |
| R→C | 0.555(±.042) | 0.538(±.029) | 0.563(±.053) | 0.538(±.029) | **0.568(±.021)** | 0.574(±.030) | 0.566(±.022) | 0.571(±.026) | **0.582(±.036)** | 0.648(±.014) |
| R→I | 0.377(±.009) | 0.341(±.047) | 0.360(±.053) | 0.341(±.047) | **0.383(±.021)** | 0.350(±.015) | 0.368(±.009) | **0.370(±.014)** | 0.365(±.007) | 0.404(±.016) |
| R→P | 0.705(±.021) | 0.711(±.012) | **0.711(±.012)** | 0.711(±.012) | 0.711(±.012) | 0.723(±.013) | 0.723(±.011) | 0.729(±.004) | **0.730(±.007)** | 0.716(±.000) |
| R→Q | 0.332(±.012) | **0.343(±.008)** | 0.329(±.017) | 0.332(±.011) | 0.332(±.011) | 0.339(±.016) | 0.338(±.014) | 0.338(±.018) | **0.339(±.019)** | 0.478(±.013) |
| R→S | 0.577(±.026) | 0.587(±.003) | **0.587(±.003)** | 0.587(±.003) | 0.587(±.003) | **0.607(±.015)** | 0.605(±.012) | 0.603(±.012) | 0.603(±.019) | 0.639(±.052) |
| mean | 0.509(±.022) | 0.504(±.020) | 0.510(±.028) | 0.502(±.020) | **0.516(±.013)** | 0.518(±.018) | 0.520(±.014) | 0.522(±.015) | **0.524(±.018)** | 0.577(±.019) |

Table 32: DSAN

| | | Importance Weighted Validation | | | | Importance Weighted Aggregation | | | | |
|---|---|---|---|---|---|---|---|---|---|---|
| | SO | Boost (IWV) | KuLSIF (IWV) | LR (IWV) | EW (IWV) | Boost (Agg) | KuLSIF (Agg) | LR (Agg) | EW (Agg) | TB |
| R→C | 0.552(±.041) | **0.568(±.062)** | 0.568(±.062) | 0.568(±.062) | 0.546(±.041) | 0.574(±.014) | 0.579(±.017) | 0.577(±.019) | **0.582(±.022)** | 0.607(±.022) |
| R→I | 0.373(±.014) | 0.376(±.021) | **0.391(±.036)** | 0.370(±.015) | 0.380(±.027) | 0.389(±.044) | 0.409(±.009) | 0.398(±.024) | **0.413(±.009)** | 0.413(±.041) |
| R→P | 0.709(±.026) | 0.714(±.005) | **0.714(±.005)** | 0.714(±.005) | 0.714(±.005) | 0.723(±.018) | **0.724(±.015)** | 0.722(±.014) | 0.722(±.014) | 0.721(±.003) |
| R→Q | 0.332(±.009) | **0.355(±.014)** | 0.355(±.014) | 0.355(±.014) | 0.355(±.014) | 0.323(±.010) | 0.321(±.009) | **0.324(±.011)** | 0.324(±.010) | 0.370(±.013) |
| R→S | 0.579(±.022) | 0.576(±.006) | **0.576(±.006)** | 0.576(±.006) | 0.576(±.006) | **0.599(±.012)** | 0.598(±.013) | 0.596(±.012) | 0.596(±.012) | 0.609(±.011) |
| mean | 0.509(±.023) | 0.518(±.021) | **0.521(±.025)** | 0.516(±.020) | 0.514(±.019) | 0.521(±.019) | 0.526(±.013) | 0.523(±.016) | **0.527(±.014)** | 0.544(±.018) |

Table 33: HoMM

| | | Importance Weighted Validation | | | | Importance Weighted Aggregation | | | | |
|---|---|---|---|---|---|---|---|---|---|---|
| | SO | Boost (IWV) | KuLSIF (IWV) | LR (IWV) | EW (IWV) | Boost (Agg) | KuLSIF (Agg) | LR (Agg) | EW (Agg) | TB |
| R→C | 0.555(±.042) | 0.568(±.021) | **0.568(±.021)** | 0.568(±.021) | 0.566(±.022) | 0.582(±.025) | 0.590(±.025) | 0.590(±.025) | **0.593(±.021)** | 0.585(±.034) |
| R→I | 0.377(±.009) | **0.398(±.040)** | 0.379(±.030) | 0.379(±.030) | 0.379(±.030) | **0.389(±.009)** | 0.374(±.013) | 0.380(±.018) | 0.370(±.009) | 0.415(±.016) |
| R→P | 0.704(±.020) | 0.709(±.010) | **0.709(±.010)** | 0.709(±.010) | 0.709(±.010) | 0.728(±.004) | 0.732(±.008) | 0.732(±.008) | **0.733(±.007)** | 0.735(±.007) |
| R→Q | 0.332(±.015) | **0.353(±.025)** | 0.353(±.025) | 0.353(±.025) | 0.353(±.025) | **0.343(±.013)** | 0.341(±.012) | 0.343(±.019) | 0.342(±.020) | 0.392(±.018) |
| R→S | 0.578(±.022) | 0.583(±.006) | 0.583(±.006) | 0.583(±.006) | **0.588(±.012)** | 0.606(±.010) | **0.606(±.010)** | 0.605(±.009) | 0.603(±.009) | 0.605(±.008) |
| mean | 0.509(±.022) | **0.522(±.020)** | 0.518(±.018) | 0.518(±.018) | 0.519(±.020) | 0.530(±.012) | 0.529(±.013) | **0.530(±.016)** | 0.528(±.013) | 0.546(±.017) |

Table 34: MMDA

