# OpenReview forum: "Binary Losses for Density Ratio Estimation"
_ICLR.cc/2025/Conference — ICLR 2025 Poster_

### Official Review · Reviewer_55K4 · 2024-10-28

**Soundness:** 4
**Presentation:** 3
**Contribution:** 3
**Rating:** 8
**Confidence:** 2

**Summary:**

A standard technique to estimate the ratio between densities of two probability distributions $P$ and $Q$ from samples from each of the distributions is to train a binary classifier that distinguishes the samples. Namely, one labels samples from $P$ with the label $1$, and samples from $Q$ with the label $-1$, and empirically minimizes a loss function on these samples. Depending on the loss function that is chosen, it is known that the as the number of samples tends to infinity, the empirical minimizer effectively minimizes a certain Bregman divergence between the true density ratio and the estimated density ratio.

However, the convex potential of this Bregman divergence depends on the chosen loss function. Namely, the Bregman divergence to the true density ratio that our estimator ends up minimizing depends heavily on the loss function that we chose. As it turns out, most commonly used loss functions (like the logistic loss), correspond to Bregman divergences that do not appropriately penalize discrepancies at large density ratio values, instead penalizing density ratio errors at smaller values---this might lead to suboptimal density ratio estimates in many applications.

This paper takes an inverted approach: given a convex potential $\phi$ (and a ``probability link function'' $g$), the paper characterizes a unique loss function $l_{\phi, g}$, such that upon empirically minimizing $l_{\phi, g}$, as the number of samples grows, the Bregman divergence with potential $\phi$ is minimized. The paper furthermore proposes a canonical link function $g$, which induces a convex loss function $l$, and is hence computationally amenable to empirical risk minimization.

One convenient application of the characterization in this work is that it gives a way for a practitioner to design a loss function that would have the properties they desire. For example, as elaborated in Section 5 by the authors, if one cares about penalizing errors in larger density ratios more than errors in smaller density ratios, one can specify a potential $\phi$ for a Bregman divergence $D_\phi$ that ensures this, and thereafter using the characterization in the paper, obtain the associated loss function $l_\phi$. If one now minimizes $l_\phi$ on the samples, then in the limit, one would be minimizing $D_\phi$ (which was chosen so as to prioritize accurate estimation of large density ratios). In particular, the authors specifically propose two convex potentials $\phi$ for this purpose--the Exponential Weight (EW) function and polynomial weight functions, and derive the associated loss functions from their characterization.

Finally, the authors empirically validate minimizing these loss functions as compared to the standard loss functions on a variety of synthetic as well as real-world datasets. They show that minimizing their loss functions leads to better performance on importance weighting tasks on a range of datasets. The experimental evaluation appears quite thorough and extensive.

**Strengths:**

The characterization provided by the authors significantly completes the picture laid out in prior work by Menon and Ong 2016, and work by Reid and Williamson (see Remark 1 for specifics). The motivation for considering the inverted problem is convincing---one can imagine ascribing certain desired properties of an estimator to the minimizer of a Bregman divergence, and thereafter, using the characterization derived in the paper, obtain the correct loss function to minimize on the data that realizes the minimization of this Bregman divergence (and hence has the desried property). The authors specifically consider the property "small errors on large density ratios", and obtain strong empirical results for minimizing the loss functions through their characterization. In my view, this is a valuable contribution that enhances our toolbox for estimating density ratios in a principled manner. The writing of the paper is also geneally good, although at some places, it becomes dense with a lot of assumed context.

**Weaknesses:**

Up until you mention your first contribution, the reader has only looked at the pseudocode of Algorithm 1, which uses a probability link function $\Psi$. There has been no mention of $g$ as yet. Correct me if I am wrong, but my understanding is that $g(x)$ in Algorithm 1 is simply $\Psi^{-1}(x)/1-\Psi^{-1}(x)$---it would be helpful to at least mention this before introducing "Density ratio link $g$" in line 78. Because otherwise, the reader, who has just seen $\Psi$ in Algorithm 1, is a little confused about where $g$ sprang out of nowhere, and how it is relevant.

---

Minor/typos: \
Line 78: I believe the loss function for an arbitrary $g$ is not convex, but only strictly roper composite (the loss function for the canonical $g$, as stated in the next sentence, is convex).

Line 273: I believe in the denominator, there is a typo (should be $g_{can}$ instead of $g$)

**Questions:**

In the conclusion, you mention that the sample complexity of these tasks is not known. Do you mean to say that the empirical risk minimizer of the loss converges to the minimizer of the Bregman divergence only as the number of samples goes to infinity, but we do not know finite-sample error bounds for the empirical minimizer (similar to how in PAC learning theory, this would correspond to an additional "complexity of F"/#samples error term)?

---

> ### Author Response · Authors · 2024-11-19
> **Answer**
>
> We thank you for your comments which help to improve our work and appreciate all your positive comments!
> All mentioned minor issues are clarified in the new manuscript.
>
> Regarding your question: yes, we refer to the concevrgence of the empirical risk minimizer to the true density ratio in terms of some suitable error term as, e.g., the Bregman divergence. Sample complexity bounds of [1] don't hold, as our novel loss functions are not self-concordance.
>
> [1] Zellinger, Werner, Stefan Kindermann, and Sergei V. Pereverzyev. "Adaptive learning of density ratios in RKHS." The Journal of Machine Learning Research 24.1 (2023): 18863-18891.
>
> Thank you again for your time and effort used to review our manuscript.

---

> > ### Comment · Reviewer_55K4 · 2024-11-25
> > **Response to author comment**
> >
> > Thank you for the clarification. I maintain that the contributions in this paper are valuable, and hence maintain my positive score

---

### Official Review · Reviewer_Ps3k · 2024-10-31

**Soundness:** 3
**Presentation:** 2
**Contribution:** 3
**Rating:** 6
**Confidence:** 3

**Summary:**

This work examines the estimation of the Radon-Nikodym derivative, which is the ratio of two probability densities. In classical algorithms, an incorrect choice of binary loss function can lead to biased estimates. The author first derived the necessary properties for an appropriate loss function. Based on this analysis, novel loss functions were proposed, demonstrating improved parameter selection in both simulated and real data examples.

**Strengths:**

The theoretical analysis appears solid, and several real datasets are used to demonstrate the proposed loss function. The authors provided detailed comparison between new results and previous analysis.

**Weaknesses:**

Although the author discussed how their results improve upon previous work, they did not elaborate on how their proof differs from prior analyses. Consequently, the challenges of the proof, as well as the novelty and contributions of the theoretical analysis, remain unclear. Additionally, the writing could be improved; including more high-level explanations of the motivation and results would make it easier to follow.

**Questions:**

1. Theorem 1 extends previous results. What is the main challenge in extending these, and what is the key novelty in the proof?
2. Table 1 shows that EW consistently performs best for Amazon Reviews under Importance Weighted Aggregation. Is there any intuition behind this outcome?

---

> ### Author Response · Authors · 2024-11-19
> **Answer**
>
> We thank you for your comments which help to improve our work! Our answers are:
> 1. There were two main challenges.
>         The first challenge was to invert the constructions in [Menon\&Ong, Appendix B] and to non-trivially combine it with several results from [Reid\&Williamson], see the proof of Lemma 5 (updated version). The second challenge was to prove that equal expected Bregman divergences result in equal (up to affine terms) generators, see Lemma~4 (updated version).
>
>         The key novelty consists in the necessity of Eq.(8), which was (to the best of the authors' and all other reviewers knowledge) not explicitly stated before.
> 2. Our intuition is, that Amazon Reviews has more peaky density ratios. But, we are far away from a proof of this intuition.
>
> We thank you again for your time and effort used to review our work.
> If you are satisfied with our answers and extensions, we kindly ask you to take this into account in your final decision.

---

> > ### Comment · Reviewer_Ps3k · 2024-11-25
> > **Response to author**
> >
> > Thank you for addressing my questions. I will maintain my current score.

---

### Official Review · Reviewer_6SXE · 2024-11-02

**Soundness:** 3
**Presentation:** 3
**Contribution:** 3
**Rating:** 6
**Confidence:** 3

**Summary:**

The paper addresses the challenge of estimating the ratio of two probability densities from observations. Typically, this is done using binary classifiers, but the efficacy of the estimators is significantly affected by the choice of the binary loss function used. The authors characterize loss functions that result in statistically favorable density ratio estimations, particularly focusing on achieving low errors in large density ratio values—a departure from classical approaches that perform well on small values. They introduce novel loss functions and demonstrate their application in parameter selection for deep domain adaptation tasks. Numerical experiments and real-world applications illustrate the practical benefits of these loss functions.

**Strengths:**

1. Compared to the related literature, this paper introduces a new framework for constructing novel loss functions, prioritizing an accurate estimation of large density ratio values over smaller ones.
2. It provides a thorough mathematical foundation, characterizing the types of loss functions that align with specific error measures derived from Bregman divergences. The comparison with the related literature is good.
3.  The work shows large practical implementation through empirical data and real application in deep domain adaptation. The simulation work is extensive to demonstrate the effectiveness of the proposed method.

**Weaknesses:**

1. The paper does not delve into the sample complexity of the proposed methods, which could be critical for understanding their efficiency in various scenarios.
2. While it improves estimation for large density values, the impact on performance for smaller values isn't thoroughly explored.
3. A more detailed introduction of the experiments should be considered.

**Questions:**

1. Is the existence of the density ratio always guaranteed? For example, if the two densities have no overlapping support, in which case, the definition seems to fail.  Can the proposed method perform a good estimation?
2. If one distribution has a light tail and the other a heavy tail, how would that impact the estimation?
3. It seems the author considered only a one-dimensional case in the experiment, how about a multi-dimensional case when $d>0$? This is more common in covariate shift problems.
4. What are the biggest difficulties and challenges for deriving the the sample complexity of the proposed methods?

**Details Of Ethics Concerns:**

No details of concerns beyond the above.

---

> ### Author Response · Authors · 2024-11-19
> **Answer**
>
> We thank you for your time and effort used to review our work! our answers to your questions are as follows:
> 1. Indeed, the existence is not always guaranteed. We always assume that $P$ is absolutely continuous w.r.t.~$Q$, i.e.~$P\ll Q$, which guarantees the existences by the Radon-Nikod\'ym derivative. That is, disjoint supports are not allowed.
> 2. In principle, dividing a heavy tail by a light tail should increase the density ratio values in the tail and making it simpler to estimate with our approaches, see also Figure 1. However, we did not test this explicitly.
> 3. All problems in our real world experiments are higher dimensional, bag of words representations of texts (Amazon reviews), images (MiniDomainNet) and time series of body sensor signals (HHAR).
>
> We thank you again for your time and effort used to review our work.
> If you are satisfied with our answers and extensions, we kindly ask you to take this into account in your final decision.

---

> > ### Comment · Reviewer_6SXE · 2024-11-25
> > **Thank you**
> >
> > Thank the authors for their effort and very detailed response to my comments. After going through the other reviewers' comments and the authors' responses, I choose to keep my positive rates.

---

### Official Review · Reviewer_1wrz · 2024-11-03

**Soundness:** 2
**Presentation:** 2
**Contribution:** 1
**Rating:** 6
**Confidence:** 2

**Summary:**

The authors present theoretical results characterizing strictly proper binary loss functions that lead to minimizers of Bregman divergences in probability density estimation. According to these theoretical results, they propose a novel loss function that prioritizes accurate estimation of large density ratio values over smaller ones. They also empirically validate the effectiveness of their proposed loss function through numerical experiments, demonstrating that the novel loss function can lead to improvements in parameter selection for domain adaptation tasks.

**Strengths:**

References to existing research are sufficiently provided.

**Weaknesses:**

#### Major Weaknesses:
1. There are concerns regarding the novelty of the theoretical results presented in this study. Specifically, results such as the necessity of Equation (8) in Theorem 1 appear to be easily derived from findings  in prior work referenced by this study ([1], [2], and [3]). A detailed examination of this issue is provided below.
2. Additionally, the canonical form of the density ratio link, given in Equation (10), does not constitute a new result, as it can be derived from results presented in prior studies ([1]). A detailed examination of this issue is provided below.
3. The proposed loss function, derived from Equation (11) in Section 5, lacks a clear connection to the theoretical results previously established in Section 4.
4. Moreover, it is unclear how the proposed loss function specifically addresses shortcomings associated with existing loss functions. The authors are encouraged to include mathematical analysis, such as theorems, to clarify the properties of the proposed loss function.

#### Minor Weakness:
5. Figures 2 and 3:
    - The axis titles are missing, making it difficult to interpret the graphs.
    - In particular, Figure 3 lacks explanatory labels for each axis, which are needed to understand these experimental results.


---
Hereafter, details of the major weaknesses, specifically Weaknesses 1 and 2, are discussed.

#### About major weakness 1:
Equation (8) can be derived as follows:
1. From Theorem 4 in [2], we know that $B_{\phi} = B_{\phi'}$ for any $\phi'$ with $\phi'(y) = \phi(y) + c_2 y + c_1$, where $c_2$ and $c_1$ are constants. This fact implies that terms such as $\hat{\eta} c_2 $ and $c_1$ in the definition of $\gamma(\cdot)$ (line 236) are redundant.
2. From Theorem 4 in [1], we know that $L(\eta, \mu) = \underline{L}(z) + (\eta - \mu) \underline{L}'(\mu)$.
3. Additionally, we have $\underline{L}(\eta) = - \phi(\eta)$ because $\underline{L} = L(\eta, \eta) = \eta l_1(\eta) + (1 - \eta) l_2(\eta) = \gamma(\eta) = - \phi(\eta)$.
4. Thus, $L(\eta, \Psi^{-1}(y)) = - \phi(\Psi^{-1}(y)) - (\eta - \Psi^{-1}(y)) \phi' (\Psi^{-1}(y))$, where Equation (8) represents the cases for $\eta = 0$ and $\eta = 1$ in this equation.

#### About major weakness 2:
From Corollary 3 in [1] and the discussion in Section 6.1 of [1], it follows that $(g^{-1}_{can})' (c) = w(c) = - \underline{L}''(c) = \phi''(c)$. There appears to be no significant difference between Equation (10) and this equation.

---

[1] Reid, M. D., & Williamson, R. C. (2010). Composite binary losses. The Journal of Machine Learning Research, 11, 2387-2422.

[2] Reid, M. D., & Williamson, R. C. (2011). Information, Divergence and Risk for Binary Experiments. Journal of Machine Learning Research, 12(3).

[3] Menon, A., & Ong, C. S. (2016, June). Linking losses for density ratio and class-probability estimation. In International Conference on Machine Learning (pp. 304-313). PMLR.

**Questions:**

- Considering major weaknesses 1 and 2 discussed above, could you provide more additional discussions to clarify the novel contributions of your study?
- Considering major weaknesses 3 and 4 discussed above, could you provide further detailed information to elucidate the effectiveness of your approach discussed in Section 5?

---

> ### Author Response · Authors · 2024-11-19
> **Answer**
>
> Thank you very much for your comments which help to improve our work! Our answers are as follows:
> - While we agree with the essential structure of your proof, we disagree with your points 1 and 3. One counterexample for your point 3 (stating $\underline{L}(\eta)=-\phi(\eta)$) is KuLSIF (Example 1) with
>     $$
>     \ell_1(y):=-y, \ell_{-1}(y):=\frac{y^2}{2},\Psi^{-1}(y):=\frac{y}{1+y}
>     $$
>     such that
>     $$
>     \underline{L}(\eta)=\eta\ell_1(\Psi(\eta))+(1-\eta)\ell_{-1}(\Psi(\eta))= -\frac{\eta^2}{1+\eta}+\frac{1}{1+\eta}\frac{\eta^2}{2}=\frac{-\eta^2}{2+2\eta}
>     $$
>     and, by [3, Proposition 3],
>     $$
>     -\phi(\eta)=(1+\eta) \underline{L}(\frac{\eta}{1+\eta})=-\frac{\eta^2}{2}
>     $$
>     which are different for $\underline{L}(\frac{1}{2})=\frac{1}{4}\neq\frac{1}{8}=-\phi(\frac{1}{2})$.
>     Instead of point 3, Lemma 5 can be used (updated version).
>
>     Furthermore, concerning your point 1, we note that Theorem 4 in [2] proves equality of Bregman divergences for equal (up to affine terms) generators. However, we need equality (up to affine terms) of generators when we know equality of *expected* Bregman generator functions, i.e., the other direction.
>     Instead of your point 1, our Lemma 4 (updated version) can be used, which proves the equality of generators by constructing two witness probability measures in a non-trivial two-page proof.
>
>     To improve our presentation, we extended the title, abstract, the introduction, Remark 1 and we split Lemma 4 (submitted version) in two Lemmas (4 and 5 in updated version), see the green text in updated submission document.
> - We disagree that the canonical loss in [1, Section 6.1] is the same as the loss derived from $g_\mathrm{can}$.
>     First, from the same KuLSIF example as used above, we see that the last equality of your derivation does not hold, since $-\underline{L}''(\eta)\neq \phi''(\eta)$, e.g., for $\eta=\frac{1}{2}$.
>     Second, the equation $\Psi'(c)=w(c)$ in [1, Section 6.1] leads to a loss function that differs from the one derived by Eq.(10) by a non-trivial factor of $(1+c)^3$, see Remark 3 (updated version).
> - The loss functions follow directly from Section 4 by plugging $\phi$ and $g_\mathrm{can}$ in Eq.(8). We added a remark and uploaded a simple Mathematica notebook for algebraic correctness.
> - We prove that our novel loss functions assign larger weight to large values of density ratio values than to smaller ones; which is in contrast to related approaches (see [3]).
>     Moreover, we illustrate this behavior in numerical examples (Figure 1, Figure 2, Figure 3).
>     Finally, we show that our losses outperform others in extensive state-of-the-art benchmark experiments.
>
> We thank you again for your time and effort used to review our work.
> If you are satisfied with our answers and extensions in the updated manuscript, we kindly ask you to take this into account in your final decision.

---

> > ### Comment · Reviewer_1wrz · 2024-11-25
> >
> > Thank you for your response. I acknowledge that the updated version addresses some of my concerns, and I have adjusted my score accordingly.

---

### Official Review · Reviewer_FZmy · 2024-11-03

**Soundness:** 3
**Presentation:** 2
**Contribution:** 2
**Rating:** 6
**Confidence:** 2

**Summary:**

The authors characterize the set of loss functions that, when used in density ratio estimation for binary classification, lead to the minimization of a particular Bregman divergence. This approach is motivated by the observation that some commonly used losses (such as exponential or logistic) yield density ratio estimates that minimize a similar Bregman divergence expression. After identifying these losses, they design a new family of losses aimed at accurately estimating large values of the density ratio, in contrast to standard losses that focus on estimating small ratio values. They apply these designed losses to estimate density ratios in Gaussian RKHS and for unsupervised domain adaptation.

**Strengths:**

1. The paper appears to be self-contained and introduces all the tools and notation it uses.

2. The experimental results are fairly extensive for a theoretical paper.

3. The theoretical results, especially Lemma 4 in the appendix, seem to rely on non-trivial applications of several previously established results.

**Weaknesses:**

1. The presentation of the paper could be significantly improved. The paper uses heavy notation — for example, understanding $B_{-\underline{L}^\circ}(\beta,g\circ f)$ requires substantial effort. Another clear example is Remark 1, which is completely incomprehensible unless the reader is already familiar with everything it covers.

2. The novelty of this work is unclear. I believe the authors would agree that the main contribution of this paper is theoretical and primarily represented by Theorem 1. However, given Remark 1, it is not evident how substantial this theorem’s contribution really is.

3. It is not clear why one should focus on minimizing equation (1). According to the beginning of Section 2, there are "many" density ratio estimation methods that lead to minimizing (1), with four examples provided. What does "many" mean in this context, and why should one limit themself to this specific type of minimizers?

4. Some parts of the experimental results, like Section 6.2, contain too much irrelevant information for readers, making it easy to miss the main points. I would consider moving some of this information to the appendices.

**Questions:**

1. Could you briefly outline the most significant theoretical contribution of your paper and the challenges involved in achieving it?

2. Could you explain the intuition behind the flatness of your method in Figure 2? If the method aims to estimate the ratio accurately for high values, shouldn’t it follow the top of the curve closely? Furthermore, in the lower row of the figure for $\alpha=0.01$, I am not sure I understand why one would prefer your estimate over KuLSIF.

3. Since you mention that standard methods prioritize estimating smaller values and you focus on higher values, what happens if one applies standard methods to the inverse ratio, i.e., estimating $dQ/dP$?

Typo: In footnote 2, the next-to-last equation contains probabilities that should be conditioned on $x$: it should be $\rho(y=1∣x)\rho(x)$ instead of $\rho(x,y=1)\rho(x)$.

---

> ### Author Response · Authors · 2024-11-19
> **Answer**
>
> We thank you for your comments which help to improve our work! We especially appreciate your notes on extensiveness of experiments and non-triviality of theoretical results.
> Our answers are as follows:
> - We agree that the theory requires some effort to learn. To clarify the confusions, we added several remarks in the updated paper to clarify the main points.
> - Our contribution is to *provide a complete set of techniques for designing loss functions for Algorithm 1 that (a) prioritize the estimation of large density ratio values and (b) allow high performance in practice.* This requires three main contributions:
>   + Characterization: The technical characterization (Theorem 1) of loss functions satisfying Eq. (1).
>   + Losses: The design of novel losses (Section 5) for increasing weight functions $\phi''$.
>   + Experiments: State-fo-the-art performance in benchmark experiments.
> - Eq.(1) is the error of any estimator computed by Algorithm 1. The motivation for using Algorithm 1 is the same as the motivation for using generative adversarial networks or classifier-based statistical tests: the relation between discriminative and generative approaches.
> - We completely agree with the amount of information. However, trading this off with reproducibility of experiments and the wish of other reviewers to extend this section, we decided to leave this Section as it is.
> - Minor typo: Sorry, we are confused; what difference between the two do you want to highlight?
>
> We thank you for your detailed questions which especially help to improve our presentation. Our answers are:
> 1.  The main contribtions are stated above. The corresponding challenges were:
>     \begin{itemize}
> - Characterization: There were two main challenges.
>         The first challenge was to invert the constructions in [Menon\&Ong, Appendix B] and to non-trivially combine it with several results from [Reid\&Williamson], see the proof of Lemma 5 (updated version). The second challenge was to prove that equal expected Bregman divergences result in equal (up to affine terms) generators, see Lemma 4 (updated version) and it's associated non-trivial two-page proof.
> - Losses: The main contribution was to find suitable increasing weight functions which can be efficiently optimize and clearly lead to better weightings than related approaches in our numerical examples.
> - Experiments: The main challenge was to train 9174 neural networks with 11 different deep learning algorithms on 484 parameter selection tasks.
> 2. Figure 2 shows the effect of regularization using the experiment of [Zellinger et al., 2023, JMLR]. The Figure is not aimed for benchmarking, but rather visualizes the effect of better prediction larger values (flatness). It shows the consequently worse approximations for smaller values.
> 3. This is a good idea which, however, requires the inverse ratio to exist.
>
> We thank you again for your time and effort used to review our work.
> If you are satisfied with our answers and extensions, we kindly ask you to take this into account in your final decision.

---

> > ### Comment · Reviewer_FZmy · 2024-11-25
> >
> > Thank you for your response. I acknowledge that the updated version addresses some of my concerns, and I have adjusted my score accordingly.

---

### Author Response · Authors · 2024-11-19
**General Answer**

We thank all reviewers for the invested time, your efforts and the constructive comments which help to improve our work! We especially appreciate:
- Consensus about correctness of all proofs and the effectiveness in empirical evaluations
- That our loss functions are (to the best of the authors and reviewers’ knowledge) the first losses for density ratio estimation that assign higher weight to larger values
- Explicit emphasis of 55K4, Ps3k, 6SXE, FZmy on the extensiveness of our empirical evaluations

Reviewers' concerns are mainly about (a) the presentation of contributions and (b) the novelty beyond loss functions and SoTA performance. Our main answers are:
- Concerning (a): Our contribution is to provide a complete set of techniques for designing loss functions for Algorithm 1 that (a) prioritize the estimation of large density ratio values and (b) allow high performance in practice. Three key contributions are given:
  + Characterization: The technical characterization (Theorem 1) of loss functions satisfying Eq. (1).
  + Losses: The design of novel losses (Section~5) for increasing weight functions $\phi''$.
  + Experiments: State-fo-the-art performance in benchmark experiments involving 9174 neural networks, 484 parameter selection tasks and three datasets for text, images and human body sensor signals.

  We added several comments around the manuscript to clarify the main contributions.
- Concerning (b): The reviewers agree (a) on the novelty of the loss functions in Section 5 and (b) its state-of-the-art improvements on extensive empirical evaluations. Our third novelty is the (c) necessity of the loss function in Eq. (8).
    Our proof has four non-trivial components: (a) the inversion of the constructions developed in [Menon\&Ong, Appendix B] done in Lemma 5 (updated version), (b) the application of several results from [Reid\&Williamson], (c) applying Savage's theorem and (d) our technical Lemma 4 (updated version) and it's associated non-trivial two-page proof.

---

### Meta-Review · Area_Chair_X7mn · 2024-12-16

**Metareview:**

The paper addresses an important problem, definitely in a proper way (no pun intended). In my decision to accept the paper, I have taken into account the initial opinions and their revised version, the responses of the authors, and the paper (I read it).

It is important to acknowledge that the paper adopts a first principle approach to the DR problem. I would have loved the paper to go all the way forward, up to deriving in a more formal way the content of Section 5, but this part is definitely a tricky bit (this is not to say that there is no principled / general way to address it). At least, the problem of tackling large DRs is approached in a right way, from the standpoint of the design of the loss function and it is an interesting work that tries to alleviate the problem of estimating large DRs by pointing at specific losses using this theory. I commend the authors not just for their result, but also for the non-trivial path adopted. Reviewers (in particular 1wrz) had a legitimate question on the significance of the material brought to the paper, but I think the authors have made a decent explanation of their contribution. Personally, I do not just consider the result but also the *path* chosen by the authors to addressing their problem as a positive contribution which deserves publication. I can only encourage the authors to make this even more clear in the camera ready version.

The authors may remark that addressing the sample complexity of their method may take advantage of the property of Bregman divergences that authorizes to link DR and class probability estimation (a reduction may also be useful to tackle this additional problem). This property is in fact a specific case of a property on the perspective transform of Bregman divergences. For a general treatment and further use, see "A scaled Bregman Theorem with Applications" by Nock, Menon and Ong, NeurIPS'16.

**Additional Comments On Reviewer Discussion:**

I particularly appreciated the attention to details of the authors in answering technical questions (e.g. 1wrz).

---

### Decision · Program_Chairs · 2025-01-22

Accept (Poster)